# When Can We Learn General-Sum Markov Games with a Large Number of Players Sample-Efficiently?

**Ziang Song**
School of Mathematical Sciences, Peking University
songziang@pku.edu.cn

**Song Mei**
Department of Statistics, UC Berkeley
songmei@berkeley.edu

**Yu Bai**
Salesforce Research
yu.bai@salesforce.com

## Abstract

Multi-agent reinforcement learning has made substantial empirical progresses in solving games with a large number of players. However, theoretically, the best known sample complexity for finding a Nash equilibrium in general-sum games scales exponentially in the number of players due to the size of the joint action space, and there is a matching exponential lower bound. This paper investigates what learning goals admit better sample complexities in the setting of $m$-player general-sum Markov games with $H$ steps, $S$ states, and $A_i$ actions per player. First, we design algorithms for learning an $\varepsilon$-Coarse Correlated Equilibrium (CCE) in $\widetilde{\mathcal{O}}(H^5 S \max_{i \le m} A_i / \varepsilon^2)$ episodes, and an $\varepsilon$-Correlated Equilibrium (CE) in $\widetilde{\mathcal{O}}(H^6 S \max_{i \le m} A_i^2 / \varepsilon^2)$ episodes. This is the first line of results for learning CCE and CE with sample complexities polynomial in $\max_{i \le m} A_i$. Our algorithm for learning CE integrates an adversarial bandit subroutine which minimizes a weighted swap regret, along with several novel designs in the outer loop. Second, we consider the important special case of Markov Potential Games, and design an algorithm that learns an $\varepsilon$-approximate Nash equilibrium within $\widetilde{\mathcal{O}}(S \sum_{i \le m} A_i / \varepsilon^3)$ episodes (when only highlighting the dependence on $S$, $A_i$, and $\varepsilon$), which only depends linearly in $\sum_{i \le m} A_i$ and significantly improves over existing efficient algorithms in the $\varepsilon$ dependence. Overall, our results shed light on what equilibria or structural assumptions on the game may enable sample-efficient learning with many players.

## 1 Introduction

Multi-agent reinforcement learning (RL) has achieved substantial recent successes in solving artificial intelligence challenges such as GO (Silver et al., 2016; 2018), multi-player games with team play such as Starcraft (Vinyals et al., 2019) and Dota2 (Berner et al., 2019), behavior learning in social interactions (Baker et al., 2019), and economic simulation (Zheng et al., 2020; Trott et al., 2021). In many applications, multi-agent RL is able to yield high quality policies for multi-player games with a large number of players (Wang et al., 2016; Yang et al., 2018).

Despite these empirical progresses, theoretical understanding of when we can sample-efficiently solve multi-player games with a large number of players remains elusive, especially in the setting of multi-player Markov games. A main bottleneck here is the *exponential blow-up* of the *joint action space*—The total number of joint actions in a generic game with simultaneous plays is equal to the product of the number of actions for each player, which scales exponentially in the number of players. Such an exponential dependence is indeed known to be unavoidable in the worst-case for certain standard problems. For example, for learning an approximate Nash equilibrium from payoff queries in an one-step multi-player general-sum game, the query complexity lower bound of Chen et al. (2015) and Rubinstein (2016) shows that at least exponentially many queries (samples) is

required, even when each player only has two possible actions and the query is noiseless. Moreover, for learning Nash equilibrium in Markov games, the best existing sample complexity upper bound also scales with the size of the joint action space (Liu et al., 2021).

Nevertheless, these exponential lower bounds do not completely rule out interesting theoretical inquiries—there may well be other notions of equilibria or additional structures within the game that allow us to learn with a better sample complexity. This motivates us to ask the following

> **Question**: When can we solve general-sum Markov games with sample complexity milder than exponential in the number of players?

This paper makes steps towards answering the above question by considering multi-player general-sum Markov games (MGs) with $m$ players, $H$ steps, $S$ states, and $A_i$ actions per player. We make two lines of investigations: (1) Can we learn alternative notions of equilibria with better sample complexity than learning Nash; (2) Can the Nash equilibrium be learned with better sample complexity under additional structural assumptions on the game. This paper makes contributions on both ends, which we summarize as follows.

- We first design an algorithm that learns the $\varepsilon$-approximate Coarse Correlated Equilibrium (CCE) with $\widetilde{\mathcal{O}}(H^5 S \max_{i \in [m]} A_i / \varepsilon^2)$ episodes of play (Section 3). Our algorithm CCE-V-LEARNING is a multi-player adaptation of the Nash V-Learning algorithm of Bai et al. (2020).

- We design an algorithm CE-V-LEARNING which learns the stricter notion of $\varepsilon$-approximate Correlated Equilibrium (CE) with $\widetilde{\mathcal{O}}(H^6 S \max_{i \in [m]} A_i^2 / \varepsilon^2)$ episodes of play (Section 4). For Markov games, these are the first line of sample complexity results for learning CE and CCE that only scales polynomially with $\max_{i \in [m]} A_i$, and improves significantly in the $A_i$ dependency over the current best algorithm which scales with $\prod_{i \in [m]} A_i$.

- Technically, our algorithm CE-V-LEARNING makes several major modifications over CCE-V-LEARNING in order to learn the CE (Section 4.2). Notably, inspired by the connection between CE and *low swap-regret* learning, we use a mixed-expert Follow-The-Regularized Leader algorithm within its inner loop to achieve low swap-regret for a particular adversarial bandit problem. Our analysis also contains new results for adversarial bandits on weighted swap regret and weighted regret with predicable weights, which may be of independent interest.

- Finally, we consider learning Nash equilibrium in *Markov Potential Games* (MPGs), an important subclass of general-sum Markov games. By a reduction to single-agent RL, we design an algorithm NASH-CA that achieves $\widetilde{\mathcal{O}}(\Phi_{\max} H^3 S \sum_{i \in [m]} A_i / \varepsilon^3)$ sample complexity, where $\Phi_{\max} \leq Hm$ is the bound on the potential function (Section 5). Compared with the recent result of Leonardos et al. (2021), we significantly improves the $\varepsilon$ dependence from their $1/\varepsilon^6$.

## 1.1 RELATED WORK

**Learning equilibria in general-sum games** The sample (query) complexity of learning Nash, CE, and CCE from samples in *one-step* (i.e. normal form) general-sum games with $m$ players and $A_i$ actions per player has been studied extensively in literature (Hart & Mas-Colell, 2000; Hart, 2005; Stoltz, 2005; Cesa-Bianchi & Lugosi, 2006; Blum & Mansour, 2007; Fearnley et al., 2015; Babichenko & Barman, 2015; Chen et al., 2015; Fearnley & Savani, 2016; Goldberg & Roth, 2016; Babichenko, 2016; Rubinstein, 2016; Hart & Nisan, 2018). It is known that learning Nash equilibrium requires exponential in $m$ samples in the worst case (Rubinstein, 2016), whereas CE and CCE admit efficient $\text{poly}(m, \max_{i \leq m} A_i)$-sample complexity algorithms by independent no-regret learning (Hart & Mas-Colell, 2000; Hart, 2005; Syrgkanis et al., 2015; Goldberg & Roth, 2016; Chen & Peng, 2020; Daskalakis et al., 2021). Our results for learning CE and CCE can be seen as extension of these works into Markov games. We remark that even when the game is fully known, the computational complexity for finding Nash in general-sum games is PPAD-hard (Daskalakis, 2013).

**Markov games** Markov games (Shapley, 1953; Littman, 1994) is a widely used framework for game playing with sequential decision making, e.g. in multi-agent reinforcement learning. Algorithms with asymptotic convergence have been proposed in the early works of Hu & Wellman (2003); Littman (2001); Hansen et al. (2013). A recent line of work studies the non-asymptotic sample complexity for learning Nash in two-player zero-sum Markov games (Bai & Jin, 2020; Xie et al., 2020; Bai et al.,

2020; Zhang et al., 2020; Liu et al., 2021; Chen et al., 2021; Jin et al., 2021; Huang et al., 2021) and learning various equilibria in general-sum Markov games (Liu et al., 2021; Bai et al., 2021), building on techniques for learning single-agent Markov Decision Processes sample-efficiently (Azar et al., 2017; Jin et al., 2018). Learning the Nash equilibrium in general-sum Markov games are much harder than that in zero-sum Markov games. Liu et al. (2021) present the first line of results for learning Nash, CE, and CCE in general-sum Markov games; however their sample complexity scales with $\prod_{i \leq m} A_i$ due to the model-based nature of their algorithm. Algorithms for computing CE in extensive-form games has been widely studied (Von Stengel & Forges, 2008; Celli et al., 2020; Farina et al., 2021; Morrill et al., 2021), though we remark Markov games and extensive-form games are different frameworks and our results do not imply each other.

**Markov potential games** Lastly, a recent line of works considers Markov potential games (Macua et al., 2018; Leonardos et al., 2021; Zhang et al., 2021), a subset of general-sum Markov games in which the Nash equilibrium admits more efficient algorithms. Leonardos et al. (2021) gives a sample-efficient algorithm based on the policy gradient method (Agarwal et al., 2021). The special case of Markov cooperative games is studied empirically in e.g. Lowe et al. (2017); Yu et al. (2021). For one step potential games, Kleinberg et al. (2009); Palaiopanos et al. (2017); Cohen et al. (2017a) show the convergence to Nash equilibria of no-regret dynamics.

## 2 PRELIMINARIES

We present preliminaries for multi-player general-sum Markov games as well as the solution concept of (approximate) Nash equilibrium. Alternative solution concepts and other concrete subclasses of Markov games considered in this paper will be defined in the later sections.

**Markov games** A multi-player general sum Markov game (MG; Shapley (1953); Littman (1994)) with $m$ players can be described by a tuple $\text{MG}(H, \mathcal{S}, \{\mathcal{A}_i\}_{i=1}^m, \mathbb{P}, \{r_i\}_{i=1}^m)$, where $H$ is the episode length, $\mathcal{S}$ is the state space with $|\mathcal{S}| = S$, $\mathcal{A}_i$ is the action space for the $i^{\text{th}}$ player with $|\mathcal{A}_i| = A_i$. Without loss of generality, we assume $\mathcal{A}_i = [A_i]$. We let $\boldsymbol{a} := (a_1, \cdots, a_m)$ denote the vector of joint actions taken by all the players and $\mathcal{A} = \mathcal{A}_1 \times \cdots \times \mathcal{A}_m$ denote the joint action space. Throughout this paper we assume that $S$ and $A_i$ are finite. The transition probability $\mathbb{P} = \{\mathbb{P}_h\}_{h \in [H]}$ is the collection of transition matrices, where $\mathbb{P}_h(\cdot|s, \boldsymbol{a}) \in \Delta_{\mathcal{S}}$ denotes the distribution of the next state when actions $\boldsymbol{a}$ are taken at state $s$ at step $h$. The rewards $r_i = \{r_{h,i}\}_{h \in [H], i \in [m]}$ is the collection of reward functions for the $i^{\text{th}}$ player, where $r_{h,i}(s, \boldsymbol{a}) \in [0, 1]$ gives the deterministic[1] reward of $i^{\text{th}}$ player if actions $\boldsymbol{a}$ are taken at state $s$ at step $h$. Without loss of generality, we assume the initial state $s_1$ is deterministic. A key feature of general-sum games is that the rewards $r_i$ are in general different for each player $i$, and the goal of each player is to maximize her own cumulative reward.

**Markov product policy, value function** A product policy is a collection of $m$ policies $\pi := \{\pi_i\}_{i \in [m]}$ where $\pi_i$ is the general (potentially history-dependent) policy for the $i$-th player. We first focus on the case of Markov product policies, in which $\pi_i = \{\pi_{h,i} : \mathcal{S} \to \Delta_{\mathcal{A}_i}\}_{h \in [H]}$, and $\pi_{h,i}(a_i|s)$ is the probability for the $i^{\text{th}}$ player to take action $a_i$ at state $s$ at step $h$. For a policy $\pi$ and $i \in [m]$, we use $\pi_{-i} := \{\pi_j\}_{j \in [m], j \neq i}$ to denote the policy of all but the $i^{\text{th}}$ player. The value function $V_{h,i}^\pi(s) : \mathcal{S} \to \mathbb{R}$ is defined as the expected cumulative reward for the $i^{\text{th}}$ player when policy $\pi$ is taken starting from state $s$ and step $h$:

$$V_{h,i}^\pi(s) := \mathbb{E}_\pi \left[ \sum_{h'=h}^H r_{h',i}(s_{h'}, \boldsymbol{a}_{h'}) \middle| s_h = s \right]. \tag{1}$$

**Best response & Nash equilibrium** For any product policy $\pi = \{\pi_i\}_{i \in [m]}$, the *best response* for the $i^{\text{th}}$ player against $\pi_{-i}$ is defined as any policy $\pi^\dagger$ such that $V_{1,i}^{\pi^\dagger, \pi_{-i}}(s_1) = \sup_{\pi_i'} V_{1,i}^{\pi_i', \pi_{-i}}(s_1)$. For any Markov product policy, this best response is guaranteed to exist (and be Markov) as the above maximization problem is equivalent to solving a Markov Decision Process (MDP) for the $i^{\text{th}}$ player. We will also use the notation $V_{1,i}^{\dagger, \pi_{-i}}(s_1)$ to denote the above value function $V_{1,i}^{\pi^\dagger, \pi_{-i}}(s_1)$.

---

[1]Our results can be straightforwardly generalized to Markov games with stochastic rewards.

We say $\pi$ is a *Nash equilibrium* (e.g. Nash (1951); Pérolat et al. (2017)) if all players play the best response against other players, i.e., for all $i \in [m]$,

$$V_{1,i}^{\pi}(s_1) = V_{1,i}^{\dagger, \pi_{-i}}(s_1).$$

Note that in general-sum MGs, there may exist multiple Nash equilibrium policies with different value functions, unlike in two-player zero-sum MGs (Shapley, 1953). To measure the suboptimality of any policy $\pi$, we define the NE-gap as

$$\text{NE-gap}(\pi) := \max_{i \in [m]} \left[ \sup_{\mu_i} V_{1,i}^{\mu_i, \pi_{-i}}(s_1) - V_{1,i}^{\pi}(s_1) \right].$$

For any $\varepsilon \geq 0$, we say $\pi$ is $\varepsilon$-approximate Nash equilibrium ($\varepsilon$-Nash) if $\text{NE-gap}(\pi) \leq \varepsilon$.

**General correlated policy & Its best response**   A general correlated policy $\pi$ is a set of $H$ maps $\pi := \{\pi_h : \Omega \times (\mathcal{S} \times \mathcal{A})^{h-1} \times \mathcal{S} \to \Delta_{\mathcal{A}}\}_{h \in [H]}$. The first argument of $\pi_h$ is a random variable $\omega \in \Omega$ sampled from some underlying distribution, and the other arguments contain all the history information and the current state information (unlike Markov policies in which the policies only depend on the current state information). The output of $\pi_h$ is a general distribution of actions in $\mathcal{A} = \mathcal{A}_1 \times \cdots \times \mathcal{A}_m$ (unlike product policies in which the action distribution is a product distribution).

For any correlated policy $\pi = \{\pi_h\}_{h \in [H]}$ and any player $i$, we can define a marginal policy $\pi_{-i}$ as a set of $H$ maps $\pi_{-i} := \{\pi_{h,-i} : \Omega \times (\mathcal{S} \times \mathcal{A})^{h-1} \times \mathcal{S} \to \Delta_{\mathcal{A}_{-i}}\}_{h \in [H]}$ where $\mathcal{A}_{-i} := \mathcal{A}_1 \times \cdots \times \mathcal{A}_{i-1} \times \mathcal{A}_{i+1} \times \cdots \times \mathcal{A}_m$, and the output of $\pi_{h,-i}$ is defined as the marginal distribution of the output of $\pi_h$ restricted to the space $\mathcal{A}_{-i}$. For any general correlated policy $\pi$, we can define its initial state value function $V_{1,i}^{\pi}(s_1)$ similar as (1). The best response value of the $i^{\text{th}}$ player against $\pi$ is $V_{1,i}^{\dagger, \pi_{-i}}(s_1) = \sup_{\mu_i} V_{1,i}^{\mu_i, \pi_{-i}}(s_1)$, where $V_{1,i}^{\mu_i, \pi_{-i}}(s_1)$ is the value function of the policy $(\mu_i, \pi_{-i})$ (the $i^{\text{th}}$ player plays according to general policy $\mu_i$, and all other players play according to $\pi_{-i}$), and the supremum is taken over all general policy $\mu_i$ of the $i^{\text{th}}$ player.

**Learning setting**   Throughout this paper we consider the interactive learning (i.e. exploration) setting where algorithms are able to play episodes within the MG and observe the realized transitions and rewards. Our focus is on the PAC sample complexity (i.e. number of episodes of play) for any learning algorithm to output an approximate equilibrium.

## 2.1 Exponential lower bound for learning approximate Nash equilibrium

The focus of this paper is the setting where the number of players $m$ is large. Intuitively, as the joint action space has size $|\mathcal{A}| = \prod_{i=1}^{m} A_i$ which scales exponentially in $m$ (if each $A_i \geq 2$), naive algorithms for learning Nash equilibrium may learn all $r_i(\boldsymbol{a})$ by enumeratively querying all $\boldsymbol{a} \in \mathcal{A}$, and this costs exponential in $m$ samples. Unfortunately, recent work shows that such exponential in $m$ dependence is unavoidable in the worst-case for any algorithm—there is an $\exp(\Omega(m))$ sample complexity lower bound for learning approximate Nash, even in *one-step* general-sum games (Chen et al., 2015; Rubinstein, 2016) (see Proposition A.2 for formal statement).

This suggests that the Nash equilibrium as a solution concept may be too hard to learn efficiently for MGs with a large number of players, and calls for alternative solution concepts or additional structural assumptions on the game in order to achieve an improved $m$ dependence.

## 3 Efficient Learning of Coarse Correlated Equilibria (CCE)

Given the difficulty of learning Nash when the number of players $m$ is large , we consider learning other relaxed notions of equilibria for general-sum MGs. Two standard notions of equilibria for games are the Correlated Equilibrium (CE) and Coarse Correlated Equilibrium (CCE), and they satisfy $\{\text{Nash}\} \subset \{\text{CE}\} \subset \{\text{CCE}\}$ for general-sum MGs (Nisan et al., 2007).

We begin by considering learning CCE (most relaxed notion above) for Markov games.

**Definition 1** ($\varepsilon$-approximate CCE for general-sum MGs)**.** *We say a (general) correlated policy $\pi$ is an $\varepsilon$-approximate Coarse Correlated Equilibrium ($\varepsilon$-CCE) if*

$$\max_{i \in [m]} \left( V_{1,i}^{\dagger, \pi_{-i}}(s_1) - V_{1,i}^{\pi}(s_1) \right) \leq \varepsilon.$$

*We say $\pi$ is an (exact) CCE if the above is satisfied with $\varepsilon = 0$.*

The following result shows that there exists an algorithm that can learn an $\varepsilon$-approximate CCE in general-sum Markov games within $\widetilde{\mathcal{O}}(H^5 S \max_{i \in [m]} A_i / \varepsilon^2)$ episodes of play.

**Theorem 2** (Learning $\varepsilon$-approximate CCE for general-sum MGs). *Suppose we run the* CCE-V-LEARNING *algorithm (Algorithm 4) for all $m$ players and*

$$K \geq \mathcal{O}\left( \frac{H^5 S \max_{i \in [m]} A_i \iota}{\varepsilon^2} + \frac{H^4 S \iota^3}{\varepsilon} \right)$$

*episodes ($\iota = \log(m \max_{i \in [m]} A_i HSK/(p\varepsilon))$ is a log factor). Then with probability at least $1 - p$, the certified policy $\widehat{\pi}$ defined in Algorithm 2 is an $\varepsilon$-CCE, i.e. $\max_{i \in [m]} (V_{1,i}^{\dagger, \widehat{\pi}_{-i}}(s_1) - V_{1,i}^{\widehat{\pi}}(s_1)) \leq \varepsilon$.*

**Mild dependence on action space** For small enough $\varepsilon$, the sample complexity featured in Theorem 2 scales as $\widetilde{\mathcal{O}}(H^5 S \max_{i \in [m]} A_i / \varepsilon^2)$. Most notably, this is the first algorithm that scales with $\max_{i \in [m]} A_i$, and exhibits a sharp difference in learning Nash and learning CCE in view of the $\exp(\Omega(m))$ lower bound for learning Nash in Proposition A.2. Indeed, existing algorithms such as Multi-Nash-VI Algorithm with CCE subroutine (Liu et al., 2021) does require $\widetilde{\mathcal{O}}(H^4 S^2 \prod_{i=1}^m A_i / \varepsilon^2)$ episodes of play, which scales with $\prod_{i \in [m]} A_i$ due to its model-based nature. We achieve significantly better dependence on $A_i$ and also $S$, though slightly worse $H$ dependence.

**Overview of algorithm and techniques** Our CCE-V-LEARNING algorithm (deferred to Appendix B.1 due to space limit) is a multi-player adaptation of the Nash V-Learning algorithm of Bai et al. (2020); Tian et al. (2021) for learning Nash equilibria in two-player zero-sum MGs. Similar as Bai et al. (2020), we show that this algorithm enjoys a "no-regret" like guarantee for each player at each $(h, s)$ (Lemma B.3). We also adopted the choice of hyperparameters in Tian et al. (2021) so that the sample complexity has a slightly better dependence in $H$. When combined with the "certified correlated policy" procedure (Algorithm 2), our algorithm outputs a policy that is $\varepsilon$-CCE. Our certified policy procedure is adapted from the certified policy of Bai et al. (2020), and differs in that ours output a *correlated* policy for all the players whereas Bai et al. (2020) outputs a product policy. The key feature enabling this $\max_{i \in [m]} A_i$ dependence is that this algorithm uses decentralized learning for each player to learn the value function ($V$), instead of learning the $Q$ function (as in Liu et al. (2021)) that requires sample size scales as $\prod_{i \in [m]} A_i$. The proof of Theorem 2 is in Appendix B.

## 4 EFFICIENT LEARNING OF CORRELATED EQUILIBRIA (CE)

In this section, we move on to considering the harder problem of learning Correlated Equilibria (CE). We first present the definition of CE in Markov games.

**Definition 3** (Strategy modification for $i^{\text{th}}$ player). *A strategy modification $\phi := \{\phi_{h,s}\}_{(h,s) \in [H] \times \mathcal{S}}$ for player $i$ is a set of $H \times S$ functions $\phi_{h,s} : (\mathcal{S} \times \mathcal{A})^{h-1} \times \mathcal{A}_i \to \mathcal{A}_i$. A strategy modification $\phi$ can be composed with any policy $\pi$ to give a modified policy $\phi \diamond \pi$ defined as follows: At any step $h$ and state $s$ with the history information $\tau_{h-1} = (s_1, \boldsymbol{a}_1, \cdots, s_{h-1}, \boldsymbol{a}_{h-1})$, if $\pi$ chooses to play $\boldsymbol{a} = (a_1, \ldots, a_m)$, the modified policy $\phi \diamond \pi$ will play $(a_1, \ldots, a_{i-1}, \phi_{h,s}(\tau_{h-1}, a_i), a_{i+1}, \ldots, a_m)$. We use $\Phi_i$ denote the set of all possible strategy modifications for player $i$.*

**Definition 4** ($\varepsilon$-approximate CE for general-sum MGs). *We say a (general) correlated policy $\pi$ is an $\varepsilon$-approximate CE ($\varepsilon$-CE) if*

$$\max_{i \in [m]} \sup_{\phi \in \Phi_i} \left( V_{1,i}^{\phi \diamond \pi}(s_1) - V_{1,i}^{\pi}(s_1) \right) \leq \varepsilon.$$

*We say $\pi$ is an (exact) CE if the above is satisfied with $\varepsilon = 0$.*

Our definition of CE follows (Liu et al., 2021) and is a natural generalization of the CE for the well-studied special case of one-step (i.e. normal form) games (Nisan et al., 2007).

### 4.1 ALGORITHM DESCRIPTION

Our algorithm CE-V-LEARNING (Algorithm 1) builds further on top of CCE-V-LEARNING and Nash V-Learning, and makes several novel modifications in order to learn the CE. The key feature of CE-V-LEARNING is that it uses a weighted swap regret algorithm (mixed-expert FTRL) for every $(s, h, i)$. At a high-level, CE-V-LEARNING consists of the following steps:

- Line 6-11 (Sample action using mixed-expert FTRL): For each $(h, s)$ we maintain $A_i$ "sub-experts" indexed by $b' \in [A_i]$ (Each sub-expert represents an independent "expert" that runs her own FTRL algorithm). Sub-expert $b'$ first computes an action distribution $q^{b'}(\cdot) \in \Delta_{\mathcal{A}_i}$ via Follow-the-Regularized-Leader (FTRL; Line 8). Then we employ a two-step sampling procedure to obtain the action: First sample a sub-expert $b$ from a suitable distribution $\mu$ computed from $\{q^{b'}\}_{b' \in [A_i]}$, then sample the actual action $a_{h,i}$ from $q^b$.

- Line 13-17 (Take action and record observations): Player $i$ takes action $a_{h,i}$ and observes other player's actions, the reward, and the next state. Sub-expert $b$ then computes a loss estimator and weight according to the observations, which will be used in future FTRL updates.

- Line 19 (Optimistic value update): Updates the optimistic estimate of the value $\overline{V}_{h,i}$ using step-size $\alpha_t$ and bonus $\overline{\beta}_t$.

Finally, after executing Algorithm 1 for $K$ episodes, we use the certified correlated policy procedure (Algorithm 2) to obtain our final output policy $\widehat{\pi}$. This procedure is a direct modification of the certified policy procedure of (Bai et al., 2020) and outputs a correlated policy (because the randomly sampled $k$ and $l$ in line 1 and line 4 of Algorithm 2 are used by all the players) instead of product policy. The same procedure is also used for learning CCEs earlier in Section 3.

Here we specify the hyperparameters used in Algorithm 1:

$$\alpha_t = (H + 1)/(H + t), \quad \eta_t = \sqrt{\iota/(A_i t)}, \quad \overline{\beta}_t = cH^2 A_i \sqrt{\iota/t} + 2cH^2 \iota/t. \tag{2}$$

The constants $\alpha_t^j$ used in Algorithm 2 is defined as

$$\alpha_t^0 := \prod_{k=1}^t (1 - \alpha_k), \quad \alpha_t^j := \alpha_j \prod_{k=j+1}^t (1 - \alpha_k). \tag{3}$$

Note that for any $t \geq 1$, $\{\alpha_t^j\}_{1 \leq j \leq t}$ sums to one and defines a distribution over $[t]$.

### 4.2 OVERVIEW OF TECHNIQUES

Here we briefly overview the techniques used in Algorithm 1.

**Minimizing swap regret via mixed-expert FTRL** The key technical advance in our Algorithm 1 over CCE-V-LEARNING and Nash V-Learning is the use of mixed-expert FTRL (Line 6-11). The purpose of this is to allow the algorithm to achieve low *swap regret* at each $(h, s)$ in a suitable sense—For one-step (normal form) games, it is known that combining low-swap-regret learning for each player leads to an approximate CE (Stoltz, 2005; Cesa-Bianchi & Lugosi, 2006). To integrate this into Markov games, we utilize a celebrated reduction from low-swap-regret learning to usual low-regret learning (Blum & Mansour, 2007), which for any bandit problem with $A_i$ actions maintains $A_i$ *sub-experts* each running its own FTRL algorithm. Our particular application builds upon the *two-step randomization* scheme of Ito (2020) which first samples a sub-expert $b$ and the action from this sub-expert. The distribution $\mu(\cdot)$ for sampling the sub-expert is carefully chosen by solving a linear system (Line 10) so that $\mu$ also coincides with the (marginal) distribution of the sampled action, from which the reduction follows.

**FTRL with predictable weights** Applied naively, the above reduction does not directly work for our purpose, as our analysis requires minimizing the *weighted* swap regret with weights $\alpha_t^i$, whereas the reduction of Ito (2020) relies crucially on the vanilla (average) regret. We address this challenge by using a slightly modified FTRL algorithm for each sub-expert that takes in random but *predictable* weights (i.e. depending fully on prior information and "external" randomness). We present the analysis for such FTRL algorithm in Appendix F.4, and the consequent analysis for the weighted swap regret in Appendix F.1-F.3, both of which may be of independent interest.

**Proposal distributions** Finally, a nuanced but important new design in CE-V-LEARNING is that all sub-experts compute a *proposal* action distribution to sample the sub-expert and the associated action.

---

**Algorithm 1** CE-V-LEARNING for general-sum MGs ($i$-th player's version)

---

**Require:** Hyperparameters: $\{\alpha_t^j\}_{1 \le j \le t \le K}$, $\{\alpha_t\}_{1 \le t \le K}$, $\{\eta_t\}_{1 \le t \le K}$, $\{\bar{\beta}_t\}_{1 \le t \le K}$.

1: **Initialize:** For any $(s, a, h)$, set $\overline{V}_{h,i}(s) \leftarrow H$, $N_h(s) \leftarrow 0$. Set $\mu_h(a|s) \leftarrow 1/A_i$, $q_h^{b'}(a|s) \leftarrow 1/A_i$, $\ell_{h,t}^{b'}(a|s) \leftarrow 0$, $N_h^{b'}(s) \leftarrow 0$ for all $(b', a, h, s, t) \in [A_i] \times [A_i] \times [H] \times \mathcal{S} \times [K]$.

2: **for** episode $k = 1, \ldots, K$ **do**

3:      Receive $s_1$.

4:      **for** step $h = 1, \ldots, H$ **do**

5:          // Compute *proposal* action distributions by FTRL

6:          Update accumulator $t := N_h(s_h) \leftarrow N_h(s_h) + 1$. Set $u_t \leftarrow \alpha_t^t / \alpha_t^1$.

7:          Let $t_{b'} \leftarrow N_h^{b'}(s_h)$ for all $b' \in [A_i]$ for shorthand.

8:          Compute the action distribution for all sub-experts $b' \in [A_i]$:

$$q_h^{b'}(a|s_h) \propto_a \exp\left(-(\eta_{t_{b'}}/u_t) \sum_{\tau=1}^{t_{b'}} w_{h,\tau}(b'|s_h) \ell_{h,\tau}^{b'}(a|s_h)\right).$$

9:          // Sample sub-expert $b$ and action from $q^b(\cdot)$

10:         Compute $\mu_h(\cdot|s_h) \in \Delta_{[A_i]}$ by solving $\mu_h(\cdot|s_h) = \sum_{b'=1}^{A_i} \mu_h(b'|s_h) q_h^{b'}(\cdot|s_h)$.

11:         Sample sub-expert $b \sim \mu_h(\cdot|s_h)$, and then action $a_{h,i} \sim q_h^b(\cdot|s_h)$.

12:         // Take action and feed the observations to sub-expert $b$

13:         Take action $a_{h,i}$, observe the actions $\boldsymbol{a}_{h,-i}$ from all other players.

14:         Observe reward $r_{h,i} = r_{h,i}(s_h, a_{h,i}, \boldsymbol{a}_{h,-i})$ and the next state $s_{h+1}$ from the environment.

15:         Update accumulator for the sampled sub-expert: $t_b := N_h^b(s_h) \leftarrow N_h^b(s_h) + 1$.

16:         Compute and update loss estimator

$$\ell_{h,t_b}^b(a|s_h) \leftarrow \frac{\left[H - h + 1 - (r_{h,i} + \min\{\overline{V}_{h+1,i}(s_{h+1}), H - h\})\right]/H \cdot \mathbf{1}\{a_{h,i} = a\}}{q_h^b(a|s_h) + \eta_{t_b}}.$$

17:         Set $w_{h,t_b}(b|s_h) \leftarrow u_t$.

18:         // Optimistic value update

19:         $\overline{V}_{h,i}(s_h) \leftarrow (1 - \alpha_t)\overline{V}_{h,i}(s_h) + \alpha_t\left(r_{h,i}(s_h, a_{h,i}, \boldsymbol{a}_{h,-i}) + \overline{V}_{h+1,i}(s_{h+1}) + \overline{\beta}_t\right)$.

---

**Algorithm 2** Certified correlated policy $\widehat{\pi}$ for general-sum MGs

---

1: Sample $k \leftarrow \text{Uniform}([K])$.

2: **for** step $h = 1, \ldots, H$ **do**

3:      Observe $s_h$, and set $t \leftarrow N_h^k(s_h)$ (the value of $N_h(s_h)$ at the beginning of the $k$'th episode).

4:      Sample $l \in [t]$ with $\mathbb{P}(l = j) = \alpha_t^j$ (c.f. Eq. (3)).

5:      Update $k \leftarrow k_h^l(s_h)$ (the episode at the end of which the state $s_h$ is observed exactly $l$ times).

6:      Jointly take action $(a_{h,1}, a_{h,2}, \ldots, a_{h,m}) \sim \prod_{i=1}^m \mu_{h,i}^k(\cdot|s_h)$, where $\mu_{h,i}^k(\cdot|s_h)$ is the policy $\mu_{h,i}(\cdot|s_h)$ at the beginning of the $k$'th episode.

---

Then, only the sampled sub-expert takes this action, and all other proposal distributions are discarded. This is different from the original algorithms of (Blum & Mansour, 2007; Ito, 2020) in which the FTRL update come *after* the sub-expert sampling and only happens for the sampled sub-expert. Our design is required here as otherwise the sub-experts are required to predict the next time when it is sampled in order to compute the weighted FTRL update, which is impossible.

### 4.3 THEORETICAL GUARANTEE

We are now ready to present the theoretical guarantee for our CE-V-LEARNING algorithm.

**Theorem 5** (Learning $\varepsilon$-approximate CE for general-sum MGs). *Suppose we run the* CE-V-LEARNING *algorithm (Algorithm 1) for all $m$ players for*

$$K \ge \mathcal{O}\left(\frac{H^6 S \max_{i \in [m]} A_i^2 \iota}{\varepsilon^2} + \frac{H^4 S \max_{i \in [m]} A_i \iota^3}{\varepsilon}\right)$$

episodes ($\iota = \log(m \max_{i \in [m]} A_i HSK/(p\varepsilon))$ is a log factor). Then with probability at least $1 - p$, the certified correlated policy $\widehat{\pi}$ defined in Algorithm 2 is an $\varepsilon$-CE, i.e. $\max_{i \in [m]} \sup_{\phi \in \Phi_i}(V_{1,i}^{\phi \diamond \widehat{\pi}}(s_1) - V_{1,i}^{\widehat{\pi}}(s_1)) \leq \varepsilon$.

**Discussions** To the best of our knowledge, Theorem 5 presents the first result for learning CE that scales polynomially with $\max_{i \in [m]} A_i$, which is significantly better than the best known existing algorithm of Multi-Nash-VI with CE subroutine (Liu et al., 2021) whose sample complexity scales with $\prod_{i \in [m]} A_i$. Similar as in Theorem 2, this follows as our CE-V-LEARNING uses decentralized learning for each player to learn the value function ($V$). We also observe that our sample complexity for learning CE is higher than for learning CCE by a factor of $\widetilde{\mathcal{O}}(H \max_{i \in [m]} A_i)$; the additional $\max_{i \in [m]} A_i$ factor is expected as CE is a strictly harder notion of equilibrium. The proof of Theorem 5 can be found in Appendix C.

## 5 LEARNING NASH EQUILIBRIA IN MARKOV POTENTIAL GAMES

In this section, we consider learning Nash equilibria in Markov Potential Games (MPGs), an important subclass of general-sum MGs. Despite the curse of number of players of learning Nash in general-sum MGs, recent work shows that learning Nash in MPGs does not require sample size exponential in $m$, by using stochastic policy gradient based algorithms (Leonardos et al., 2021; Zhang et al., 2021). In this section, we provide an alternative algorithm NASH-CA that also achieves a mild dependence on $m$ and an improved dependence on $\varepsilon$ by a simple reduction to single-agent learning.

### 5.1 MARKOV POTENTIAL GAMES

We first present the definition of MPGs. Our definition is the finite-horizon variant[2] of the definitions of Macua et al. (2018); Leonardos et al. (2021); Zhang et al. (2021) and is slightly more general as we only require (4) on the total return. Throughout this section, $\pi$ denotes a Markov product policy.

**Definition 6.** *(Markov potential games) A general-sum Markov game is a Markov potential game if there exists a potential function $\Phi$ mapping any product policy to a real number in $[0, \Phi_{\max}]$, such that for any $i \in [m]$, any two policies $\pi_i, \pi_i'$ of the $i^{th}$ player, and any policy $\pi_{-i}$ of other players, the difference of the value functions of the $i^{th}$ player with policies $(\pi_i, \pi_{-i})$ and $(\pi_i', \pi_{-i})$ is equals the difference of the potential function on the same policies, i.e.,*

$$V_{1,i}^{\pi_i, \pi_{-i}}(s_1) - V_{1,i}^{\pi_i', \pi_{-i}}(s_1) = \Phi(\pi_i, \pi_{-i}) - \Phi(\pi_i', \pi_{-i}). \tag{4}$$

Note that the range of the potential function $\Phi_{\max}$ admits a trivial upper bound $\Phi_{\max} \leq mH$ (this can be seen by varying $\pi_i$ for one $i$ at a time). An important example of MPGs is Markov Cooperative Games (MCGs) where all players share the same reward $r_i \equiv r$.

### 5.2 ALGORITHM AND THEORETICAL GUARANTEE

We present a simple algorithm NASH-CA (Nash Coordinate Ascent) for learning an $\varepsilon$-Nash in MPGs. As its name suggests, the algorithm operates by solving single-agent Markov Decision Processes (MDPs) one player at a time, and intrinsically performing coordinate ascent on the potential function of the Markov game. Due to the potential structure of MPGs and the boundedness of the potential function, the local improvements of players across the steps can have an accumulative effect on the potential function, and the algorithm is guaranteed to stop after a bounded number of steps. We give the full description of the NASH-CA in Algorithm 3. We remark that NASH-CA is additionally guaranteed to output a *pure-strategy* Nash equilibrium (cf. Appendix D for definition).

**Theorem 7** (Sample complexity for NASH-CA). *For Markov potential games, with probability at least $1 - p$, Algorithm 3 terminates within $4\Phi_{max}/\varepsilon$ steps of the while loop, and outputs an $\varepsilon$-approximate (pure-strategy) Nash equilibrium. The total episodes of play is at most*

$$K = \mathcal{O}\left(\frac{\Phi_{\max} H^3 S \sum_{i=1}^{m} A_i \iota}{\varepsilon^3} + \frac{\Phi_{\max} H^3 S^2 \sum_{i=1}^{m} A_i \iota^2}{\varepsilon^2}\right),$$

*where $\iota = \log(\frac{mHSK \max_{1 \leq i \leq m} A_i}{\varepsilon p})$ is a log factor.*

---

[2]Our results can easily adapted to the discounted infinite time horizon setup.

---

**Algorithm 3** NASH-CA for Markov potential games

---

**Require:** Error tolerance $\varepsilon$
1: **Initialize:** $\pi = \{\pi_i\}_{i \in [m]}$, where $\pi_i = \{\pi_{h,i}\}_{(h,i) \in [H] \times [m]}$ for some deterministic policy $\pi_{h,i}$.
2: **while** true **do**
3:     Execute policy $\pi$ for $N = \Theta(\frac{H^2 \iota}{\varepsilon^2})$ episodes and obtain $\widehat{V}_{1,i}(\pi)$ which is the empirical average of the total return under policy $\pi$.
4:     **for** player $i = 1, \ldots, m$ **do**
5:         Fix $\pi_{-i}$, and let the $i^{\text{th}}$ player run UCBVI-UPLOW (Algorithm 7) for $K_i = \Theta(\frac{H^3 S A_i \iota}{\varepsilon^2} + \frac{H^3 S^2 A_i \iota^2}{\varepsilon})$ episodes and get a new deterministic policy $\widehat{\pi}_i$.
6:         Execute policy $(\widehat{\pi}_i, \pi_{-i})$ for $N = \Theta(\frac{H^2 \iota}{\varepsilon^2})$ episodes and obtain $\widehat{V}_{1,i}(\widehat{\pi}_i, \pi_{-i})$ which is the empirical average of the total return under policy $(\widehat{\pi}_i, \pi_{-i})$.
7:         Set $\Delta_i \leftarrow \widehat{V}_{1,i}(\widehat{\pi}_i, \pi_{-i}) - \widehat{V}_{1,i}(\pi)$.
8:     **if** $\max_{i \in [m]} \Delta_i > \varepsilon/2$ **then**
9:         Update $\pi_j \leftarrow \widehat{\pi}_j$ where $j = \arg\max_{i \in [m]} \Delta_i$.
10:     **else**
11:         **return** $\pi$

---

**Discussions** For small enough $\varepsilon$, the sample complexity for the NASH-CA algorithm in the above theorem is $\widetilde{\mathcal{O}}(\Phi_{\max} H^3 S \sum_{i \le m} A_i / \varepsilon^3)$. As $\Phi_{\max} \le mH$, this at most scales with the number of players as $m \sum_{i \le m} A_i$, which is much better than the exponential in $m$ sample complexity for general-sum MGs without additional structures. Compared with recent results on learning Nash via policy gradients (Leonardos et al., 2021; Zhang et al., 2021), the NASH-CA algorithm also achieves $\text{poly}(m, \max_{i \le m} A_i)$ dependence, and significantly improves on the $\varepsilon$ dependence from their $\varepsilon^{-6}$ to $\varepsilon^{-3}$. In addition, our algorithm does not require assumptions on bounded distribution mismatch coefficient as they do, due to the exploration nature of our single-agent MDP subroutine.

Also, compared with the sample complexity bound $\widetilde{\mathcal{O}}(H^4 S^2 \prod_{i=1}^m A_i / \varepsilon^2)$ of the Nash-VI algorithm (Liu et al., 2021) for general-sum MGs (not restricted to MPGs), our NASH-CA algorithm doesn't suffer from the exponential dependence on $m$ thanks to the MPG structure. We do achieve a looser in the dependence on $\varepsilon$, yet overall our sample complexity is still better unless $\varepsilon < (\sum_{i=1}^m A_i)/(\prod_{i=1}^m A_i)$ is exponentially small. The proof of Theorem 7 can be found in Appendix D.

**A lower bound** To accompany Theorem 7, we establish a sample complexity lower bound of $\Omega(H^2 \sum_{i=1}^m A_i / \varepsilon^2)$ for learning pure-strategy Nash in MCGs and hence MPGs (Theorem E.1 in Appendix E). This lower bound improves in the $A_i$ dependence over the naive reduction to single-player MDPs (Domingues et al., 2021), which gives $\Omega(H^3 S \max_{i \in [m]} A_i / \varepsilon^2)$, though is loose on the $S, H$ dependence. The improved $A_i$ dependence is achieved by constructing a novel class of hard instances of on one-step games (Lemma E.2), which may be of further technical interest. However, there is still a large gap between these lower bounds and the best current upper bound of either our $\widetilde{\mathcal{O}}(\sum_{i=1}^m A_i / \varepsilon^3)$ or the $\widetilde{\mathcal{O}}(\prod_{i=1}^m A_i / \varepsilon^2)$ of Liu et al. (2021), which we leave as future work.

## 6 CONCLUSION

This paper investigates the question of when can we solve general-sum Markov games (MGs) sample-efficiently with a mild dependence on the number of players. Our results show that this is possible for learning approximate (Coarse) Correlated Equilibria in general-sum MGs, as well as learning approximate Nash equilibrium in Markov potential games. In both cases, our sample complexity bounds improve over existing results in many aspects. Our work opens up many interesting directions for future work, such as sharper algorithms for both problems, sample complexity lower bounds, or how to perform sample-efficient learning in general-sum MGs with function approximations. In addition to Markov potential games, it would also be interesting to explore alternative structural assumptions that permit sample-efficient learning.

ACKNOWLEDGEMENT

Ziang Song is partially supported by the elite undergraduate training program of School of Mathematical Sciences in Peking University.

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

# A  EXPONENTIAL IN $m$ LOWER BOUND FOR LEARNING NASH IN GENERAL-SUM MGS

In this section, we give a sample complexity lower bound for computing approximate Nash equilibrium in one-step binary-action general-sum MGs ($H = 1$, $S = 1$ and $A_i = 2$) which has an exponential dependence in $m$, the number of players. The result is built on the lower bound of query complexity in Rubinstein (2016).

We use $\mathcal{G}$ to denote the one-step Markov game ($H = 1$ and $S = 1$), in which there are $m$ players and $A = 2$ actions for each player. We index the players by $[m] = \{1, \ldots, m\}$ and denote the actions space of each player by $[A] = \{1, 2\}$. Since we restricted attention to binary-action games (i.e. $A = 2$), the total number of joint actions is $2^m$.

We define a (exact) *query* as the procedure where the algorithm queries a joint action $\boldsymbol{a} \in [A]^m$ and observes the (deterministic) reward $r_i(\boldsymbol{a}) \in [0, 1]$. We define the *query complexity* (Chen et al., 2015) for learning $\varepsilon$-approximate Nash equilibrium (ANE) as the following.

**Definition A.1** (Query complexity). *The query complexity $QC_p(\mathrm{ANE}(m, \varepsilon))$ for learning $\varepsilon$-ANE is defined as the smallest $n$ such that there exists a randomized oracle algorithm $\mathcal{A}$ satisfying the following: for any binary-action, $m$-player game $\mathcal{G}$, the algorithm $\mathcal{A}$ can use no more than $n$ sequential queries of the reward to output an $\varepsilon$-ANE with probability at least $1 - p$.*

In one-step MGs with deterministic reward, the query complexity is equivalent to the sample complexity, since each query obtains a reward entry. The following result in Rubinstein (2016) gives a $2^{\Omega(m)}$ query complexity lower bound for learning $\varepsilon_0$-ANE in $m$-player binary action games.

---

**Algorithm 4** CCE-V-Learning for General-sum MGs ($i$-th player's version)

---

**Require:** Hyperparameters: $\{\alpha_t\}_{1 \leq t \leq K}$, $\{\eta_t\}_{1 \leq t \leq K}$, $\{\bar{\beta}_t\}_{1 \leq t \leq K}$.

1: **Initialize:** For any $(s, a, h)$, set $\overline{V}_{h,i}(s) \leftarrow H$, $\underline{V}_{h,i}(s) \leftarrow 0$, $\overline{L}_{h,i} \leftarrow 0$, $\mu_h(a|s) \leftarrow 1/A_i$, $N_h(s) \leftarrow 0$.

2: **for** episode $k = 1, \ldots, K$ **do**

3:     Receive $s_1$.

4:     **for** step $h = 1, \ldots, H$ **do**

5:         Take action $a_h \sim \mu_h(\cdot|s_h)$, observe the action $\boldsymbol{a}_{h,-i}$ from the other players.

6:         Observe reward $r_{h,i} = r_{h,i}(s_h, a_h, \boldsymbol{a}_{h,-i})$ and next state $s_{h+1}$ from the environment.

7:         Update accumulators: $t := N_h(s_h) \leftarrow N_h(s_h) + 1$.

8:         $\overline{V}_{h,i}(s_h) \leftarrow (1 - \alpha_t)\overline{V}_{h,i}(s_h) + \alpha_t \left( r_{h,i}(s_h, a_h, \boldsymbol{a}_{h,-i}) + \overline{V}_{h+1,i}(s_{h+1}) + \overline{\beta}_t \right)$

9:         $\underline{V}_{h,i}(s_h) \leftarrow (1 - \alpha_t)\underline{V}_{h,i}(s_h) + \alpha_t \left( r_{h,i}(s_h, a_h, \boldsymbol{a}_{h,-i}) + \underline{V}_{h+1,i}(s_{h+1}) - \overline{\beta}_t \right)$

10:         **for** all $a \in \mathcal{A}_i$ **do**

11:           $\bar{\ell}_{h,i}(s_h, a) \leftarrow \frac{1}{H}[H - h + 1 - r_{h,i} - \min\{\overline{V}_{h+1,i}(s_{h+1}), H - h\}]\mathbf{1}\{a_h = a\}/[\mu_h(a_h|s_h) + \eta_t]$.

12:         $\overline{L}_{h,i}(s_h, a) \leftarrow (1 - \alpha_t)\overline{L}_{h,i}(s_h, a) + \alpha_t\overline{L}_{h,i}(s_h, a)$

13:         Set $\mu_h(\cdot|s_h) \propto \exp[-(\eta_t/\alpha_t)\overline{L}_{h,i}(s_h, \cdot)]$.

---

**Proposition A.2** (Corollary 4.5, (Rubinstein, 2016)). *There exists absolute constants $\varepsilon_0 > 0$ and $c > 0$, such that for all $m$,*

$$QC_p(\text{ANE}(m, \varepsilon_0)) = 2^{\Omega(m)}, \text{ where } p = 2^{-cm}.$$

This result shows that it is impossible for any algorithm to learn an $\varepsilon_0$-ANE for every binary action game with probability at least $(1 - p)$ using $\text{poly}(m, \log(1/p))$ samples: such an algorithm with $p = 2^{-cm}$ would only use $\text{poly}(m, \log(2^{cm})) = \text{poly}(m)$ samples, yet the sample complexity lower bound in Proposition A.2 requires at least $2^{\Omega(m)} = \exp(\Omega(m))$ samples. Since Proposition A.2 allows $\varepsilon_0 = \Theta(1)$, this also rules out the possibility of learning $\varepsilon$-ANE with $\text{poly}(m, \log(1/p), 1/\varepsilon)$ samples for all small $\varepsilon$.

## B   Proofs for Section 3

### B.1   Algorithm for learning CCE in general-sum Markov games

Our algorithm used to learn CCE in general-sum MGs is a combination of Algorithm 4 and Algorithm 2. In particular, Algorithm 4 computes a set of policies and plays these policies in each episode. Algorithm 2 used the full history in Algorithm 4 to produce a certified, general correlated policy which we will show to be a CCE (we will also use the same Algorithm 2 to produce the certified policy in the algorithm of learning CE). During the execution of Algorithm 2, if the index $t$ is 0 at some step $h$, the certified policy can choose any action at and after step $h$.

In Algorithm 4, we choose the hyper-parameters as follows:

$$\alpha_t = \frac{H + 1}{H + t}, \quad \eta_t = \sqrt{\frac{H\iota}{A_i t}}, \quad \overline{\beta}_t = c\sqrt{\frac{H^3 A_i \iota}{t}} + 2c\frac{H^2 \iota}{t},$$

where $c > 0$ is some absolute constant, and $\iota = \log(\frac{m \max_{i \in [m]} A_i HSK}{p\varepsilon})$ is a log factor. The choice of $\eta_t$ follows the V-OL algorithm in Tian et al. (2021) which helps to shave off an $H$ factor in the sample complexity compared with the original Nash V-Learning algorithm in Bai et al. (2020).

Here, we have a short comment on the log factor $\iota$. In fact, we need $\iota$ to be $C\log(\frac{m \max_{i \in [m]} A_i HSK}{p\varepsilon})$ for some absolute constant $C$. For the cleanness of the results, in this paper, we ignore this difference since this would not harm the correctness of all the results we present.

### B.2   Proof of Theorem 2

We begin with an auxiliary lemma on $\alpha_t^j$ (its definition is in (3)).

**Lemma B.1** (Lemma 4.1 in Jin et al. (2018)). *The following properties hold for $\alpha_t^j$:*

*1.* $\frac{1}{\sqrt{t}} \leq \sum_{j=1}^t \frac{\alpha_t^j}{\sqrt{j}} \leq \frac{2}{\sqrt{t}}$ *for every $t \geq 1$.*

*2.* $\max_{j \in [t]} \alpha_t^j \leq \frac{2H}{t}$ *and* $\sum_{j=1}^t \left(\alpha_t^j\right)^2 \leq \frac{2H}{t}$ *for every $t \geq 1$.*

*3.* $\sum_{t=j}^\infty \alpha_t^j = 1 + \frac{1}{H}$ *for every $j \geq 1$.*

*4.* $\sum_{j=1}^t \frac{\alpha_t^j}{j} \geq \frac{1}{2t}$ *for every $t \geq 1$.*

Property 4 above does not appear in (Jin et al., 2018), for which we provide a quick proof here:

$$\sum_{j=1}^t \frac{\alpha_t^j}{j} \geq \sum_{j=[t/2]}^t \frac{\alpha_t^j}{j} \geq 1/t \sum_{j=[t/2]}^t \alpha_t^j \overset{(i)}{\geq} 1/(2t) \sum_{j=1}^t \alpha_t^j = 1/(2t).$$

Here, (i) uses $\alpha_t^j$ is increasing in $j$ for fixed $t$. $\qquad\square$

**Some notations** The following notations will be used repeatedly (throughout this section and the next section). At the beginning of any episode $k$ for a particular state $s_h$, we denote $k_h^1(s_h) < \cdots < k_h^{N_h^k(s_h)}(s_h) < k$ to be all the episodes that the state $s_h$ was visited, where $N_h^k(s_h)$ is the times the state $s_h$ has been visited before the start of $k$-th episode. When there is no confusion, we sometimes will write $k^j = k_h^j(s_h)$ in short. For player $i$, we let $\underline{V}_{h,i}^k, \overline{V}_{h,i}^k, \mu_i^k$ denote the values and policies maintained by Algorithm 4 at the beginning of $k$-th episode, and $\boldsymbol{a}_h^k$ denote taken action at step $h$ and episode $k$. For any joint policy $\mu_h$ (over all players), reward function $r$ and value function $V$, we define the operators $\mathbb{P}_h$ and $\mathbb{D}_{\mu_h}$ as

$$[\mathbb{P}_h V](s, \boldsymbol{a}) := \mathbb{E}_{s' \sim \mathbb{P}_h(\cdot|s,\boldsymbol{a})} V(s'), \tag{5}$$

$$\mathbb{D}_{\mu_h}[r + \mathbb{P}_h V](s) := \mathbb{E}_{\boldsymbol{a}_h \sim \mu_h}[r(s, \boldsymbol{a}_h) + \mathbb{E}_{s_{h+1} \sim \mathbb{P}_h(\cdot|s,\boldsymbol{a}_h)} V(s_{h+1})]. \tag{6}$$

Towards proving Theorem 2, we begin with a simple consequence of the update rule in Algorithm 4, which will be used several times later.

**Lemma B.2** (Update rule). *Fix a state $s$ in time step $h$ and fix an episode $k$, let $t = N_h^k(s)$ and suppose $s$ was previously visited at episodes $k^1 < \cdots < k^t < k$ at the $h$-th step. The update rules in Algorithm 4 gives the following equations:*

$$\overline{V}_{h,i}^k(s) = \alpha_t^0 H + \sum_{j=1}^t \alpha_t^j \left[ r_{h,i}\left(s, a_h^{k^j}, \boldsymbol{a}_{h,-i}^{k^j}\right) + \overline{V}_{h+1,i}^{k^j}\left(s_{h+1}^{k^j}\right) + \overline{\beta}_j \right],$$

$$\underline{V}_{h,i}^k(s) = \sum_{j=1}^t \alpha_t^j \left[ r_{h,i}\left(s, a_h^{k^j}, \boldsymbol{a}_{h,-i}^{k^j}\right) + \underline{V}_{h+1,i}^{k^j}\left(s_{h+1}^{k^j}\right) - \overline{\beta}_j \right].$$

We next present and prove the following lemma which helps to explain why our choice of the bonus term is $\overline{\beta}_t$. The constant $c$ in $\overline{\beta}_t$ is actually the same with the constant $c$ in this lemma.

**Lemma B.3** (Per-state guarantee). *Fix a state $s$ in time step $h$ and fix an episode $k$, let $t = N_h^k(s)$ and suppose $s$ was previously visited at episodes $k^1 < \cdots < k^t < k$ at the $h$-th step. With probability at least $1 - \frac{p}{2}$, for any $(i, s, h, t) \in [m] \times \mathcal{S} \times [H] \times [K]$, there exist a constant $c$ s.t.*

$$\max_{\mu \in \Delta_{\mathcal{A}_i}} \sum_{j=1}^t \alpha_t^j \mathbb{D}_{\mu \times \mu_{h,-i}^{k^j}} \left( r_{h,i} + \mathbb{P}_h \min\{\overline{V}_{h+1,i}^{k^j}, H - h\} \right)(s)$$

$$- \sum_{j=1}^t \alpha_t^j \left[ r_{h,i}\left(s, a_h^{k^j}, \boldsymbol{a}_{h,-i}^{k^j}\right) + \min\{\overline{V}_{h+1,i}^{k^j}(s_{h+1}^{k^j}), H - h\} \right] \leq c\sqrt{H^3 A_i \iota/t} + cH^2 \iota/t.$$

**Proof of Lemma B.3**     First, we decompose

$$\max_{\mu} \sum_{j=1}^{t} \alpha_t^j \mathbb{D}_{\mu \times \mu_{h,-i}^{k^j}} \left( r_{h,i} + \mathbb{P}_h \min\{\overline{V}_{h+1,i}^{k^j}, H - h\} \right)(s)$$

$$- \sum_{j=1}^{t} \alpha_t^j \left[ r_{h,i} \left( s, a_h^{k^j}, \boldsymbol{a}_{h,-i}^{k^j} \right) + \min\{\overline{V}_{h+1,i}^{k^j}(s_{h+1}^{k^j}), H - h\} \right]$$

into $R^\star(i, s, h, t) + U(i, s, h, t)$ where

$$R^\star(i, s, h, t) := \max_{\mu} \sum_{j=1}^{t} \alpha_t^j \mathbb{D}_{\mu \times \mu_{h,-i}^{k^j}} \left( r_{h,i} + \mathbb{P}_h \min\{\overline{V}_{h+1,i}^{k^j}, H - h\} \right)(s)$$

$$- \sum_{j=1}^{t} \alpha_t^j \mathbb{D}_{\mu_{h,i}^{k^j} \times \mu_{h,-i}^{k^j}} \left( r_{h,i} + \mathbb{P}_h \min\{\overline{V}_{h+1,i}^{k^j}, H - h\} \right)(s),$$

and

$$U(i, s, h, t) := \sum_{j=1}^{t} \alpha_t^j \mathbb{D}_{\mu_{h,i}^{k^j} \times \mu_{h,-i}^{k^j}} \left( r_{h,i} + \mathbb{P}_h \min\{\overline{V}_{h+1,i}^{k^j}, H - h\} \right)(s)$$

$$- \sum_{j=1}^{t} \alpha_t^j \left[ r_{h,i} \left( s, a_h^{k^j}, \boldsymbol{a}_{h,-i}^{k^j} \right) + \min\{\overline{V}_{h+1,i}^{k^j}(s_{h+1}^{k^j}), H - h\} \right].$$

We first bound $U(i, s, h, t)$. Define $\mathcal{F}_l$ as the $\sigma$-algebra generated by all the random variables up to the time when $s_h$ is observed at the $l$-th episode. Recall that for $j \geq 1$, $k^j = k_h^j(s) = \inf\{l > k^{j-1} : s \text{ is visited at step } h \text{ in episode } l\}$ (with convention $k^0 = 0$). Then $\{k^j\}_{j \geq 1}$ is a sequence of increasing stopping times w.r.t. $\{\mathcal{F}_l\}_{l \geq 1}$. Define $\mathcal{G}_j = \mathcal{F}_{k^{j+1}}$. So $\{\mathcal{G}_j\}_{j \geq 0}$ is also a filtration. Under $\mathcal{G}_{j-1} (= \mathcal{F}_{k^j})$, by the definition of operator $\mathbb{D}$ and $\mathbb{P}$, we have

$$\mathbb{E}\left[ r_{h,i} \left( s, a_h^{k^j}, \boldsymbol{a}_{h,-i}^{k^j} \right) + \min\{\overline{V}_{h+1,i}^{k^j}(s_{h+1}^{k^j}), H - h\} \bigg| \mathcal{G}_{j-1} \right]$$

$$= \mathbb{D}_{\mu_{h,i}^{k^j} \times \mu_{h,-i}^{k^j}} \left( r_{h,i} + \mathbb{P}_h \min\{\overline{V}_{h+1,i}^{k^j}, H - h\} \right)(s).$$

So we can apply Azuma-Hoeffding inequality. Note that $\sum_{j=1}^{t} (\alpha_t^j)^2 \leq 2H/t$ by Lemma B.1. Using Azuma-Hoeffding inequality, we have with probability at least $1 - \frac{p}{4mHSK}$

$$U(i, s, h, t) = \sum_{j=1}^{t} \alpha_t^j \mathbb{D}_{\mu_{h,i}^{k^j} \times \mu_{h,-i}^{k^j}} \left( r_{h,i} + \mathbb{P}_h \min\{\overline{V}_{h+1,i}^{k^j}, H - h\} \right)(s)$$

$$- \sum_{j=1}^{t} \alpha_t^j \left[ r_{h,i} \left( s, a_h^{k^j}, \boldsymbol{a}_{h,-i}^{k^j} \right) + \min\{\overline{V}_{h+1,i}^{k^j}(s_{h+1}^{k^j}), H - h\} \right]$$

$$\leq \sqrt{2H^2 \log(4mHSK/p) \sum_{j=1}^{t} (\alpha_t^j)^2} \leq 2\sqrt{H^3 \iota / t}.$$

After taking a union bound, we have the following statement is true with probability at least $1 - p/4$,

$$U(i, s, h, t) \leq 2\sqrt{H^3 \iota / t} \quad \text{for all } (i, s, h, t) \in [m] \times \mathcal{S} \times [H] \times [K].$$

Then we bound $R^\star(i, s, h, t)$. For fixed $(i, s, h)$, if we define the loss function

$$\ell_j(a) = \frac{1}{H} \mathbb{E}_{a_i = a, \boldsymbol{a}_{-i} \sim \mu_{h,-i}^{k^j}} [H - h + 1 - r_{h,i}(s, \boldsymbol{a}) - \mathbb{P}_h \min\{\overline{V}_{h+1,i}^{k^j}, H - h\}(s)] \in [0, 1],$$

---

**Algorithm 5** Correlated policy $\widehat{\pi}_h^k$ for general-sum Markov games

1: **for** step $h' = h, \ldots, H$ **do**
2:     Observe $s_{h'}$, and set $t \leftarrow N_{h'}^k(s_{h'})$.
3:     Sample $l \in [t]$ with $\mathbb{P}(l = j) = \alpha_t^j$.
4:     Update $k \leftarrow k_t^l(s_{h'})$.
5:     Jointly take action $(a_{h',1}, a_{h',2}, \ldots, a_{h',m}) \sim \prod_{i=1}^m \mu_{h',i}^k(\cdot|s_{h'})$.

---

then $R^\star(i, s, h, t) = H \max_\mu \sum_{j=1}^t \alpha_t^j \left\langle \mu - \mu_{h,i}^{k^j}, \ell_j \right\rangle$ becomes the weighted regret with weight $\alpha_t^j$. Note that the update rule for $\mu_{h,i}^{k^j}(\cdot|s)$ is essentially performing Follow-the-Regularized-Leader (FTRL) algorithm with changing step size for each state $s$ and each step $h$ to solve an adversarial bandit problem. Lemma 17 in (Bai et al., 2020)[3] bounds the weight regret with high probability. Using that lemma, we have with probability at least $1 - \frac{p}{4mHS}$,

$$R^\star(i, s, h, t) \leq \frac{H \alpha_t^t \log A_i}{\eta_t} + \frac{3HA_i}{2} \sum_{j=1}^t \eta_j \alpha_t^j + H \sqrt{2\iota \sum_{j=1}^t (\alpha_t^j)^2} + \frac{H}{2} \max_{j \leq t} \alpha_t^j \iota + H \max_{j \leq t} \alpha_t^j \iota / \eta_t$$

simultaneously for all $t \in [K]$. By Lemma B.1 and $\eta_t = \sqrt{\frac{H\iota}{A_i t}}$, we have with probability at least $1 - \frac{p}{4mHS}$

$$R^\star(i, s, h, t) \leq 2\sqrt{H^3 A_i \iota / t} + 3\sqrt{H^3 A_i \iota / t} + 2\sqrt{H^3 \iota / t} + H^2 \iota / t + 2\sqrt{H^3 A_i \iota / t}$$
$$\leq 10\sqrt{H^3 A_i \iota / t} + 10 H^2 \iota / t \quad \text{for all } t \in [K].$$

Again, taking a union bound in all $(i, s, h) \in [m] \times \mathcal{S} \times [H]$, we have with probability at least $1 - p/4$,

$$R^\star(i, s, h, t) \leq 10\sqrt{H^3 A_i \iota / t} + 10 H^2 \iota / t \quad \text{for all } (i, s, h, t) \in [m] \times \mathcal{S} \times [H] \times [K].$$

Finally, we concluded that with probability at least $1 - p/2$, we have

$$U(i, s, h, t) + R^\star(i, s, h, t) \leq c\sqrt{H^3 A_i \iota / t} + c H^2 \iota / t \quad \text{for all } (i, s, h, t) \in [m] \times \mathcal{S} \times [H] \times [K]$$

for some absolute constant $c$. $\qquad\square$

Recall that the certified policy $\widehat{\pi}$ as in Algorithm 2 is a nested mixture of policies. We further define policies $\{\widehat{\pi}_h^k\}_{h \in [H], k \in [K]}$ in Algorithm 5. By construction, the relationship between $\widehat{\pi}$ and $\widehat{\pi}_h^k$ is that when players jointly play policy the $\widehat{\pi}$, they first sample $k$ from Uniform$([K])$, then they play together the policy $\widehat{\pi}_1^k$ (Algorithm 5 for $h = 1$) with the same sampled $k$. As a result, we have the following relationship:

$$V_{1,i}^{\widehat{\pi}}(s_1) = \frac{1}{K} \sum_{k=1}^K V_{1,i}^{\widehat{\pi}_1^k}(s_1). \tag{7}$$

**Definition B.4** (Policy starting from the $h$-th step). *We define the policy starting from the $h$-th step for player $i$ as $\pi_{\geq h,i} := \{\pi_{h',i} : \Omega \times (\mathcal{S} \times \mathcal{A})^{h'-h} \times \mathcal{S} \rightarrow \Delta_{\mathcal{A}_i}\}_{h'=h}^H$. At each step $h' \geq h$, $\pi_{\geq h,i}$ samples action based on current state, the history starting from the $h$-th step and a random number $\omega \in \Omega$. We use $\Pi_{\geq h,i}$ to denote all policies for player $i$ starting from the $h$-th step. Similar to Section 2, we can define general correlated policy starting from the $h$-th step (where the random numbers may be correlated for different players), and we use $\Pi_{\geq h}$ to denote all such general correlated policy starting from the $h$-th step.*

---

[3] A very similar result is Lemma F.3 in our paper. However, here we need Lemma 17 in (Bai et al., 2020) to get the optimal $H$ dependency.

For $\pi \in \Pi_{\geq h}$, we can define the value function starting from the $h$-th step as:

$$V_{h,i}^\pi(s) := \mathbb{E}_\pi\left[\sum_{h'=h}^H r_{h',i}|s_h = s\right]. \tag{8}$$

We also define the value function of the best response as:

$$V_{h,i}^{\dagger,\pi_{h,-i}}(s) := \max_{\mu_i \in \Pi_{\geq h,i}} V_{h,i}^{\mu_i \times \pi_{h,-i}}(s).$$

One example of a policy starting from the $h$-th step is $\widehat{\pi}_h^k$ defined in Algorithm 5, so that we can define $V_{h,i}^{\widehat{\pi}_h^k}(s)$ and $V_{h,i}^{\dagger,\widehat{\pi}_{h,-i}^k}(s)$.

**Lemma B.5** (Valid upper and lower bounds). *We have*

$$\overline{V}_{h,i}^k(s) \geq V_{h,i}^{\dagger,\widehat{\pi}_{h,-i}^k}(s), \quad \underline{V}_{h,i}^k(s) \leq V_{h,i}^{\widehat{\pi}_h^k}(s)$$

*for all $(i,s,h,k) \in [m] \times \mathcal{S} \times [H] \times [K]$ with probability at least $1-p$.*

**Proof of Lemma B.5**    We prove this lemma by backward induction over $h \in [H+1]$. The base case of $h = H+1$ is true as all the value functions equal 0 by definition. Suppose the claim is true for $h+1$. We begin with upper bounding $V_{h,i}^{\dagger,\widehat{\pi}_{h,-i}^k}(s)$. Let $t = N_h^k(s)$ and $k^j = k_h^j(s)$ for $1 \leq j \leq t$ to be the $j$'th time that $s$ is previously visited. By the definition of certified policies $\widehat{\pi}_{h,i}^k$ and by the value iteration formula of MGs, we have for any policy $\mu_i \in \Pi_{\geq h,i}$,

$$V_{h,i}^{\mu_i,\widehat{\pi}_{h,-i}^k}(s) = \sum_{j=1}^t \alpha_t^j \mathbb{E}_{\boldsymbol{a}\sim\mu_{h,i}\times\mu_{h,-i}^{k^j}}\left[r_{h,i}(s,\boldsymbol{a}) + \mathbb{E}_{s'\sim\mathbb{P}_h(\cdot|s,\boldsymbol{a})}V_{h+1,i}^{(\mu_{h+1:H,i}|s,\boldsymbol{a}),\widehat{\pi}_{h+1,-i}^{k^j}}(s')\right]$$

$$\leq \sum_{j=1}^t \alpha_t^j \mathbb{E}_{\boldsymbol{a}\sim\mu_{h,i}\times\mu_{h,-i}^{k^j}}\left[r_{h,i}(s,\boldsymbol{a}) + \mathbb{E}_{s'\sim\mathbb{P}_h(\cdot|s,\boldsymbol{a})}V_{h+1,i}^{\dagger,\widehat{\pi}_{h+1,-i}^{k^j}}(s')\right].$$

Here, $\mu_{h,i}$ is the policy $\mu_i$ at the $h$-th step, and $(\mu_{h+1:H,i}|s,\boldsymbol{a})$ is the policy $\mu_i$ from time $h+1$ to $H$ with history information at $h$-th step to be $s_h = s$ and $\boldsymbol{a}_h = \boldsymbol{a}$. By the definition of $\Pi_{\geq h,i}$, we have $(\mu_{h+1:H,i}|s,\boldsymbol{a}) \in \Pi_{\geq h+1,i}$ which implies the inequality in the equation above. By taking supremum over $\mu_i \in \Pi_{\geq h,i}$ and using the definition of the operator $\mathbb{D}$ in (6), we have

$$V_{h,i}^{\dagger,\widehat{\pi}_{h,-i}^k}(s) \leq \sup_{\mu_i\in\Delta_{\mathcal{A}_i}} \sum_{j=1}^t \alpha_t^j \mathbb{D}_{\mu_i\times\mu_{h,-i}^{k^j}}[r_{h,i} + \mathbb{P}_h V_{h+1,i}^{\dagger,\widehat{\pi}_{h+1,-i}^{k^j}}](s).$$

Conditional on the high probability event in Lemma B.3, we use the inductive hypothesis to obtain

$$V_{h,i}^{\dagger,\widehat{\pi}_{h,-i}^k}(s) \leq \sup_{\mu_i} \sum_{j=1}^t \alpha_t^j \mathbb{D}_{\mu_i\times\mu_{h,-i}^{k^j}}[r_{h,i} + \mathbb{P}_h \min\{\overline{V}_{h+1,i}^{k^j}, H-h\}](s)$$

$$\leq \sum_{j=1}^t \alpha_t^j\left[r_{h,i}\left(s,a_h^{k^j},\boldsymbol{a}_{h,-i}^{k^j}\right) + \min\{\overline{V}_{h+1,i}^{k^j}(s_{h+1}^{k^j}), H-h\}\right] + c\sqrt{H^3 A_i\iota/t} + cH^2\iota/t$$

$$\overset{(i)}{\leq} \sum_{j=1}^t \alpha_t^j\left[r_{h,i}\left(s,a_h^{k^j},\boldsymbol{a}_{h,-i}^{k^j}\right) + \min\{\overline{V}_{h+1,i}^{k^j}(s_{h+1}^{k^j}), H-h\} + \overline{\beta}_j\right]$$

$$\leq \sum_{j=1}^t \alpha_t^j\left[r_{h,i}\left(s,a_h^{k^j},\boldsymbol{a}_{h,-i}^{k^j}\right) + \overline{V}_{h+1,i}^{k^j}\left(s_{h+1}^{k^j}\right) + \overline{\beta}_j\right]$$

$$= \overline{V}_{h,i}^k(s).$$

Here, (i) uses our choice of $\overline{\beta}_j = c\sqrt{\frac{H^3 A_i\iota}{j}} + 2c\frac{H^2\iota}{j}$ and $\frac{1}{\sqrt{t}} \leq \sum_{j=1}^t \frac{\alpha_t^j}{\sqrt{j}}$, $\frac{1}{t} \leq \sum_{j=1}^t \frac{2\alpha_t^j}{j}$.

Meanwhile, for $\underline{V}_{h,i}^k(s)$, by the definition of certified policy and inductive hypothesis,

$$V_{h,i}^{\widehat{\pi}_h^k}(s) = \sum_{j=1}^t \alpha_t^j \mathbb{D}_{\mu_h^{k^j}}[r_{h,i} + \mathbb{P}_h V_{h+1,i}^{\widehat{\pi}^{k^j}}](s)$$

$$\geq \sum_{j=1}^t \alpha_t^j \mathbb{D}_{\mu_h^{k^j}}[r_{h,i} + \mathbb{P}_h \max\{\underline{V}_{h+1,i}^{k^j}, 0\}](s).$$

Then we note that $\left\{ \mathbb{D}_{\mu_h^{k^j}}[r_{h,i} + \mathbb{P}_h \max\{\underline{V}_{h+1,i}^{k^j}, 0\}](s) - \left[ r_{h,i}\left(s, a_h^{k_h^j}, \boldsymbol{a}_{h,-i}^{k_h^j}\right) + \max\{\underline{V}_{h+1,i}^{k^j}(s_{h+1}^{k_h^j}), 0\}\right]\right\}_{j \geq 1}$ is a martingale difference w.r.t. the filtration $\{\mathcal{G}_j\}_{j \geq 0}$, which is defined in the proof of Lemma B.3. So by Azuma-Hoeffding inequality, with probability at least $1 - \frac{p}{2mSKH}$

$$\sum_{j=1}^t \alpha_t^j \mathbb{D}_{\mu_h^{k^j}}[r_{h,i} + \mathbb{P}_h \max\{\underline{V}_{h+1,i}^{k^j}, 0\}](s)$$

$$\geq \sum_{j=1}^t \alpha_t^j \left[ r_{h,i}\left(s, a_h^{k_h^j}, \boldsymbol{a}_{h,-i}^{k_h^j}\right) + \max\{\underline{V}_{h+1,i}^{k^j}(s_{h+1}^{k_h^j}), 0\}\right] - 2\sqrt{\frac{H^3 \iota}{t}}. \tag{9}$$

On this event, we have

$$\sum_{j=1}^t \alpha_t^j \mathbb{D}_{\mu_h^{k^j}}[r_{h,i} + \mathbb{P}_h \max\left\{\underline{V}_{h+1,i}^{k^j}, 0\right\}](s)$$

$$\overset{(i)}{\geq} \sum_{j=1}^t \alpha_t^j \left[ r_{h,i}\left(s, a_h^{k_h^j}, \boldsymbol{a}_{h,-i}^{k_h^j}\right) + \max\left\{\underline{V}_{h+1,i}^{k^j}(s_{h+1}^{k_h^j}), 0\right\} - \overline{\beta}_j\right]$$

$$= \underline{V}_{h,i}^k(s).$$

Here, (i) uses $\sum_{j=1}^t \alpha_t^j \beta_j \geq 2\sum_{j=1}^t \alpha_t^j \sqrt{H^3 \iota / j} \geq 2\sqrt{H^3 \iota / t}$.

As a result, the backward induction would work well for all $h$ as long as the inequalities in Lemma B.3 and (9) hold for all $(i, s, h, k) \in [m] \times \mathcal{S} \times [H] \times [K]$. Taking a union bound in all $(i, s, h, k) \in [m] \times \mathcal{S} \times [H] \times [K]$, we have with probability at least $1 - p/2$, the inequality in (9) is true simultaneously for all $(i, s, h, k) \in [m] \times \mathcal{S} \times [H] \times [K]$. Therefore the inequalities in Lemma B.3 and (9) hold simultaneously for all $(i, s, h, k) \in [m] \times \mathcal{S} \times [H] \times [K]$ with probability at least $1 - p$. This finishes the proof of this lemma. □

Equipped with these lemmas, we are ready to prove Theorem 2.

**Proof of Theorem 2** Conditional on the high probability event in Lemma B.5 (this happens with probability at least $1 - p$), we have

$$\overline{V}_{h,i}^k(s) \geq V_{h,i}^{\dagger, \widehat{\pi}_{h,-i}^k}(s), \quad \underline{V}_{h,i}^k(s) \leq V_{h,i}^{\widehat{\pi}_h^k}(s)$$

for all $(i, s, h, k) \in [m] \times \mathcal{S} \times [H] \times [K]$. Then, choosing $h = 1$ and $s = s_1$, we have

$$V_{1,i}^{\dagger, \widehat{\pi}_{1,-i}^k}(s_1) - V_{1,i}^{\widehat{\pi}_1^k}(s_1) \leq \overline{V}_{1,i}^k(s_1) - \underline{V}_{1,i}^k(s_1).$$

Moreover, by (7), value function of certified policy can be decomposed as

$$V_{1,i}^{\widehat{\pi}}(s) = \frac{1}{K}\sum_{k=1}^K V_{1,i}^{\widehat{\pi}_1^k}(s), \quad V_{1,i}^{\mu_1, \widehat{\pi}_{-i}}(s_1) = \frac{1}{K}\sum_{k=1}^K V_{1,i}^{\mu_1, \widehat{\pi}_{1,-i}^k}(s_1)$$

where the decomposition is due to the first line in the Algorithm 2: sample $k \leftarrow \text{Uniform}([K])$.

So we have

$$V_{1,i}^{\dagger,\widehat{\pi}^{-i}}(s_1) - V_{1,i}^{\widehat{\pi}}(s_1) \le \frac{1}{K}\sum_{k=1}^{K}\left(V_{1,i}^{\dagger,\widehat{\pi}_1^k,-i}(s_1) - V_{1,i}^{\widehat{\pi}_1^k}(s_1)\right)$$

$$\le \frac{1}{K}\sum_{k=1}^{K}\left(\overline{V}_{1,i}^k(s_1) - \underline{V}_{1,i}^k(s_1)\right).$$

To prove $\widehat{\pi}$ is an approximate CCE, we only need to bound $\sum_{k=1}^{K}\left(\overline{V}_{1,i}^k(s_1) - \underline{V}_{1,i}^k(s_1)\right)$. Letting $\delta_{h,i}^k := \overline{V}_{h,i}^k(s_h^k) - \underline{V}_{h,i}^k(s_h^k)$ and $t = N_h^k(s_h^k)$. Suppose $s_h^k$ was previously visited at episodes $k^1,\ldots,k^t$ at the $h$-th step. By the update rule,

$$\delta_{h,i}^k = \overline{V}_{h,i}^k(s_h^k) - \underline{V}_{h,i}^k(s_h^k)$$

$$= \alpha_t^0 H + \sum_{j=1}^{t}\alpha_t^j[\overline{V}_{h+1,i}^k(s_{h+1}^{k^j}) - \underline{V}_{h+1,i}^k(s_{h+1}^{k^j}) + 2\overline{\beta}_j]$$

$$= \alpha_t^0 H + \sum_{j=1}^{t}\alpha_t^j\delta_{h+1,i}^{k^j} + \sum_{j=1}^{t}2\alpha_t^j\overline{\beta}_j$$

$$= \alpha_t^0 H + \sum_{j=1}^{t}\alpha_t^j\delta_{h+1,i}^{k^j} + 2c\sum_{j=1}^{t}\alpha_t^j\sqrt{\frac{A_i H^3\iota}{j}} + 4c\sum_{j=1}^{t}\alpha_t^j\frac{H^2\iota}{j}.$$

We can use Lemma B.1 which gives $\sum_{j=1}^{t}\alpha_t^j/\sqrt{j} \le 2/t$ and $\max_{j\le t}\alpha_t^j \le 2H/t$ to get

$$\delta_{h,i}^k \le \alpha_t^0 H + \sum_{j=1}^{t}\alpha_t^j\delta_{h+1,i}^{k^j} + 4c\sqrt{H^3 A_i\iota/t} + 8cH^3\iota(1+\log t)/t,$$

where we also uses $\sum_{j=1}^{t}1/j \le 1 + \log t$.

Taking the summation w.r.t. k, we begin by the first two terms;

$$\sum_{k=1}^{K}\alpha_t^0 H = \sum_{k=1}^{K}H\mathbf{1}\{t=0\} = \sum_{k=1}^{K}H\mathbf{1}\{N_h^k(s_h^k)=0\} \le SH.$$

$$\sum_{k=1}^{K}\sum_{j=1}^{N_h^k(s_h^k)}\alpha_{N_h^k(s_h^k)}^j\delta_{h+1,i}^{k_h^j(s_h^k)} \overset{(i)}{\le} \sum_{k'=1}^{K}\delta_{h+1,i}^{k'}\sum_{j=N_h^{k'}(s_h^{k'})+1}^{\infty}\alpha_j^{N_h^{k'}(s_h^{k'})} \overset{(ii)}{\le} \left(1+\frac{1}{H}\right)\sum_{k'=1}^{K}\delta_{h+1,i}^{k'},$$

where (i) is by changing the order of summation and (ii) is by Lemma B.1.

So

$$\sum_{k=1}^{K}\delta_{h,i}^k \le SH + \left(1+\frac{1}{H}\right)\sum_{k=1}^{K}\delta_{h+1,i}^k + 4c\sqrt{H^3 A_i\iota}\sum_{k=1}^{K}\frac{1}{\sqrt{N_h^k(s)}} + 8cH^3\iota\sum_{k=1}^{K}\frac{1+\log N_h^k(s)}{N_h^k(s)}.$$

By pigeonhole argument,

$$\sum_{k=1}^{K}\frac{1}{\sqrt{N_h^k(s)}} = \sum_{s\in\mathcal{S}}\sum_{n=1}^{N_h^K(s)}\frac{1}{\sqrt{n}} \le \mathcal{O}(1)\sqrt{SK}.$$

Similarly,

$$\sum_{k=1}^{K}\frac{1+\log N_h^k(s)}{N_h^k(s)} \le \mathcal{O}(1)(1+\log(K)S(1+\log(K/S))) \le \mathcal{O}(1)S(\iota+\iota^2) \le \mathcal{O}(1)S\iota^2,$$

where we assume $\iota \geq 1$. So we have

$$\sum_{k=1}^{K} \delta_{h,i}^{k} \leq SH + \left(1 + \frac{1}{H}\right) \sum_{k=1}^{K} \delta_{h+1,i}^{k} + \mathcal{O}(1)\sqrt{H^3 A_i S K \iota} + \mathcal{O}(1)SH^3\iota^3.$$

Recursing this argument for $h \in [H]$ gives

$$\sum_{k=1}^{K} \delta_{1,i}^{k} \leq eH^2 S + \mathcal{O}(1)\sqrt{H^5 A_i S K \iota} + \mathcal{O}(1)SH^4\iota^3.$$

To conclude,

$$V_{1,i}^{\dagger,\widehat{\pi}-i}(s_1) - V_{1,i}^{\widehat{\pi}}(s_1) \leq \frac{1}{K}\sum_{k=1}^{K}\left(\overline{V}_{1,i}^{k}(s_1) - \underline{V}_{1,i}^{k}(s_1)\right) = \frac{1}{K}\sum_{k=1}^{K}\delta_{1,i}^{k}$$

$$\leq \mathcal{O}(1)SH^4\iota^3/K + \mathcal{O}\left(\sqrt{H^5 S \max_{i\in[m]} A_i \iota/K}\right).$$

Therefore, $K \geq \mathcal{O}(\frac{H^5 S \max_{i\in[m]} A_i \iota}{\varepsilon^2} + \frac{SH^4\iota^3}{\varepsilon})$ guarantees that we have $V_{1,i}^{\dagger,\widehat{\pi}-i}(s_1) - V_{1,i}^{\widehat{\pi}}(s_1) \leq \varepsilon$ for all $i \in [m]$. This complete the proof of Theorem 2. $\qquad\square$

## C   PROOFS FOR SECTION 4

In this section we prove Theorem 5. We first define a set of *lower* value estimates $\underline{V}_{h,i}^{k}(s)$ (along with the upper estimates used in Algorithm 1) via the following update rule:

$$\underline{V}_{h,i}(s_h) \leftarrow (1 - \alpha_t)\,\underline{V}_{h,i}(s_h) + \alpha_t\left(r_{h,i}(s_h, a_h, \boldsymbol{a}_{h,-i}) + \underline{V}_{h+1,i}(s_{h+1}) - \overline{\beta}_t\right). \qquad (10)$$

We emphasize that $\underline{V}_{h,i}^{k}(s)$ are analyses quantities only for simplifying the proof, and are not used by the algorithm.

We restate and use several notations we introduced in the last section. At the beginning of any episode $k$ for a particular state $s_h$, we denote $k_h^1(s_h) < \cdots < k_h^{N_h^k(s_h)}(s_h) < k$ to be all the episodes that the state $s_h$ was visited, where $N_h^k(s_h)$ is the times the state $s_h$ has been visited before the start of $k$-th episode. When there is no confusion, we sometimes will write $k^j = k_h^j(s_h)$ in short. For player $i$, we let $\underline{V}_{h,i}^{k}, \overline{V}_{h,i}^{k}, \mu_i^k$ denote the values and policies maintained by Algorithm 4 at the beginning of $k$-th episode, and $\boldsymbol{a}_h^k$ denote taken action at step $h$ and episode $k$. For any joint policy $\mu_h$ (over all players), reward function $r$ and value function $V$, we define the operators $\mathbb{P}_h$ and $\mathbb{D}_{\mu_h}$ as

$$[\mathbb{P}_h V](s, \boldsymbol{a}) := \mathbb{E}_{s'\sim\mathbb{P}_h(\cdot|s,\boldsymbol{a})}V(s'),$$

$$\mathbb{D}_{\mu_h}[r + \mathbb{P}_h V](s) := \mathbb{E}_{\boldsymbol{a}_h\sim\mu_h}[r(s,\boldsymbol{a}_h) + \mathbb{E}_{s_{h+1}\sim\mathbb{P}_h(\cdot|s,\boldsymbol{a}_h)}V(s_{h+1})].$$

The following lemma is the same as Lemma B.2 in the CCE case (except that for a different algorithm).

**Lemma C.1** (Update rule). *Fix a state $s$ in time step $h$ and fix an episode $k$, let $t = N_h^k(s)$ and suppose $s$ was previously visited at episodes $k^1 < \cdots < k^t < k$ at the $h$-th step. The update rule for $\overline{V}_{h,i}(s)$ and $\underline{V}_{h,i}(s)$ in Algorithm 1 and (10) gives the following equations:*

$$\overline{V}_{h,i}^{k}(s) = \alpha_t^0 H + \sum_{j=1}^{t} \alpha_t^j\left[r_{h,i}\left(s, a_h^{k^j}, \boldsymbol{a}_{h,-i}^{k^j}\right) + \overline{V}_{h+1,i}^{k^j} + \overline{\beta}_j\right],$$

$$\underline{V}_{h,i}^{k}(s) = \sum_{j=1}^{t} \alpha_t^j\left[r_{h,i}\left(s, a_h^{k^j}, \boldsymbol{a}_{h,-i}^{k^j}\right) + \underline{V}_{h+1,i}^{k^j}\left(s_{h+1}^{k^j}\right) - \overline{\beta}_j\right].$$

We next prove the following lemma which helps explain our choice of the bonus term $\overline{\beta}_t$. The constant $c$ in $\overline{\beta}_t$ is the same with the constant $c$ in this lemma. For any policy modification $\varphi_i : \mathcal{A}_i \to \mathcal{A}_i$ for the $i^{\text{th}}$ player and one-step policy $\pi_h : \mathcal{S} \to \Delta_{\mathcal{A}}$ for any $h$, the modified policy $\varphi_i \diamond \pi_h$ is defined as follows: if $\pi_h$ chooses to play $\boldsymbol{a} = (a_1, \ldots, a_m)$, the modified policy $\varphi_i \diamond \pi_h$ will play $(a_1, \ldots, a_{i-1}, \varphi_i(a_i), a_{i+1}, \ldots, a_m)$. Moreover, for $\pi_{h,i} : \mathcal{S} \to \Delta_{\mathcal{A}_i}$, policy $\varphi_i \diamond \pi_{h,i}$ chooses $\varphi_i(a)$ when $\pi_{h,i}$ chooses $a$.

**Lemma C.2** (Per-state guarantee). *Fix a state $s$ in time step $h$ and fix an episode $k$, let $t = N_h^k(s)$ and suppose $s$ was previously visited at episodes $k^1 < \cdots < k^t < k$ at the $h$-th step. With probability $1 - \frac{p}{2}$, for any $(i, s, h, t) \in [m] \times \mathcal{S} \times [H] \times [K]$, there exist a constant $c$ s.t.*

$$\sup_{\varphi_i : \mathcal{A}_i \to \mathcal{A}_i} \sum_{j=1}^{t} \alpha_t^j \mathbb{D}_{\varphi_i \diamond \mu_h^{k^j}} \left( r_{h,i} + \mathbb{P}_h \min\{\overline{V}_{h+1,i}^{k^j}, H - h\} \right)(s)$$

$$- \sum_{j=1}^{t} \alpha_t^j \left[ r_{h,i} \left( s, a_h^{k^j}, \boldsymbol{a}_{h,-i}^{k^j} \right) + \min\{\overline{V}_{h+1,i}^{k^j}(s_{h+1}^{k^j}), H - h\} \right] \le cH^2 A_i \sqrt{2\iota/t} + cH^2 A_i \iota/t.$$

**Proof of Lemma C.2**     First, like the prove in Lemma B.3, we decompose

$$\sup_{\varphi_i : \mathcal{A}_i \to \mathcal{A}_i} \sum_{j=1}^{t} \alpha_t^j \mathbb{D}_{\varphi_i \diamond \mu_h^{k^j}} \left( r_{h,i} + \mathbb{P}_h \min\{\overline{V}_{h+1,i}^{k^j}, H - h\} \right)(s)$$

$$- \sum_{j=1}^{t} \alpha_t^j \left[ r_{h,i} \left( s, a_h^{k^j}, \boldsymbol{a}_{h,-i}^{k^j} \right) + \min\{\overline{V}_{h+1,i}^{k^j}(s_{h+1}^{k^j}), H - h\} \right]$$

into $R^\star(i, s, h, t) + U(i, s, h, t)$ where

$$R^\star(i, s, h, t) := \sup_{\varphi_i : \mathcal{A}_i \to \mathcal{A}_i} \sum_{j=1}^{t} \alpha_t^j \mathbb{D}_{\varphi_i \diamond \mu_h^{k^j}} \left( r_{h,i} + \mathbb{P}_h \min\{\overline{V}_{h+1,i}^{k^j}, H - h\} \right)(s)$$

$$- \sum_{j=1}^{t} \alpha_t^j \mathbb{E}_{\boldsymbol{a}_{-i} \sim \mu_{h,-i}^{k^j}} \left[ r_{h,i} \left( s, a_h^{k^j}, \boldsymbol{a}_{-i} \right) + \mathbb{E}_{s_{h+1} \sim \mathbb{P}_h(\cdot | s, a_h^{k^j}, \boldsymbol{a}_{-i})} \min\{\overline{V}_{h+1,i}^{k^j}(s_{h+1}), H - h\} \right],$$

and

$$U(i, s, h, t) := \sum_{j=1}^{t} \alpha_t^j \mathbb{E}_{\boldsymbol{a}_{-i} \sim \mu_{h,-i}^{k^j}} \left[ r_{h,i} \left( s, a_h^{k^j}, \boldsymbol{a}_{-i} \right) + \mathbb{E}_{s_{h+1} \sim \mathbb{P}_h(\cdot | s, a_h^{k^j}, \boldsymbol{a}_{-i})} \min\{\overline{V}_{h+1,i}^{k^j}(s_{h+1}), H - h\} \right]$$

$$- \sum_{j=1}^{t} \alpha_t^j \left[ r_{h,i} \left( s, a_h^{k^j}, \boldsymbol{a}_{h,-i}^{k^j} \right) + \min\{\overline{V}_{h+1,i}^{k^j}(s_{h+1}^{k^j}), H - h\} \right].$$

We first bound $U(i, s, h, t)$. By the same reason in proof of Lemma B.3, we can apply Azuma-Hoeffding inequality. Note that $\sum_{j=1}^{t} (\alpha_t^j)^2 \le 2H/t$ by Lemma B.1. Using Azuma-Hoeffding inequality, we have with probability at least $1 - \frac{p}{4mHSK}$,

$$U(i, s, h, t) \le \sqrt{2H^2 \log(4mHSK/p) \sum_{j=1}^{t} (\alpha_t^j)^2} \le 2\sqrt{H^3 \iota/t}.$$

After taking a union bound, the following statement is true with probability at least $1 - p/4$,

$$U(i, s, h, t) \le 2\sqrt{H^3 \iota/t} \quad \text{for all } (i, s, h, t) \in [m] \times \mathcal{S} \times [H] \times [K].$$

Then we bound $R^\star(i, s, h, t)$. For fixed $(i, s, h)$, we define loss function

$$\ell_j(a) = \frac{1}{H} \mathbb{E}_{a_i = a, \boldsymbol{a}_{-i} \sim \mu_{h,-i}^{k^j}} [H - h + 1 - r_{h,i}(s, \boldsymbol{a}) - \mathbb{E}_{s_{h+1} \sim \mathbb{P}_h(\cdot | s, a, \boldsymbol{a}_{-i})} \min\{\overline{V}_{h+1,i}^{k^j}(s_{h+1}), H - h\}].$$

---

**Algorithm 6** Correlated policy $\widehat{\pi}_h^k$ for general-sum Markov games

1: **for** step $h' = h, \ldots, H$ **do**
2:     Observe $s_{h'}$, and set $t \leftarrow N_{h'}^k(s_{h'})$.
3:     Sample $l \in [t]$ with $\mathbb{P}(l = j) = \alpha_t^j$.
4:     Update $k \leftarrow k_t^l(s_{h'})$.
5:     Jointly take action $(a_{h',1}, a_{h',2}, \ldots, a_{h',m}) \sim \prod_{i=1}^m \mu_{h',i}^k(\cdot|s_{h'})$.

---

Then we have $\ell_j(a) \in [0, 1]$ for all $a$ and the realized loss function:

$$\tilde{\ell}_j(a_h^{k^j}) = \frac{1}{H}[H - h + 1 - r_{h,i}(s, a_h^{k^j}, \boldsymbol{a}_{-i}^{k^j}) - \min\{\overline{V}_{h+1,i}^{k^j}(s_{h+1}^{k^j}), H - h\}] \in [0, 1]$$

is an unbiased estimator of $\ell_j(a_h^{k^j})$. Then $R^\star(i, s, h, t)$ can be written as

$$R^\star(i, s, h, t) = H \sup_{\varphi_i:\mathcal{A}_i \to \mathcal{A}_i} \sum_{j=1}^t \alpha_t^j[\ell_j(a_{h,i}^{k^j}) - \langle \varphi_i \diamond \mu_{h,i}^{k^j}, \ell_j \rangle].$$

Now, for any fixed step $h$ and state $s$, the distributions $\left\{q_h^b(\cdot|s)\right\}_b$ and visitation counts $\left\{N_h^b(s)\right\}_b$ are only updated at episodes $k_h^1(s), \ldots, k_h^t(s)$. Further, these updates are exactly equivalent to the mixed-expert FTRL update algorithm which we describe in Algorithm 8. Therefore, the $R^\star(i, s, h, t)$ above can be bounded by the weighted swap regret bound of Lemma F.1 (choosing the log term as $\iota = 4 \log \frac{10mSAHK}{p}$) to yield that

$$H \sup_{\varphi_i:\mathcal{A}_i \to \mathcal{A}_i} \sum_{j=1}^t \alpha_t^j[\ell_j(a_{h,i}^{k^j}) - \langle \varphi_i \diamond \mu_{h,i}^{k^j}, \ell_j \rangle] \leq 40H^2 A_i \sqrt{\iota/t} + 40H^2 A_i \iota/t \quad \text{for all } t \in [K]$$

with probability at least $1 - p/(4mSH)$. Taking a union bound over all $(i, s, h) \in [m] \times \mathcal{S} \times [H]$, we have with probability at least $1 - p/4$,

$$R^\star(i, s, h, t) \leq 40H^2 A_i \sqrt{\iota/t} + 40H^2 A_i \iota/t \quad \text{for all } (i, s, h, t) \in [m] \times \mathcal{S} \times [H] \times [K].$$

Finally, we conclude that with probability at least $1 - p/2$, we have

$$U(i, s, h, t) + R^\star(i, s, h, t) \leq cH^2 A_i \sqrt{\iota/t} + cH^2 A_i \iota/t \quad \text{for all } (i, s, h, t) \in [m] \times \mathcal{S} \times [H] \times [K]$$

for some absolute constant $c$. $\qquad\qquad\qquad\qquad\qquad\qquad\qquad\qquad\qquad\qquad\qquad\qquad\qquad\qquad\quad$ $\square$

We define the auxiliary certified policies $\widehat{\pi}_h^k$ in Algorithm 6 (same as Algorithm 5 for the CCE case but repeated here for clarity). Again, we have the following relationship:

$$V_{1,i}^{\widehat{\pi}}(s) = \frac{1}{K} \sum_{k=1}^K V_{1,i}^{\widehat{\pi}_1^k}(s). \tag{11}$$

**Definition C.3** (Policy modification starting from the $h$-th step). *A strategy modification starting from the $h$-th step $\phi_{\geq h} := \{\phi_{h',s}\}_{(h',s) \in \{h,h+1,\ldots,H\} \times \mathcal{S}}$ for player $i$ is a set of $S \times (H - h + 1)$ functions $\phi_{h',s} : (\mathcal{S} \times \mathcal{A})^{h'-h} \times \mathcal{A}_i \to \mathcal{A}_i$. This strategy modification $\phi_{\geq h}$ can be composed with any policy $\pi \in \Pi_{\geq h}$ (as in Definition B.4) to give a modified policy $\phi_{\geq h} \diamond \pi$ defined as follows: At any step $h' \geq h$ and state $s$ with the history information starting from the $h$-th step $\tau_{h:h'-1} = (s_h, \boldsymbol{a}_h, \cdots, s_{h'-1}, \boldsymbol{a}_{h'-1})$, if $\pi$ chooses to play $\boldsymbol{a} = (a_1, \ldots, a_m)$, the modified policy $\phi \diamond \pi$ will play $(a_1, \ldots, a_{i-1}, \phi_{h',s}(\tau_{h:h'-1}, a_i), a_{i+1}, \ldots, a_m)$. We use $\Phi_{\geq h,i}$ denote the set of all such possible strategy modifications for player $i$.*

For any $\phi \in \Phi_{\geq h,i}$, $\phi \diamond \widehat{\pi}_h^k$ also doesn't depend on the history before the $h$-th step, so $\phi \diamond \widehat{\pi}_h^k \in \Pi_{\geq h}$, which implies that $V_{h,i}^{\phi \diamond \widehat{\pi}_h^k}(s)$ is well-defined in (8).

**Lemma C.4** (Valid upper and lower bounds). *We have*

$$\overline{V}_{h,i}^k(s) \geq \sup_{\phi \in \Phi_{\geq h,i}} V_{h,i}^{\phi \diamond \widehat{\pi}_h^k}(s), \quad \underline{V}_{h,i}^k(s) \leq V_{h,i}^{\widehat{\pi}_h^k}(s)$$

*for all* $(i, k, h, s) \in [m] \times [K] \times [H] \times \mathcal{S}$ *with probability at least* $1 - p/2$.

**Proof of Lemma C.4**    We prove this lemma by backward induction over $h \in [H+1]$. The base case of $h = H+1$ is true as all the value functions equal 0. Suppose the claim is true for $h+1$. We begin with upper bounding $\sup_{\phi \in \Phi_{\geq h,i}} V_{h,i}^{\phi \diamond \widehat{\pi}_h^k}(s)$. Let $t = N_h^k(s)$ and $k^j = k_h^j(s)$ for $1 \leq j \leq t$ to be the $j$'th time that $s$ is previously visited. By the definition of certified policies $\widehat{\pi}_{h,i}^k$ and by the value iteration formula of MGs, we have for any $\phi \in \Phi_{\geq h,i}$,

$$V_{h,i}^{\phi \diamond \widehat{\pi}_h^k}(s) = \sum_{j=1}^t \alpha_t^j \mathbb{E}_{\boldsymbol{a} \sim \phi_{h,i} \diamond \mu_h^{k^j}} \left[ r_{h,i}(s, \boldsymbol{a}) + \mathbb{E}_{s' \sim \mathbb{P}_h(\cdot|s,\boldsymbol{a})} V_{h+1,i}^{(\phi_{\geq h+1}|s,\boldsymbol{a}) \diamond \widehat{\pi}_{h+1}^{k^j}}(s') \right]$$

$$\leq \sum_{j=1}^t \alpha_t^j \mathbb{E}_{\boldsymbol{a} \sim \phi_{h,i} \diamond \mu_h^{k^j}} \left[ r_{h,i}(s, \boldsymbol{a}) + \mathbb{E}_{s' \sim \mathbb{P}_h(\cdot|s,\boldsymbol{a})} \sup_{\phi \in \Phi_{\geq h+1,i}} V_{h+1,i}^{\phi \diamond \widehat{\pi}_{h+1}^{k^j}}(s') \right].$$

Here, $\phi_{h,i}$ is the strategy modification function $\phi$ at the $h$-th step, and $(\phi_{\geq h+1}|s, \boldsymbol{a})$ is the modification $\phi$ from the $h + 1$-th step with history information to be $s_h = s$ and $\boldsymbol{a}_h = \boldsymbol{a}$. By the definition of $\Phi_{\geq h,i}$, we have $(\phi_{\geq h+1}|s, \boldsymbol{a}) \in \Phi_{\geq h+1,i}$ which implies the above inequality. By taking supremum over $\phi \in \Phi_{\geq h,i}$ and using the definition of the operator $\mathbb{D}$ in (6), we have

$$\sup_{\phi \in \Phi_{\geq h,i}} V_{h,i}^{\phi \diamond \widehat{\pi}_h^k}(s) \leq \sup_{\varphi_i : \mathcal{A}_i \to \mathcal{A}_i} \sum_{j=1}^t \alpha_t^j \mathbb{D}_{\varphi_i \diamond \mu_h^{k^j}} [r_{h,i} + \mathbb{P}_h \sup_{\phi \in \Phi_{\geq h+1,i}} V_{h+1,i}^{\phi \diamond \widehat{\pi}_{h+1}^{k^j}}](s).$$

Condition on the high probability event (with probability at least $1 - p/2$) in Lemma C.2, we can use the inductive hypothesis to obtain

$$\sup_{\varphi_i : \mathcal{A}_i \to \mathcal{A}_i} \sum_{j=1}^t \alpha_t^j \mathbb{D}_{\varphi_i \diamond \mu_h^{k^j}} [r_{h,i} + \mathbb{P}_h \sup_{\phi \in \Phi_{\geq h+1,i}} V_{h+1,i}^{\phi \diamond \widehat{\pi}_{h+1}^{k^j}}](s)$$

$$\leq \sup_{\varphi_i : \mathcal{A}_i \to \mathcal{A}_i} \sum_{j=1}^t \alpha_t^j \mathbb{D}_{\varphi_i \diamond \mu_h^{k^j}} [r_{h,i} + \mathbb{P}_h \min\{\overline{V}_{h+1,i}^{k^j}, H - h\}](s)$$

$$\leq \sum_{j=1}^t \alpha_t^j \left[ r_{h,i}\left(s, a_h^{k^j}, \boldsymbol{a}_{h,-i}^{k^j}\right) + \min\{\overline{V}_{h+1,i}^{k^j}(s_{h+1}^{k^j}), H - h\} \right] + cH^2 A_i \sqrt{\iota/t} + cH^2 A_i \iota/t$$

$$\overset{(i)}{\leq} \sum_{j=1}^t \alpha_t^j \left[ r_{h,i}\left(s, a_h^{k^j}, \boldsymbol{a}_{h,-i}^{k^j}\right) + \min\{\overline{V}_{h+1,i}^{k^j}(s_{h+1}^{k^j}), H - h\} + \overline{\beta}_j \right]$$

$$\leq \sum_{j=1}^t \alpha_t^j \left[ r_{h,i}\left(s, a_h^{k^j}, \boldsymbol{a}_{h,-i}^{k^j}\right) + \overline{V}_{h+1,i}^{k^j} + \overline{\beta}_j \right]$$

$$= \overline{V}_{h,i}^k(s).$$

Here, (i) uses our choice of $\overline{\beta}_j = cH^2 A_i \sqrt{\frac{\iota}{j}} + 2c\frac{H^2 A_i \iota}{j}$ and $\frac{1}{\sqrt{j}} \leq \sum_{j=1}^t \frac{\alpha_t^j}{\sqrt{j}}$, $\frac{1}{t} \leq \sum_{j=1}^t \frac{2\alpha_t^j}{j}$, so that $cH^2 A_i \sqrt{\iota/t} + cH^2 A_i \iota/t \leq \sum \alpha_t^j \overline{\beta}_j$.

Meanwhile, for $\underline{V}_{h,i}^k(s)$, by the definition of certified policy and inductive hypothesis,

$$V_{h,i}^{\widehat{\pi}_h^k}(s) = \sum_{j=1}^t \alpha_t^j \mathbb{D}_{\mu_h^{k^j}} [r_{h,i} + \mathbb{P}_h V_{h+1,i}^{\widehat{\pi}_{h+1}^{k^j}}](s)$$

$$\geq \sum_{j=1}^t \alpha_t^j \mathbb{D}_{\mu_h^{k^j}} [r_{h,i} + \mathbb{P}_h \max\{\underline{V}_{h+1,i}^{k^j}, 0\}](s).$$

Note that $\left\{ \mathbb{D}_{\mu_h^{k^j}}[r_{h,i} + \mathbb{P}_h \max\left\{\underline{V}_{h+1,i}^{k^j}, 0\right\}](s) - \left[ r_{h,i}\left(s, a_h^{k^j}, \boldsymbol{a}_{h,-i}^{k^j}\right) + \max\{\underline{V}_{h+1,i}^{k^j}(s_{h+1}^{k^j}), 0\} \right] \right\}_{j \geq 1}$
is a martingale difference sequence w.r.t. the filtration $\{\mathcal{G}_j\}_{j \geq 0}$, which is defined in the proof of
Lemma B.3. So by Azuma-Hoeffding inequality, with probability at least $1 - \frac{p}{2mSKH}$,

$$
\sum_{j=1}^{t} \alpha_t^j \mathbb{D}_{\mu_h^{k^j}}[r_{h,i} + \mathbb{P}_h \max\{\underline{V}_{h+1,i}^{k^j}, 0\}](s)
$$
$$
\geq \sum_{j=1}^{t} \alpha_t^j \left[ r_{h,i}\left(s, a_h^{k^j}, \boldsymbol{a}_{h,-i}^{k^j}\right) + \max\{\underline{V}_{h+1,i}^{k^j}(s_{h+1}^{k^j}), 0\} \right] - 2\sqrt{\frac{H^3 \iota}{t}}. \tag{12}
$$

On this event, we have

$$
\sum_{j=1}^{t} \alpha_t^j \mathbb{D}_{\mu_h^{k^j}}[r_{h,i} + \mathbb{P}_h \max\left\{\underline{V}_{h+1,i}^{k^j}, 0\right\}](s)
$$
$$
\overset{(i)}{\geq} \sum_{j=1}^{t} \alpha_t^j \left[ r_{h,i}\left(s, a_h^{k^j}, \boldsymbol{a}_{h,-i}^{k^j}\right) + \max\left\{\underline{V}_{h+1,i}^{k^j}(s_{h+1}^{k^j}), 0\right\} - \overline{\beta}_j \right]
$$
$$
\geq \sum_{j=1}^{t} \alpha_t^j \left[ r_{h,i}\left(s, a_h^{k^j}, \boldsymbol{a}_{h,-i}^{k^j}\right) + \underline{V}_{h+1,i}^{k^j}(s_{h+1}^{k^j}) - \overline{\beta}_j \right]
$$
$$
= \underline{V}_{h,i}^{k}(s).
$$

Here, (i) uses $\sum_{j=1}^{t} \alpha_t^j \beta_j \geq 2 \sum_{j=1}^{t} \alpha_t^j \sqrt{H^3 \iota / j} \geq 2\sqrt{H^3 \iota / t}$.

As a result, the backward induction would work well for all $h$ as long as the inequalities in
Lemma C.2 and (12) hold for all $(i, s, h, k) \in [m] \times \mathcal{S} \times [H] \times [K]$. Taking a union bound in all
$(i, s, h, k) \in [m] \times \mathcal{S} \times [H] \times [K]$, we have with probability at least $1 - p/2$, the inequality in (12) is
true simultaneously for all $(i, s, h, k) \in [m] \times \mathcal{S} \times [H] \times [K]$. Therefore the inequalities in Lemma
B.3 and (12) hold simultaneously for all $(i, s, h, k) \in [m] \times \mathcal{S} \times [H] \times [K]$ with probability at least
$1 - p$. This finishes the proof of this lemma. $\qquad\square$

Equipped with these lemmas, we are ready to prove Theorem 5.

**Proof of Theorem 5**    Conditional on the high probability event in Lemma C.4 (this happens with
probability at least $1 - p$), we have

$$
\overline{V}_{h,i}^{k}(s) \geq \sup_{\phi \in \Phi_{\geq h,i}} V_{h,i}^{\phi \diamond \widehat{\pi}_h^k}(s), \quad \underline{V}_{h,i}^{k}(s) \leq V_{h,i}^{\widehat{\pi}_h^k}(s)
$$

for all $(i, s, h, k) \in [m] \times \mathcal{S} \times [H] \times [K]$. Then, choosing $h = 1$ and $s = s_1$, we have

$$
\sup_{\phi \in \Phi_i} V_{1,i}^{\phi \diamond \widehat{\pi}_1^k}(s_1) - V_{1,i}^{\widehat{\pi}_1^k}(s_1) \leq \overline{V}_{1,i}^{k}(s_1) - \underline{V}_{1,i}^{k}(s_1).
$$

Moreover, by (11), value function of certified policy can be decomposed as

$$
V_{1,i}^{\widehat{\pi}}(s_1) = \frac{1}{K} \sum_{k=1}^{K} V_{1,i}^{\widehat{\pi}_1^k}(s_1), \quad V_{1,i}^{\phi \diamond \widehat{\pi}}(s_1) = \frac{1}{K} \sum_{k=1}^{K} V_{1,i}^{\phi \diamond \widehat{\pi}_1^k}(s_1),
$$

where the decomposition is due to the first line in the Algorithm 2: sample $k \leftarrow \text{Uniform}([K])$.
Therefore we have the following bound on $\sup_{\phi \in \Phi_i} V_{1,i}^{\phi \diamond \widehat{\pi}}(s_1) - V_{1,i}^{\widehat{\pi}}(s_1)$:

$$
\sup_{\phi \in \Phi_i} V_{1,i}^{\phi \diamond \widehat{\pi}}(s_1) - V_{1,i}^{\widehat{\pi}}(s_1) = \sup_{\phi \in \Phi_i} \frac{1}{K} \sum_{k=1}^{K} V_{1,i}^{\phi \diamond \widehat{\pi}_1^k}(s_1) - V_{1,i}^{\widehat{\pi}}(s_1)
$$

$$\leq \frac{1}{K} \sum_{k=1}^{K} \left( \sup_{\phi \in \Phi_i} V_{1,i}^{\phi \diamond \widehat{\pi}_1^k}(s_1) - V_{1,i}^{\widehat{\pi}_1^k}(s_1) \right) \leq \frac{1}{K} \sum_{k=1}^{K} \left( \overline{V}_{1,i}^k(s_1) - \underline{V}_{1,i}^k(s_1) \right).$$

By Lemma C.4 Letting $\delta_{h,i}^k := \overline{V}_{h,i}^k(s_h^k) - \underline{V}_{h,i}^k(s_h^k)$ and $t = N_h^k(s_h^k)$. By the update rule, we have

$$\delta_{h,i}^k = \overline{V}_{h,i}^k(s_h^k) - \underline{V}_{h,i}^k(s_h^k)$$

$$= \alpha_t^0 H + \sum_{j=1}^{t} \alpha_t^j [\overline{V}_{h+1,i}^k(s_{h+1}^{k^j}) - \underline{V}_{h+1,i}^k(s_{h+1}^{k^j}) + 2\overline{\beta_j}]$$

$$= \alpha_t^0 H + \sum_{j=1}^{t} \alpha_t^j \delta_{h+1,i}^{k^j} + \sum_{j=1}^{t} 2\alpha_t^j \overline{\beta}_j$$

$$= \alpha_t^0 H + \sum_{j=1}^{t} \alpha_t^j \delta_{h+1,i}^{k^j} + 2cA_i H^2 \sum_{j=1}^{t} \alpha_t^j \sqrt{\frac{\iota}{j}} + 4c \sum_{j=1}^{t} \alpha_t^j \frac{H^2 A_i \iota}{j}.$$

Taking the summation w.r.t. k, by the same argument in the proof of Theorem 2, we can get

$$\max_{\phi \in \Phi} V_{1,i}^{\phi \diamond \widehat{\pi}}(s_1) - V_{1,i}^{\widehat{\pi}}(s_1) = \frac{1}{K} \sum_{k=1}^{K} \left( \overline{V}_{1,i}^k(s_1) - \underline{V}_{1,i}^k(s_1) \right) = \frac{1}{K} \sum_{k=1}^{K} \delta_{1,i}^k$$

$$\leq \mathcal{O}(1) S H^4 \max_{i \in [m]} A_i \iota^3 / K + \mathcal{O} \left( \sqrt{H^6 S \max_{i \in [m]} A_i^2 \iota / K} \right).$$

Therefore, if $K \geq \mathcal{O}(\frac{H^6 S \max_{i \in [m]} A_i^2 \iota}{\varepsilon^2} + \frac{H^4 S \max_{i \in [m]} A_i \iota^3}{\varepsilon})$, we have $\max_{\phi \in \Phi} V_{1,i}^{\phi \diamond \widehat{\pi}}(s_1) - V_{1,i}^{\widehat{\pi}}(s_1) \leq \varepsilon$ holds for all $i \in [m]$, which means $\widehat{\pi}$ is an $\varepsilon$-approximate CE. This completes the proof of Theorem 2. $\square$

# D PROOFS FOR SECTION 5

Here, we first define pure-strategy Nash equilibrium. We say policy $\pi$ is a pure strategy (deterministic policy) if and only if for any $(h, i, s) \in [H] \times [m] \times \mathcal{S}$, $\pi_{h,i}(a_i|s) = \mathbb{I}_{a_i = a'_{h,i}(s)}$ for some $a'_{h,i}(s)$. We say $\pi$ a *pure-strategy Nash equilibrium* if $\pi$ is a pure-strategy and is a Nash equilibrium. Similarly, we say $\pi$ is a pure strategy $\varepsilon$-approximate Nash equilibrium if $\pi$ is a pure strategy and NE-gap$(\pi) \leq \varepsilon$. Pure-strategy Nash equilibrium does not always exist in general-sum MGs, but is guaranteed to exist (as we will see) in Markov Potential Games.

## D.1 EXISTENCE OF PURE-STRATEGY NASH EQUILIBRIA IN MARKOV POTENTIAL GAMES

A particular property of MPGs is that, there always exists a pure-strategy Nash equilibrium. Such a property does not hold for every general-sum MG. Pure-strategy Nash equilibria are preferred in many scenarios since each player can take deterministic actions.

**Proposition D.1.** *For any Markov potential games, there exists a pure-strategy Nash equilibrium.*

See Theorem 3.1 in Leonardos et al. (2021) or Proposition 1 in Zhang et al. (2021) for a proof of Proposition D.1.

## D.2 THE UCBVI-UPLOW SUB-ROUTINE

In this subsection we consider the problem of learning a near optimal policy in the fixed horizon stochastic reward RL problem $\text{MDP}(H, \mathcal{S}, \mathcal{A}, \mathbb{P}, r)$. The setting is standard (c.f. Jin et al. (2018)) and is a special case of Markov games (c.f. Section 2) by setting the number of agents $m = 1$. We will use the same notations including policies and value functions as that of the Markov games as

---

**Algorithm 7** UCB-VI with Upper and Lower Confidence Bounds (UCBVI-UPLOW)

---

1: **Initialize:** For any $(s, a, h, s')$: $\overline{Q}_h(s, a) \leftarrow H$, $\underline{Q}_h(s, a) \leftarrow 0$, $N_h(s) = N_h(s, a) = N_h(s, a, s') \leftarrow 0$.

2: **for** episode $k = 1, \ldots, K$ **do**

3:     **for** step $h = H, \ldots, 1$ **do**

4:         **for** $(s, a) \in \mathcal{S} \times \mathcal{A}$ **do**

5:             Set $t \leftarrow N_h(s, a)$.

6:             **if** $t > 0$ **then**

7:                 $\beta \leftarrow \text{BONUS}(t, \widehat{\mathbb{V}}_h[(\overline{V}_{h+1} + \underline{V}_{h+1})/2](s, a))$ (c.f. Eq. (13))

8:                 $\gamma \leftarrow (c/H) \cdot \widehat{\mathbb{P}}_h(\overline{V}_{h+1} - \underline{V}_{h+1})(s, a)$.

9:                 $\overline{Q}_h(s, a) \leftarrow \min\left\{(r_h + \widehat{\mathbb{P}}_h\overline{V}_{h+1})(s, a) + \gamma + \beta, H\right\}$.

10:                $\underline{Q}_h(s, a) \leftarrow \max\left\{(r_h + \widehat{\mathbb{P}}_h\underline{V}_{h+1})(s, a) - \gamma - \beta, 0\right\}$

11:         **for** $s \in \mathcal{S}$ **do**

12:             $\pi_h(s) \leftarrow \arg\max_{a \in \mathcal{A}} \overline{Q}_h(s, a)$.

13:             $\overline{V}_h(s) \leftarrow \overline{Q}_h(s, \pi_h(s))$; $\underline{V}_h(s) \leftarrow \underline{Q}_h(s, \pi_h(s))$.

14:     Receive the initial state $s_1$ from the MDP.

15:     **for** step $h = 1, \ldots, H$ **do**

16:         Take action $a_h = \pi_h(s_h)$, observe reward $r_h$ and next state $s_{h+1}$.

17:         Increment $N_h(s_h)$, $N_h(s_h, a_h)$, and $N_h(s_h, a_h, s_{h+1})$ by 1.

18:         $\widehat{\mathbb{P}}_h(\cdot|s_h, a_h) \leftarrow N_h(s_h, a_h, \cdot)/N_h(s_h, a_h)$.

19: Let $(\overline{V}_h^k, \underline{V}_h^k, \pi^k)$ denote the value estimates and policy at the beginning of episode $k$.

20: **return** Policy $\pi^{k_\star}$ where $k_\star = \arg\min_{k \in [K]}(\overline{V}_1^k(s_1) - \underline{V}_1^k(s_1))$.

---

introduced in Section 2, except that we will omit the sub-scripts $i$'s since here we just have a single agent.

We consider the UCBVI-UPLOW algorithm (Algorithm 7), which is adapted from (Xie et al., 2021; Liu et al., 2021), for learning approximate optimal policy in reinforcement learning problems. Such an algorithm is used as a sub-routine in Algorithm 3 to learn approximate pure-strategy Nash equilibria in MPGs. We remark that although in Algorithm 3 we propose to use the UCBVI-UPLOW algorithm to search for a near optimal policy, many alternative algorithms can be used to find the near optimal policy (e.g., UCBVI (Azar et al., 2017) or Q-learning (Jin et al., 2018)). Here we choose the UCBVI-UPLOW algorithm because 1) it has a tight sample complexity bound; 2) it outputs a deterministic policy which can be used to find a *pure-strategy* approximate Nash equilibrium.

In the description of Algorithm 7, the $\widehat{\mathbb{P}}_h$ quantity appeared in lines 8,9,10, and 18 can be viewed either as a set of empirical probability distributions or as an operator: for any fixed $(s_h, a_h)$, we can view $\widehat{\mathbb{P}}_h(\cdot|s_h, a_h)$ as a probability distribution over $\mathcal{S}$; for any given function $V : \mathcal{S} \to \mathbb{R}$, we can view $\widehat{\mathbb{P}}_h$ as an operator by defining $(\widehat{\mathbb{P}}_h V)(s, a) := \mathbb{E}_{s' \sim \widehat{\mathbb{P}}_h(\cdot|s,a)}[V(s')]$. The $\widehat{\mathbb{V}}_h$ operator in line 7 is the empirical variance operator defined as $\widehat{\mathbb{V}}_h V = \widehat{\mathbb{P}}_h V^2 - (\widehat{\mathbb{P}}_h V)^2$. The BONUS function in the algorithm is chosen to be the Bernstein type bonus function

$$\text{BONUS}(t, \widehat{\sigma}^2) = c(\sqrt{\widehat{\sigma}^2 \iota/t} + H^2 S \iota/t). \tag{13}$$

We have the following sample complexity guarantee for the UCBVI-UPLOW algorithm returning an $\varepsilon$-approximate optimal policy.

**Lemma D.2.** *The* UCBVI-UPLOW *algorithm always returns a deterministic policy $\pi^{k_\star}$. Moreover, for any $p \in (0, 1]$, letting $\iota = \log(SAHK/p)$ and taking the number of episodes*

$$K \geq \mathcal{O}(H^3 S A \iota/\varepsilon^2 + H^3 S^2 A \iota^2/\varepsilon),$$

*then with probability at least $1 - p$, the returned policy $\pi^{k_\star}$ is $\varepsilon$-approximate optimal, i.e., $\sup_\mu V_1^\mu(s_1) - V_1^{\pi^{k_\star}}(s_1) \leq \varepsilon$.*

**Proof of Lemma D.2** First, $\pi^{k_\star}$ is obviously deterministic from line 12 of Algorithm 7.

The sample complexity guarantee of the UCBVI-UPLOW algorithm is a consequence of the sample complexity guarantee of the Nash-VI algorithm for learning Nash in zero-sum Markov games as proved in Liu et al. (2021).

More specifically, we denote the rewards and transition matrices by $\{r_h(s,a)\}_{h\in[H]}$ and $\mathbb{P} = \{P_h(s_{h+1}|s_h,a_h)\}_{h\in[H]}$ for the $\mathrm{MDP}(H,\mathcal{S},\mathcal{A},\mathbb{P},r)$. Such a MDP can be viewed as a zero-sum Markov game $\mathrm{MG}(H,\mathcal{S},\mathcal{A},\mathcal{B},\overline{\mathbb{P}},\bar{r})$, in which $(H,\mathcal{S},\mathcal{A})$ are the same as that of the MDP, the action space for the min-player is a singleton $\mathcal{B}=1$ so that the rewards $\bar{r}_h(s,a,b) \equiv r_h(s,a)$ do not depend on the action of the min-player and the transition matrices $\overline{\mathbb{P}}_h(\cdot|s,a,b) = \mathbb{P}_h(\cdot|s,a_1)$ do not depend on the action of the min-player either. Then, for any $\varepsilon$-approximate Nash equilibrium $(\mu,\nu)$ of the associated zero-sum Markov game, the max-player's policy $\mu$ must be $\varepsilon$-approximate optimal for the original MDP.

By this correspondence, the UCBVI-UPLOW algorithm is actually a specific version of the Nash-VI algorithm in Liu et al. (2021), and Line 12 in UCBVI-UPLOW is actually a specific version of line 12 in Nash-VI in Liu et al. (2021): this is because in this specific Markov game, $\overline{Q}(s,a,b)$ and $\underline{Q}(s,a,b)$ only depend on $s$ and $a$, so that $\pi_h(s) = \arg\max_{a\in\mathcal{A}} \overline{Q}_h(s,a)$ is actually in the CCE set $\mathrm{CCE}(\overline{Q}_h(s,a,b),\underline{Q}_h(s,a,b))$.

By this reduction and by Theorem 4 in Liu et al. (2021), this lemma is proved. $\qquad\square$

### D.3    PROOF OF THEOREM 7

We use superscript $t$ to represent variables at the $t$-th step (before $\pi$ is updated) of the while loop.

Because we can choose the log factor as $\iota = 4\log(\frac{mHSK \max_{i\in[m]} A_i}{\varepsilon p})$ (this doesn't affect the correctness of the theorem), for each execution of UCBVI-UPLOW, by Lemma D.2, it return a $\varepsilon/4$-optimal deterministic policy with probability at least $1 - \frac{p\varepsilon}{8m^2H}$. Taking a union bound, we have

$$\max_{\mu_i} V_{1,i}(\mu_i,\pi_{-i}^t) - V_{1,i}(\widehat{\pi}_i^t,\pi_{-i}^t) \le \varepsilon/4 \tag{14}$$

simultaneously for all $i \in [m]$ and $t \le 4mH/\varepsilon$ with probability at least $1 - p/2$. For the empirical estimator $\widehat{V}_{1,i}^t$, it's bounded in $[0,H]$. Thus by Hoeffding's inequality, for fixed $i \in [m]$ and $t$

$$\mathbb{P}(|\widehat{V}_{1,i}^t - V_{1,i}^t| \ge \varepsilon/8) \le 2\exp\left(-\frac{N\varepsilon^2}{32H^2}\right).$$

Choosing $N = CH^2\iota/\varepsilon^2$ for some large constant $C$, we have

$$\mathbb{P}(|\widehat{V}_{1,i}^t - V_{1,i}^t| \ge \varepsilon/8) \le \frac{\varepsilon p}{16m^2H}.$$

Apply this inequality to $\widehat{V}_{1,i}^t(\widehat{\pi}_i^t,\pi_{-i}^t)$ and $\widehat{V}_{1,i}^t(\pi^t)$ and taking a union bound, we have

$$|\widehat{V}_{1,i}^t(\widehat{\pi}_i^t,\pi_{-i}^t) - V_{1,i}(\widehat{\pi}_i^t,\pi_{-i}^t)| \le \varepsilon/8, \quad |\widehat{V}_{1,i}^t(\pi^t) - V_{1,i}(\pi^t)| \le \varepsilon/8 \tag{15}$$

simultaneously for all $i \in [m]$ and $t \le \frac{4mH}{\varepsilon}$ with probability at least $1 - p/2$. As a result, by (14) and (15), we have

$$\max_{\mu_i} V_{1,i}(\mu_i,\pi_{-i}^t) - V_{1,i}(\widehat{\pi}_i^t,\pi_{-i}^t) \le \varepsilon/4$$
$$|\widehat{V}_{1,i}^t(\widehat{\pi}_i^t,\pi_{-i}^t) - V_{1,i}(\widehat{\pi}_i^t,\pi_{-i}^t)| \le \varepsilon/8 \tag{16}$$
$$|\widehat{V}_{1,i}^t(\pi^t) - V_{1,i}(\pi^t)| \le \varepsilon/8$$

simultaneously for all $i \in [m]$ and $t \le 4mH/\varepsilon$ with probability at least $1 - p$. On this event,

$$\Delta_i^t = \widehat{V}_{1,i}^t(\widehat{\pi}_i^t,\pi_{-i}^t) - \widehat{V}_{1,i}^t(\pi^t)$$
$$\le V_{1,i}(\widehat{\pi}_i^t,\pi_{-i}^t) - V_{1,i}(\pi^t) + \varepsilon/4.$$

If the while loop doesn't end after the $t$-th iteration and $t \le 4mH/\varepsilon$, there exists $j^t$ s.t. $\Delta_{j^t}^t \ge \varepsilon/2$, so we have

$$
\begin{aligned}
\Phi(\pi^{t+1}) - \Phi(\pi^t) &= \Phi(\widehat{\pi}_{j^t}^t, \pi_{-j^t}^t) - \Phi(\pi^t) \\
&\overset{(i)}{=} V_{1,j^t}(\widehat{\pi}_{j^t}^t, \pi_{-j^t}^t) - V_{1,j^t}(\pi^t) \\
&\ge \Delta_{j^t}^t - \varepsilon/4 = \varepsilon/4.
\end{aligned}
$$

Here, $(i)$ follows the definition of potential function. Because $\Phi$ is bounded, the while loop ends within $4\Phi_{\max}/\varepsilon \le 4mH/\varepsilon$ steps. Therefore, (16) holds simultaneously for all $i \in [m]$ and $t$ before the end of while loop with probability at least $1 - p$. Again, on this event, if the while loop stops at the end of $t$-th step, we have $\max_{i \in [m]} \Delta_i^t \le \varepsilon/2$, then

$$
\begin{aligned}
\max_{\mu_i} V_{1,i}(\mu_i, \pi_{-i}^t) - V_{1,i}(\pi^t) &= \max_{\mu_i} V_{1,i}(\mu_i, \pi_{-i}^t) - V_{1,i}(\widehat{\pi}_i^t, \pi_{-i}^t) + V_{1,i}(\widehat{\pi}_i^t, \pi_{-i}^t) - V_{1,i}(\pi^t) \\
&\le \varepsilon/4 + \widehat{V}_{1,i}^t(\widehat{\pi}_i^t, \pi_{-i}^t) - \widehat{V}_{1,i}^t(\pi^t) + 2\varepsilon/8 \\
&= \varepsilon/2 + \Delta_i^t \\
&\le \varepsilon.
\end{aligned}
$$

So the returned policy $\pi^t$ is a $\varepsilon$-approximate Nash equilibrium. Moreover, since UCBVI-UPLOW outputs a pure-strategy policy and our initial policy is also a pure-strategy policy, we can conclude that with probability at least $1 - p$, within $4\Phi_{\max}/\varepsilon$ steps of the while loop, Algorithm 3 outputs an $\varepsilon$-approximate (pure-strategy) Nash equilibrium.

Finally, the number of episodes within each step of the while loop is

$$
\begin{aligned}
N + \sum_{i=1}^{m}(K_i + N) &= \mathcal{O}\left( \frac{H^3 S \sum_{i=1}^{m} A_i \iota}{\varepsilon^2} + \frac{H^3 S^2 \sum_{i=1}^{m} A_i \iota^2}{\varepsilon} + \frac{H^2 m \iota}{\varepsilon^2} \right) \\
&= \mathcal{O}\left( \frac{H^3 S \sum_{i=1}^{m} A_i \iota}{\varepsilon^2} + \frac{H^3 S^2 \sum_{i=1}^{m} A_i \iota^2}{\varepsilon} \right).
\end{aligned}
$$

So the total sample complexity (episodes) is at most

$$
K = \mathcal{O}\left( \frac{\Phi_{\max} H^3 S \sum_{i=1}^{m} A_i \iota}{\varepsilon^3} + \frac{\Phi_{\max} H^3 S^2 \sum_{i=1}^{m} A_i \iota^2}{\varepsilon^2} \right).
$$

This concludes the proof. □

## E  LOWER BOUND OF FINDING APPROXIMATE PURE-STRATEGY NASH EQUILIBRIUM

In this section, we present an result on the sample complexity lower bound for learning a pure-strategy Nash equilibrium in MPGs (a harder task than learning Nash as pure-strategy Nash is a stricter notion). We remark that our lower bounds are actually constructed on Markov Cooperative Games (MCGs) which is a subset of MPGs. Note that for MCGs, the potential function $\Phi$ is bounded in $[0, H]$, so by Theorem 7, the sample complexity of Nash-CA (Algorithm 3) is $\widetilde{\mathcal{O}}(\sum_{i=1}^{m} A_i/\varepsilon^3)$ highlighting the dependency on $\varepsilon$, $m$ and $A_i$, $i = 1, \ldots, m$. We would show this $\sum_{i=1}^{m} A_i$ dependency is inevitable for learning $\varepsilon$-approximate pure-strategy Nash equilibrium in MCGs by proving an $\Omega(\sum_{i=1}^{m} A_i/\varepsilon^2)$ lower bound.

We first present our main theorem:

**Theorem E.1** (Lower bound for learning pure-strategy Nash in MCGs). *Suppose $A_i = 2k$, $i = 1, \ldots, m$, $H \ge 2$, $S \ge 3$ and $m \ge 4$. Then, there exists an absolute constant $c_0$ such that for any $\varepsilon \le 0.4$ and any online finetuning algorithm $\mathcal{M}$ that outputs a pure-strategy policy $\widehat{\pi} = (\widehat{\pi}_1, \ldots, \widehat{\pi}_m)$, if the number of episodes*

$$
K \le c_0 \frac{H^2 \sum_{i=1}^{m} A_i}{\varepsilon^2} = c_0 \frac{2kmH^2}{\varepsilon^2},
$$

*then there exists general-sum Markov cooperative game $MG$ on which the algorithm $\mathcal{M}$ suffers from $\varepsilon/4$-suboptimality, i.e.*

$$\mathbb{E}_M NE\text{-}gap(\widehat{\pi}) \geq \varepsilon/4,$$

*where the expectation $\mathbb{E}_M$ is w.r.t. the randomness during the algorithm's execution within Markov game $MG$.*

This theorem can be viewed as a corollary of the following lemma by a simple reduction. We would prove this theorem in the next subsection. One-step (general-sum) game is a game with only one state and one step. In a one-step game, each player chooses an action simultaneously and then receive it's own reward. The Nash equilibrium and NE-gap can be defined similarly in one-step games.

**Lemma E.2** (Lower bound for one-step game). *Suppose $A_i = 2k$, $i = 1, \ldots, m$ and $m \geq 4$. Then, there exists an absolute constant $c_0$ such that for any $\varepsilon \leq 0.4$ and any online finetuning algorithm that outputs a **pure** strategy $\widehat{\pi} = (\widehat{\pi}_1, \ldots, \widehat{\pi}_m)$, if the number of samples*

$$n \leq c_0 \frac{\sum_{i=1}^m A_i}{\varepsilon^2} = c_0 \frac{2km}{\varepsilon^2},$$

*then there exists a one-step game $M$ with stochastic reward, on which the algorithm suffers from $\varepsilon/4$-suboptimality, i.e.*

$$\mathbb{E}_M NE\text{-}Gap(\widehat{\pi}) = \mathbb{E}_M \max_i \left[ \max_{\pi_i} u_i(\pi_i, \widehat{\pi}_{-i}) - u_i(\widehat{\pi}) \right] \geq \varepsilon/4,$$

*where the expectation $\mathbb{E}_M$ is w.r.t. the randomness during the algorithm's execution within game M. $u_i(\pi)$ is the expected reward of the $i^{th}$ player when strategy $\pi$ are taken for each player.*

The proof of this lemma is also in the next subsection. In the proof, we first construct a class of one-step games which reward is Bernoulli$(\frac{1}{2})$ or Bernoulli$(\frac{1}{2}+\varepsilon)$ depending on the taken joint-action. The proportion of joint-actions with reward Bernoulli$(\frac{1}{2} + \varepsilon)$ is relatively small. Most importantly, every pure-strategy $\varepsilon$-approximate Nash equilibrium has reward Bernoulli$(\frac{1}{2} + \varepsilon)$. So in order to find an $\varepsilon$-approximate pure-strategy Nash equilibrium, we must explore sufficient joint-actions. The number of the joint-actions with reward Bernoulli$(\frac{1}{2} + \varepsilon)$ can be bounded by the covering number of $[2k]^m$ under Hamming distance. Then we use KL divergence decomposition (Lemma E.5) to argue rigorously that we need to explore sufficient joint-actions to get an $\varepsilon$-approximate pure-strategy Nash equilibrium.

The rest of this section is organized as follows: We first prove Lemma E.2 in Section E.1, and then prove the main Theorem E.1 in Section E.2.

**Discussions of Theorem E.1**   There's $\sum_{i=1}^m A_i$ dependency[4] in the lower bound of sample complexity for finding a pure-strategy $\varepsilon$-approximate Nash equilibrium in MCGs. This bound is novel and improves the existing result. The existing proof in sample complexity's lower bound of Markov games (Bai & Jin (2020)) relies on an reduction from Markov games to single-agent MDPs, so the existing lower bound's dependency on $A_i$ $(i = 1, \ldots, m)$ is $\max_{i \in [m]} A_i$ .

Here, we don't include $S$ factor in our lower bound. The difficulty is that the NE-gap only depends on the player with the most suboptimality. For a single-agent MDP, if the player can change the policy at each state to improve the expected cumulative reward by $\varepsilon$, then the player can change policy at all state to improve the expected cumulative reward to the utmost extent. In general-sum Markov games, at different state, maybe different players can change the policy for this state to improve his expected cumulative reward by $\varepsilon$. However, the definition NE-gap only allows one player to change the policy. This difference in nature makes $S$ and $\sum_{i=1}^m A_i$ incompatible in the lower bound.

If we consider another notion of suboptimality, i.e., changing maximum to summation:

$$\text{NE-gap}'(\pi) := \sum_{i \in [m]} \left[ \sup_{\mu_i} V_{1,i}^{\mu_i, \pi_{-i}}(s_1) - V_{1,i}^{\pi}(s_1) \right].$$

This definition of NE-gap is different from the previous definition. With NE-gap$'$, if each player can change his policy to improve his expected cumulative reward by $\varepsilon$, the NE-gap$'$ would be at

---

[4]We only prove this lower bound when $A_i$ all equal to $2k$. This case is representative.

least $m\varepsilon$. Then we can similarly define $\varepsilon$-approximate Nash equilibrium as the policy $\pi$ such that NE-gap$'(\pi) \leq \varepsilon$. We simply point out that with this new definition of NE-gap$'$ and $\varepsilon$-approximate Nash equilibrium, mimicking the proof of Theorem 2 in Dann & Brunskill (2015), we can prove the sample complexity's lower bound for learning a pure-strategy $\varepsilon$-approximate Nash equilibrium in Markov (cooperative) games is $\Omega(H^2 S \sum_{i=1}^{m} A_i / \varepsilon^2)$.

### E.1 PROOF OF LEMMA E.2

For convenience, we call the joint-action (in one-step game) that is a Nash equilibrium a Nash strategy. We begin with a special case of Lemma E.2, i.e. the case when $A_i = 2$ for all $i \in [m]$.

**Lemma E.3.** *Suppose $A_i = 2$, $i = 1, \ldots, m$ and $m \geq 4$. Then, there exists an absolute constant $c_0$ such that for any $\varepsilon \leq 0.4$ and any algorithm that outputs a pure strategy $\widehat{\pi} = (\widehat{\pi}_1, \ldots, \widehat{\pi}_m)$, if the number of samples*

$$n \leq c_0 \frac{2m}{\varepsilon^2},$$

*then there exists a one step game $M$ with stochastic reward on which the algorithm suffers from $\varepsilon/4$-suboptimality, i.e.,*

$$\mathbb{E}_M NE\text{-}gap(\widehat{\pi}) = \mathbb{E}_M \max_i \left[ \max_{\pi_i} u_i(\pi_i, \widehat{\pi}_{-i}) - u_i(\widehat{\pi}) \right] \geq \varepsilon/4,$$

*where the expectation $\mathbb{E}_M$ is w.r.t. the randomness during the algorithm's execution within game M. $u_i(\pi)$ is the expected reward of the $i^{th}$ player when strategy $\pi$ are taken for each player.*

The proof of this lemma further relies on the following lemma.

**Lemma E.4.** *There exists a one-step game for $m$ players where each player has two actions. The deterministic reward is $0$ or $1$ and the number of joint actions that have $1$ is at most $\frac{2^{m+1}}{m}$. Moreover, the only pure-strategy Nash equilibria are these joint actions which have reward $1$.*

**Proof of Lemma E.4** We use $r(\boldsymbol{a})$ to denote the reward of (joint) actions $\boldsymbol{a} \in \{1, 2\}^m$ and define hamming distance $d(\boldsymbol{a}, \boldsymbol{a}') = \#\{i : a_i \neq a_i'\}$. To ensure that pure-strategy Nash equilibria must have reward $1$, we only need to ensure that for a $\boldsymbol{a} \in \{1, 2\}^m$, there exists one $\boldsymbol{a}' \in \{1, 2\}^m$ such that

$$r(\boldsymbol{a}') = 1, \quad d(\boldsymbol{a}, \boldsymbol{a}') \leq 1.$$

In other words, the set $\{\boldsymbol{a} : r(\boldsymbol{a}) = 1\}$ is a 1-net of $\{1, 2\}^m$ under the distance $d(\cdot, \cdot)$. By the definition of covering number, we only need to prove

$$\mathcal{N}(\{1, 2\}^m, d, 1) \leq \frac{2^{m+1}}{m}.$$

Define $K(m, 1) = \mathcal{N}(\{1, 2\}^m, d, 1)$. By hamming code (Hamming (1950)), we know that for any integer $k \geq 1$,

$$K(2^k - 1, 1) = 2^{2^k - k - 1}.$$

Moreover, we also have $K(n, 1) \leq 2K(n - 1, 1)$ by adding 0 and 1 behind the 1-net of $\{0, 1\}^{n-1}$. Taking largest $k$ such that $2^k - 1 \leq n$ and iterating this construction on the Hamming code we get $K(m, 1) \leq 2^{m - \lceil \log_2(m+1) \rceil} \leq \frac{2^{m+1}}{m}$. This ends the proof. $\square$

The next lemma is KL divergence decomposition (Lemma 15.1 of (Lattimore & Szepesvári, 2020)]), we restate it in one-step games.

**Lemma E.5** (KL divergence decomposition in one-step games). *For any one-step games with stochastic reward and any algorithm. Let $X = (\boldsymbol{a}_1, r_1, \boldsymbol{a}_2, r_2, \ldots, \boldsymbol{a}_n, r_n)$, where $\boldsymbol{a}_k$ is the action (adaptively) chosen by the algorithm at the $k^{th}$ round and $r_k$ is the reward received at the $k^{th}$ round after $\boldsymbol{a}_k$ is taken. $\mathbb{P}$ and $\mathbb{Q}$ are two probability measure for the stochastic reward. Let $N(\boldsymbol{a})$ be the total number of actions $\boldsymbol{a}$ in the first $n$ rounds. Then*

$$\mathrm{KL}(X|_{\mathbb{P}} \| X|_{\mathbb{Q}}) = \sum_{\boldsymbol{a}} \mathbb{E}_{\mathbb{P}}[N(\boldsymbol{a})] \mathrm{KL}(\mathbb{P}(\cdot|\boldsymbol{a}) \| \mathbb{Q}(\cdot|\boldsymbol{a})).$$

Then we return to the proof of Lemma E.3. Suppose a game $M$ satisfies the condition in Lemma E.4, by permuting the actions of $M$, we get $2^m$ games. They all satisfy the condition in Lemma E.4. Suppose the reward of the $i$-th game is $r^{(i)}(\cdot)$ $(i = 1, \cdots, 2^m)$.

We consider the following family of one-step games with stochastic reward: Let $\boldsymbol{a} = (a_1, a_2, \ldots, a_m)$.

$$\mathfrak{M}(\varepsilon) = \{\mathcal{M}^{(i)} \in \mathbb{R}^{\{1,2\}^m} \text{ with } \mathcal{M}^{(i)}(\boldsymbol{a}) = \frac{1}{2} + r^{(i)}(\boldsymbol{a})\varepsilon, \ i = 1, 2, \ldots, 2^m\},$$

where in one-step game $\mathcal{M}^{(i)}$, the reward is sampled from Bernoulli($\mathcal{M}^{(i)}(\boldsymbol{a})$) if the joint action is $\boldsymbol{a} = (a_1, a_2, \ldots, a_m)$. Moreover, we define $\mathcal{M}^{(0)}$ as a game whose reward is sampled from Bernoulli($\frac{1}{2}$) independent of the action.

We further let $\nu$ denote the uniform distribution on $\{1, 2, \ldots, 2^m\}$.

**Proof of Lemma E.3**    Fix a one-step game $\mathcal{M}^{(i)} \in \mathfrak{M}(\varepsilon)$, by Lemma E.4, it's clear that pure-strategy Nash equilibria form a set $D^{(i)} := \boldsymbol{a} \in \{\boldsymbol{a} : r^{(i)}(\boldsymbol{a}) = 1\}$. We have $\#D^{(i)} \leq 2^{m+1}/m$. For any online finetuning algorithm $\mathcal{A}$ that outputs a pure strategy $\widehat{\pi}$. Suppose $\widehat{\pi}$ takes joint-action $\widehat{\boldsymbol{a}} = (\widehat{a}_1, \widehat{a}_2, \cdots, \widehat{a}_m)$. From the structure of the $\mathfrak{M}(\varepsilon)$, we know

$$\text{NE-Gap}(\widehat{\pi}) = \varepsilon \mathbb{P}_i(\widehat{\boldsymbol{a}} \notin D^{(i)}),$$

where $\mathbb{P}_i$ is w.r.t. the randomness of the algorithm and game $\mathcal{M}^{(i)}$. So we only need to use $n$, the number of samples to give a lower bound of $\mathbb{P}_i(\widehat{\boldsymbol{a}} \notin D^{(i)})$.

Since $X = \{\boldsymbol{a}_1, r_1, \cdots, \boldsymbol{a}_n, r_n\}$ is a sufficient statistics for the posterior distribution $D^{(i)}$, we have

$$\mathbb{E}_{i\sim\nu}\mathbb{P}_i(\widehat{\boldsymbol{a}} \notin D^{(i)}) \geq \inf_g \mathbb{E}_{i\sim\nu}\mathbb{P}_i(g(X) \notin D^{(i)})$$

$$\overset{(i)}{\geq} \inf_g \mathbb{E}_{i\sim\nu}\mathbb{P}_0(g(X) \notin D^{(i)}) - \mathbb{E}_{i\sim\nu}\text{TV}(X|_{\mathbb{P}_0}, X|_{\mathbb{P}_i})$$

$$\overset{(ii)}{\geq} \frac{1}{2} - \mathbb{E}_{i\sim\nu}\text{TV}(X|_{\mathbb{P}_0}, X|_{\mathbb{P}_i})$$

$$\overset{(iii)}{\geq} \frac{1}{2} - \mathbb{E}_{i\sim\nu}\sqrt{\frac{1}{2}\text{KL}(X|_{\mathbb{P}_0}\|X|_{\mathbb{P}_i})}$$

$$\overset{(iv)}{=} \frac{1}{2} - \mathbb{E}_{i\sim\nu}\sqrt{\frac{1}{2}\sum_{\boldsymbol{a}}\mathbb{E}_{\mathbb{P}_0}[N(\boldsymbol{a})]\text{KL}(\mathbb{P}_0(\cdot|\boldsymbol{a})\|\mathbb{P}_i(\cdot|\boldsymbol{a}))}.$$

Above, (i) follows from the definition of total variation distance; (ii) uses the fact that under $\mathbb{P}_0$, we can't get any information about $\boldsymbol{a}^\star$, so $\mathbb{E}_{i\sim\nu}\mathbb{P}_0(g(X) \notin D^{(i)}) = 1 - \frac{|D^{(1)}|}{2^m} \geq \frac{1}{2}$; (iii) uses Pinsker's inequality; (iv) uses KL divergence decomposition (Lemma E.5).

For $\boldsymbol{a} \in D^{(i)}$, we have

$$\text{KL}(\mathbb{P}_0(\cdot|\boldsymbol{a})\|\mathbb{P}_i(\cdot|\boldsymbol{a})) = \text{KL}\left(\left(\frac{1}{2}, \frac{1}{2}\right)\|\left(\frac{1}{2} - \varepsilon, \frac{1}{2} + \varepsilon\right)\right)$$

$$= \frac{1}{2}\log\frac{1}{1 - 4\varepsilon^2}$$

$$\leq 4\varepsilon^2,$$

if $\varepsilon \in (0, 0.4]$. For $\boldsymbol{a} \notin D^{(i)}$, $\text{KL}(\mathbb{P}_0(\cdot|\boldsymbol{a})\|\mathbb{P}_i(\cdot|\boldsymbol{a})) = 0$. So

$$\mathbb{E}_{i\sim\nu}\mathbb{P}_i(\widehat{\boldsymbol{a}} \notin D^{(i)}) \geq \frac{1}{2} - \mathbb{E}_{i\sim\nu}\sqrt{2\sum_{\boldsymbol{a}\in D^{(i)}}\mathbb{E}_{\mathbb{P}_0}[N(\boldsymbol{a})]\varepsilon^2}$$

$$\overset{(i)}{\geq} \frac{1}{2} - \sqrt{2\mathbb{E}_{i\sim\nu}\sum_{\boldsymbol{a}\in D^{(i)}}\mathbb{E}_{\mathbb{P}_0}[N(\boldsymbol{a})]\varepsilon^2}$$

$$\overset{(ii)}{=} \frac{1}{2} - \sqrt{2\frac{n|D^{(i)}|}{2^m}\varepsilon^2}$$

$$\geq \frac{1}{2} - \sqrt{4\frac{n}{m}\varepsilon^2}.$$

Above, $(i)$ uses Jensen's inequality, $(ii)$ uses the fact that $\mathbb{E}_{\mathbb{P}_0}[N(\boldsymbol{a})]$ is independent of $i$ which gives

$$\mathbb{E}_{i\sim\nu}\sum_{\boldsymbol{a}\in D^{(i)}}\mathbb{E}_{\mathbb{P}_0}[N(\boldsymbol{a})] = \frac{1}{2^m}\sum_{\boldsymbol{a}}\mathbb{E}_{\mathbb{P}_0}[N(\boldsymbol{a})]\sum_{i:\boldsymbol{a}\in D^{(i)}}1$$

$$= \frac{1}{2^m}n|D^{(1)}|,$$

where $\sum_{i:\boldsymbol{a}\in D^{(i)}}1 = |D^{(1)}|$ is because of the permutation. Finally, we choose $c_0 = \frac{1}{32}$, if $n \leq \frac{1}{32}\frac{2m}{\varepsilon^2}$, then

$$\mathbb{E}_{i\sim\nu}\mathbb{P}_i(\widehat{\boldsymbol{a}} \notin D^{(i)}) \geq \frac{1}{4}.$$

So there's a game instance $\mathcal{M}^{(i)}$ on which the algorithm suffer from $\varepsilon/4$ sub-optimality. $\qquad\square$

The difference between Lemma E.3 and E.2 is the size of action space. To prove the $A_i = 2k$ case, we need to generalized E.4 to the case each player have $2k$ actions.

**Lemma E.6.** *For all positive integers $m$ and $k$. There exists a one-step game for $m$ players where each player has $2k$ actions. The deterministic reward is $0$ or $1$ and the number of joint actions that have 1 is at most $\frac{2\cdot(2k)^m}{km}$. Moreover, the only pure-strategy Nash equilibria are these joint actions which has reward 1.*

**Proof of Lemma E.6** We would prove this lemma by induction. Without loss of generality, we suppose the action for each player is $1, 2, \ldots, 2k$. First, we define the hamming distance between two vectors.

$$d(\boldsymbol{x}, \boldsymbol{y}) := \#\{i : \boldsymbol{x}(i) \neq \boldsymbol{y}(i)\}.$$

The joint action space is $[2k]^m = \{1, 2, \ldots, 2k\}^m$. Suppose we have an 1-net of $[2k]^m$, $D$, which means for every $y \in [2k]^m$, we can find a $\boldsymbol{x} \in D$, s.t. $d(\boldsymbol{x}, \boldsymbol{y}) \leq 1$. If the reward of a game satisfy:

$$r(\boldsymbol{a}) = 1 \text{ for all } \boldsymbol{a} \in D, \quad \text{and} \quad r(\boldsymbol{a}) = 0 \text{ for all } \boldsymbol{a} \notin D.$$

Then for every pure strategy $\boldsymbol{a}$ which has reward 0, we can find another pure strategy $\boldsymbol{a}'$ s.t. $r(\boldsymbol{a}') = 1$ and $d(\boldsymbol{a}, \boldsymbol{a}') = 1$. This means that one player can change to obtain higher reward. So $\boldsymbol{a}$ is not a pure-strategy Nash equilibrium.

As a result, we can construct a game based on the 1-net. The only thing left to be verified is that the number of joint actions that have reward 1 is at most $\frac{2\cdot(2k)^m}{km}$. Note that the number of joint actions that have reward 1 is actually $|D|$, so we need to prove

$$\mathcal{N}([2k]^m, d, 1) \leq \frac{2 \cdot (2k)^m}{km}, \tag{17}$$

where $\mathcal{N}(\cdot, d, \varepsilon)$ denotes the covering number.

For $k = 1$, from the proof of Lemma E.4, (17) is true. For $k > 1$, we first decompose $[2k]^m$ into $k^m$ smaller blocks, i.e.

$$\Omega(l_1, l_2, \ldots, l_m) = \{2l_1 - 1, 2l_1\} \times \{2l_2 - 1, 2l_2\} \times \cdots, \times\{2l_m - 1, 2l_m\},$$

where $\{l_1, l_2, \ldots, l_m\} \in [k]^m$. Among these $k^m$ blocks, we choose the blocks which satisfy $k|l_1 + l_2 + \cdots + l_m$. Apparently, there are $k^{m-1}$ blocks satisfying $m|l_1 + l_2 + \cdots + l_m$. Because (17) is true for $k = 1$, we can pick a 1-net of $\Omega(1, 1, \ldots, 1)$ with at most $\frac{2\cdot2^m}{m}$ elements. By a translation, (all elements add a constant vector), we can pick a 1-net of each block with at most $\frac{2\cdot2^m}{m}$ elements. Totally there are at most $\frac{2\cdot2^m k^{m-1}}{m} = \frac{2\cdot(2k)^m}{km}$ elements. These elements form a set $P$. We would prove $P$ is a 1-net of $[2k]^m$.

In fact, for any $\boldsymbol{x} \in [2k]^m$, suppose $\boldsymbol{x}$ is in $\Omega(l_1, l_2, \ldots, l_m)$. After translating this block to $\Omega(1, 1, \ldots, 1)$, $\boldsymbol{x}$ is coincided with $\boldsymbol{x_0}$.

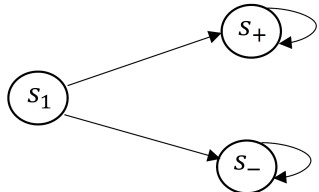

Figure 1: A class of hard-to-learn MDPs where $s_+$ and $s_-$ are absorbing state.

If $\boldsymbol{x_0} \in P$, change $l_1$ to $l'_1$ such that $k | l'_1 + l_2 + \cdots + l_m$. Suppose $\boldsymbol{x}'$ is the vector in block $\Omega(l'_1, l_2, \ldots, l_m)$ that corresponds to $\boldsymbol{x_0}$ in $\Omega(1, 1, \ldots, 1)$. This gives $\boldsymbol{x}' \in P$. Moreover, $d(\boldsymbol{x}, \boldsymbol{x}') \leq 1$ because the translation from $\Omega(l_1, l_2, \ldots, l_m)$ to $\Omega(l'_1, l_2, \ldots, l_m)$ only changes the first component.

If $\boldsymbol{x_0} \notin P$, we can find $\boldsymbol{x'_0}$ in $P \cap \Omega(1, 1, \ldots, 1)$ such that $d(\boldsymbol{x_0}, \boldsymbol{x'_0}) = 1$ because $P|_{\Omega(1,1,\ldots,1)}$ is a 1-net. Suppose $\boldsymbol{x_0}$ and $\boldsymbol{x'_0}$ differs at the $j$-th component. We can change $l_j$ to $l'_j$ s.t. $k | l_1 + l_2 + \cdots + l_{j-1} + l'_j + l_{j+1} + \cdots + l_m$. Suppose $\boldsymbol{x'_0}$ corresponds to $\boldsymbol{x}'$ in $\Omega(l_1, \ldots, l_m)$ and $\boldsymbol{x}''$ in $\Omega(l_1, \ldots, l_{j-1}, l'_j, l_{j+1}, \ldots, l_m)$. By the definition of $P$, we know $\boldsymbol{x}'' \in P$. Moreover, $\boldsymbol{x}$ and $\boldsymbol{x}'$ only differ at the $j$-th component because of translation and the fact that $\boldsymbol{x_0}$ and $\boldsymbol{x'_0}$ only differ at the $j$-th component. $\boldsymbol{x}'$ and $\boldsymbol{x}''$ also only differ at the $j$-th component because of translation. So we have $\boldsymbol{x}$ and $\boldsymbol{x}''$ may only differ at the $j$-th component, i.e. $d(\boldsymbol{x}, \boldsymbol{x}'') \leq 1$.

Recall that $\boldsymbol{x}'' \in P$. To conclude, we can always find another vector $\boldsymbol{y} \in P$ such that $d(\boldsymbol{x}, \boldsymbol{y}) \leq 1$, this means $P$ is a 1-net of $[2k]^m$. So (17) is proved. $\qquad\square$

Using this fundamental lemma and suppose a game $M$ satisfies the condition in Lemma E.6, by permuting the actions of $M$, we get $(2k)^m$ games. They all satisfy the condition in Lemma E.6. Suppose the reward of the $i$-th game is $r^{(i)}(\cdot)$ $(i = 1, \cdots, (2k)^m)$ .

We consider the following family of one-step games with stochastic reward: Let $\boldsymbol{a} = (a_1, a_2, \ldots, a_m)$.

$$\mathfrak{M}(\varepsilon) = \{\mathcal{M}^{(i)} \in \mathbb{R}^{[2k]^m} \text{ with } \mathcal{M}^{(i)}(\boldsymbol{a}) = \frac{1}{2} + r^{(i)}(\boldsymbol{a})\varepsilon, \ i = 1, 2, \ldots, (2k)^m\}, \qquad (18)$$

where in one-step game $\mathcal{M}^{(i)}$, the reward is sampled from $\text{Bernoulli}(\mathcal{M}^{(i)}(\boldsymbol{a}))$ if the joint action is $\boldsymbol{a} = (a_1, a_2, \ldots, a_m)$. Moreover, we define $\mathcal{M}^{(0)}$ as a game whose reward is sampled from $\text{Bernoulli}(\frac{1}{2})$ independent of the action.

Then by the same argument in proof of Lemma E.3, we can easily prove Lemma E.2.

### E.2 PROOF OF THEOREM E.1

We are now ready to prove Theorem E.1 based on Lemma E.2.

Because $S \geq 3$, we construct a class of general-sum Markov games as follows (see figure 1): the set of all joint-actions $\mathcal{A}$ can be divided into two sets $\mathcal{A}^+$ an $\mathcal{A}^-$.

**Transition** Starting from $s_1$, for $\boldsymbol{a} \in \mathcal{A}^+$, $\mathbb{P}_1(s_+|s_1, \boldsymbol{a}) = \frac{1}{2} + \frac{\varepsilon}{2(H-1)}$ and $\mathbb{P}_1(s_-|s_1, \boldsymbol{a}) = \frac{1}{2} - \frac{\varepsilon}{2(H-1)}$. For $\boldsymbol{a} \in \mathcal{A}^-$, $\mathbb{P}_1(s_+|s_1, \boldsymbol{a}) = \mathbb{P}_1(s_-|s_1, \boldsymbol{a}) = \frac{1}{2}$. For all $h \geq 1$ and $\boldsymbol{a} \in \mathcal{A}$, $\mathbb{P}_h(s_+|s_+, \boldsymbol{a}) = 1$ and $\mathbb{P}_h(s_-|s_-, \boldsymbol{a}) = 1$.

**Rewards** For any $\boldsymbol{a}$ and $i \in [m]$, $r_{1,i}(s, \boldsymbol{a}) = 0$ $r_{h,i}(s_+, \boldsymbol{a}) = 1$ and $r_{h,i}(s_-, \boldsymbol{a}) = 0$, where $h > 1$. This also implies the Markov game is cooperative.

If we choose $\mathcal{A}^+$ as the joint action with reward $\text{Bernoulli}(\frac{1}{2} + \varepsilon)$ in $\mathcal{M}^{(i)}$ defined in (18). By the construction, for any pure-strategy policy $\pi$, if the joint actions taken at step 1 is $\boldsymbol{a} \notin \mathcal{A}^+$, we have

---

**Algorithm 8** Mixed-expert FTRL for weighted adversarial bandits

---

**Require:** Hyper-parameters $\{\eta_t\}_{1 \le t \le T}$, $\{\gamma_t\}_{1 \le t \le T}$. Sequence $\{\alpha_t^i\}_{1 \le i \le t \le T}$.
1: **Initialize**: Accumulators $t_{b'} \leftarrow 0$ for all $b' \in [A]$.
2: **for** $t = 1, 2, \ldots, T$ **do**
3:     Compute weight $u_t \leftarrow \alpha_t^t / \alpha_t^1$.
4:     Compute action distributions for all sub-experts $b' \in [A]$ by FTRL:

$$q^{b'}(a) \propto_a \exp\left( -(\eta_{t_{b'}}/u_t) \cdot \sum_{\tau=1}^{t_{b'}} w_\tau(b') \widehat{\ell}_\tau^{b'}(a) \right).$$

5:     Compute $p^t \in \Delta_{[A]}$ by solving the linear system $p^t(\cdot) = \sum_{b'=1}^{A} p^t(b') q^{b'}(\cdot)$.
6:     Sample sub-expert $b \sim p^t$.
7:     Sample action $a^t \sim q^b$ from sub-expert $b$.
8:     Play the action $a^t$, and observe the (bandit-feedback) loss $\tilde{\ell}_t(a^t)$.
9:     Update accumulator for sub-expert $b$: $t_b \leftarrow t_b + 1$.
10:    Compute the loss estimate $\widehat{\ell}_{t_b}^b(\cdot) \in \mathbb{R}_{\ge 0}^A$ as

$$\widehat{\ell}_{t_b}^b(a) \leftarrow \frac{\tilde{\ell}_{t_b}(a)\mathbf{1}\{a = a^t\}}{q^b(a) + \gamma_{t_b}}.$$

11:    Set $w_{t_b}(b) \leftarrow u_t$.

---

NE-gap$(\pi) \ge \varepsilon$. The only useful information in each episode is the transition at step 1. Transition from $s_1$ to $s_+$ can be viewed as getting a 1 reward and transition from $s_1$ to $s_-$ can be viewed as getting a 0 reward. So learning pure-strategy Nash equilibrium of this class of Markov games is equivalent to learning pure-strategy Nash equilibrium of a game for $m$ players with stochastic reward. As a result, by Lemma E.2, if the number of episodes

$$K \le c_0 \frac{\sum_{i=1}^{m} A_i}{\varepsilon^2/(H-1)^2} \le 4c_0 \frac{H^2 \sum_{i=1}^{m} A_i}{\varepsilon^2},$$

then there exists general-sum Markov cooperative game $MG$ on which the algorithm suffers from $\varepsilon/4$-suboptimality, i.e.

$$\mathbb{E}_M \text{NE-gap}(\widehat{\pi}) \ge \varepsilon/4.$$

This finish the proof of Theorem E.1.        □

## F   Adversarial Bandit with Low Weighted Swap Regret

In this section, we describe and analyze our main algorithm for adversarial bandits with low weighted swap regret. Our Algorithm, Mixed-expert Follow-The-Regularized-Leader (FTRL) for weighted adversarial bandits, is described in Algorithm 8.

**Problem setting** Throughout this section, we consider a standard (adversarial) bandit problem with action space is $[A] = \{1, \ldots, A\}$ for some $A \ge 1$. We assume that the realized loss values $\tilde{\ell}_t \in [0,1]^A$, and let $\ell_t(a) := \mathbb{E}_t[\tilde{\ell}_t(a)]$ denote the expected loss conditioned on (as usual) all information *before* the sampling of the action $a^t$ in Line 6 & 7. The hyperparameters in Algorithm 8 are chosen as

$$\eta_t = \gamma_t = \sqrt{\iota/(At)},$$

where the log factor

$$\iota := 4\log(8HAT/p). \tag{19}$$

---

## F.1 MAIN RESULT FOR ADVERSARIAL BANDIT WITH LOW WEIGHTED SWAP REGRET

We first define a strategy modification. A strategy modification is a function $F : [A] \to [A]$ which can also be applied to any action distribution $\mu \in \Delta_A$, such that $F \diamond \mu$ gives the swap distribution which takes action $F(a)$ with probability $\mu(a)$.

The swap regret (Blum & Mansour, 2007; Ito, 2020) measures the difference between the cumulative realized loss for the algorithm and that for swapped action sequences generated by an arbitrary strategy modification $F$. Here, we consider a weighted version of the swap regret with some non-negative weights $\{\alpha_t^i\}_{1 \le i \le t}$, defined as

$$R_{\text{swap}}(t) := \max_{F:[A] \to [A]} \sum_{i=1}^{t} \alpha_t^i [\ell_i(a^i) - \ell_i(F(a^i))]. \tag{20}$$

We will also consider a slightly modified version of the swap regret used in our analyses for learning CE, defined as

$$\widetilde{R}_{\text{swap}}(t) := \max_{F:[A] \to [A]} \sum_{i=1}^{t} \alpha_t^i [\ell_i(a^i) - \langle F \diamond p^i, \ell_i \rangle], \tag{21}$$

where $p^t$ is $t$-th action distribution (from which the action $a^t$ is sampled from) played by the algorithm.

We now state our main result of this section.

**Lemma F.1** (Bound on weighted swap regret). *If we execute Algorithm 8 for $T$ rounds and the weights $\{\alpha_t^i\}$ are chosen according to (2), then with probability at least $1 - p/2$, we have the following bounds on the swap regret simultaneously for all $t \in [T]$:*

$$R_{\text{swap}}(t) = \max_{F:[A] \to [A]} \sum_{i=1}^{t} \alpha_t^i [\ell_i(a^i) - \ell_i(F(a^i))] \le C(HA\sqrt{\iota/t} + HA\iota/t),$$

$$\widetilde{R}_{\text{swap}}(t) = \max_{F:[A] \to [A]} \sum_{i=1}^{t} \alpha_t^i [\ell_i(a^i) - \langle F \diamond p^i, \ell_i \rangle] \le C(HA\sqrt{\iota/t} + HA\iota/t),$$

*where $C > 0$ is some absolute constant.*

The rest of this section is devoted to proving Lemma F.1, organized as follows. We first present some important properties of Algorithm 8 in Section F.2. We then prove Lemma F.1 in Section F.3 using new auxiliary results on weighted adversarial bandits with predictable weights. Lastly, these auxiliary results are stated and proved in Appendix F.4.

## F.2 PROPERTIES OF ALGORITHM 8

**Solution of the linear system** In Line 5, we need to compute $p^t \in \Delta_{[A]}$ by solving the linear system $p^t(\cdot) = \sum_{b'=1}^{A} p^t(b') q^{b'}(\cdot)$. Such $p^t(\cdot)$ can be interpreted as the stationary distribution of a Markov chain over $[A]$ with transition probability $\mathbb{P}(b|a) = q^a(b)$. This guarantees the existence of $p^t \in \Delta_{[A]}$ such that $p^t(\cdot) = \sum_{b'=1}^{A} p^t(b') q^{b'}(\cdot)$. Moreover, $p^t(\cdot)$ can be computed efficiently (see e.g. Cohen et al. (2017b)).

**FTRL update for each sub-expert** Here we show that, the updates for any particular sub-expert $b \in [A]$ in Algorithm 8 is exactly equivalent to an FTRL update (with loss sequence being the ones only over the episodes in which $b$ is sampled) which we summarize in Algorithm 9. This is important as our proof relies on reducing the weighted swap regret to a linear combination of weighted regrets for each sub-expert $b \in [A]$.

To see this, fix any $b \in [A]$. When $b$ is sampled in Line 6 at episode $t$, we have accumulator $t_b$ and weight $w_{t_b}(b) = u_t = \alpha_t^t/\alpha_t^1$ at the end of episode $t$. Action $a^t$ is chosen from distribution

$$q^b(a) \propto_a \exp\left(-(\eta_{t_b}/w_{t_b}) \cdot \sum_{\tau=1}^{t_b-1} w_\tau(b')\widehat{\ell}_\tau^b(a)\right)$$

after $w_{t_b}(b) = u_t = \alpha_t^t / \alpha_t^1$ is observed. The loss estimate $\widehat{\ell}_{t_b}$ is

$$\widehat{\ell}_{t_b}^b(a) \leftarrow \frac{\widetilde{\ell}_{t_b}(a)\mathbf{1}\{a = a^t\}}{q^b(a) + \gamma_{t_b}}.$$

So from the sub-expert $b$'s perspective, she is performing FTRL (follow the regularized leader) with changing step size and random weight (We summarize this in Algorithm 9). Suppose the sub-expert $b$ is chosen at episode $t = k_1, k_2, \ldots, k_{t_b}$, then the weighted regret for sub-expert $b$ becomes

$$R_t(b) := \sup_{\theta^* \in \Delta_A} \left[ \sum_{\tau=1}^{t_b} w_\tau(b)(\ell_{k_\tau}(a^{k_\tau}) - \langle \theta^*, \ell_{k_\tau} \rangle) \right]. \tag{22}$$

**Bounded weight** $w_i(b)$   Recall $\{\alpha_t^i\}$ is chosen as in (2). We have the following bound:

$$\frac{\alpha_t^t}{\alpha_t^1} \leq \frac{1}{\alpha_t^1} = \frac{1}{\alpha_1(1 - \alpha_2)\cdots(1 - \alpha_t)} = \frac{(H+2)(H+3)\cdots(H+t)}{1 \cdot 2 \cdots (t-1)} \leq (H+t)^{H+1}.$$

So for any $t \leq T$, we have

$$\alpha_t^t / \alpha_t^1 \leq (H+T)^{2H} \equiv W. \tag{23}$$

The weight $w_{n_b}(b) = \alpha_t^t / \alpha_t^1$ have a (non-random) upper bound $W$.

### F.3   PROOF OF LEMMA F.1

We use $b^i$ to denote the sampled sub-expert $b$ at the $i$-th episode. Define $\mathcal{G}_i$ as the $\sigma$-algebra generated by all the random variables observed up to the end of the $i$-th episode.

First observe that for $R_{\text{swap}}(t)$ we have the bound

$$R_{\text{swap}}(t) = \max_{F:[A]\to[A]} \sum_{i=1}^t \alpha_t^i[\ell_i(a^i) - \ell_i(F(a^i))]$$

$$\leq \underbrace{\max_{F:[A]\to[A]} \sum_{i=1}^t \alpha_t^i[\ell_i(a^i) - \ell_i(F(b^i))]}_{\text{I}} + \underbrace{\max_{F:[A]\to[A]} \sum_{i=1}^t \alpha_t^i[\ell_i(F(b^i)) - \ell_i(F(a^i))]}_{\text{II}},$$

also, for $\widetilde{R}_{\text{swap}}(t)$ we have the bound

$$\widetilde{R}_{\text{swap}}(t) = \max_{F:[A]\to[A]} \sum_{i=1}^t \alpha_t^i[\ell_i(a^i) - \langle F \diamond p^i, \ell_i \rangle]$$

$$\leq \underbrace{\max_{F:[A]\to[A]} \sum_{i=1}^t \alpha_t^i[\ell_i(a^i) - \ell_i(F(b^i))]}_{\text{I}} + \underbrace{\max_{F:[A]\to[A]} \sum_{i=1}^t \alpha_t^i[\ell_i(F(b^i)) - \langle F \diamond p^i, \ell_i \rangle]}_{\widetilde{\text{II}}},$$

Term II and $\widetilde{\text{II}}$ can be bounded by concentration in a similar fashion; here we first focus on term II. Observe that at the $i$-th episode, as $p^i$ obtained in Line 5 solves the equation

$$p^i(a) = \sum_{b'=1}^A p^i(b')q^{b'}(a),$$

we have $b^i \sim p^i(\cdot)$ and $a^i \sim q^{b^i}(\cdot)$ has the same marginal distribution conditioned on $\mathcal{G}_{i-1}$, where $\mathcal{G}_{i-1}$ denote the $\sigma$-algebra generated by $\ell_i$ and all the random variables in the first $i-1$ episodes. Therefore, fixing any strategy modification $F : [A] \to [A]$, we have $\{\alpha_t^i[\ell_i(F(b^i)) - \ell_i(F(a^i))]\}_{1\leq i\leq t}$ is a bounded martingale difference sequence w.r.t. the filtration $\{\mathcal{G}_i\}$, so by Azuma-Hoeffding inequality and $\sum_{i=1}^t (\alpha_t^j)^2 \leq 2H/t$,

$$\mathbb{P}\left( \sum_{i=1}^t \alpha_t^i[\ell_i(F(b^i)) - \ell_i(F(a^i))] \geq 2\sqrt{\frac{H \log 1/p}{t}} \right) \leq p.$$

As there is at most $A^A$ strategy modifications, we can substitute $p$ with $p/(4A^AT)$, and take a union bound to get

$$\text{II} = \max_{F:[A]\to[A]} \sum_{i=1}^{t} \alpha_t^i[\ell_i(F(b^i)) - \ell_i(F(a^i))] \leq 2\sqrt{HA\log(AT/p)/t} \leq C\sqrt{HA\iota/t} \qquad (24)$$

simultaneously for all $t \in [T]$ with probability at least $1 - p/4$. We also note that, by a similar argument (as $b^i$ is also distributed according to $p^i$ conditioned on the past), we have that

$$\widetilde{\text{II}} = \max_{F:[A]\to[A]} \sum_{i=1}^{t} \alpha_t^i[\ell_i(F(b^i)) - \langle F \diamond p^i, \ell_i \rangle] \leq C\sqrt{HA\iota/t}. \qquad (25)$$

Therefore for bounding both $R_{\text{swap}}(t)$ and $\widetilde{R}_{\text{swap}}(t)$, it suffices to bound term I.

We next bound term I. Define $\mathcal{U}_b := \{i \in [t] : b^i = b\}$ and let $n_b^t$ be the value of $t_b$ at the end of the $t$-th episode, i.e. $n_b^t = \#\{i : b^i = b\}$. We also suppose the sub-expert $b$ was chosen at episode $t = k_1^b, k_2^b, \ldots, k_{n_b^t}^b$ up to episode $t$. Then we have

$$\sum_{i=1}^{t} \alpha_t^i[\ell_i(a^i) - \ell_i(F(b^i))] = \sum_{b\in\mathcal{A}} \sum_{i\in\mathcal{U}_b} \alpha_t^i[\ell_i(a^i) - \ell_i(F(b))]$$

$$= \sum_{b\in\mathcal{A}} \sum_{\tau=1}^{n_b^t} \alpha_t^1 \frac{\alpha_t^{k_\tau^b}}{\alpha_t^1}[\ell_{k_\tau^b}(a^{k_\tau^b}) - \ell_{k_\tau^b}(F(b))]$$

$$= \sum_{b\in\mathcal{A}} \sum_{\tau=1}^{n_b^t} \alpha_t^1 w_\tau(b)[\ell_{k_\tau^b}(a^{k_\tau^b}) - \ell_{k_\tau^b}(F(b))].$$

Here the last equation is because our choice of $w_\tau(b) = \alpha_{k_\tau^b}^{k_\tau^b}/\alpha_{k_\tau^b}^1 = \alpha_t^{k_\tau^b}/\alpha_t^1$ which is a simple corollary from the definition of $\alpha_t^i$ in Eq. (3).

Then because

$$\sum_{\tau=1}^{n_b^t} w_\tau(b)[\ell_{k_\tau^b}(a^{k_\tau^b}) - \ell_{k_\tau^b}(F(b))] \leq \max_{\theta^*} \sum_{\tau=1}^{n_b^t} w_\tau(b)[\ell_{k_\tau^b}(a^{k_\tau^b}) - \langle \ell_{k_\tau^b}, \theta^* \rangle] = R_t(b),$$

where $R_t(b)$ defined in (22) is the weighted regret $R_t(b)$ for sub-expert $b$, we can use our result on weighted adversarial bandits with predictable weights (Lemma F.2) to bound this term (The upper bound $W$ of the weight $w_\tau(b)$ can be taken as $(H + T)^{2H}$ by the calculation of (23). Moreover, $w_\tau(b) = \alpha_t^{k_\tau^b}/\alpha_t^1$ and $\{\alpha_t^j\}_{j=1}^{t}$ is increasing, so $\{w_\tau(b)\}_{\tau\geq 1}$ is non-decreasing.). Recall that our choice of log term is $\iota = 4\log\frac{4HAT}{p} = \log\frac{AT\lceil\log_2 W\rceil}{p'}$ where $p' \leq p/(4A)$. Thus by Lemma F.2, with probability at least $1 - p/(4A)$,

$$R_t(b) \leq 15 \max_{\tau\leq n_b^t} w_\tau(b)\left[\sqrt{An_b^t\iota} + \iota\right] \quad \text{for all } t \in [T].$$

Taking a union bound, we get

$$R_t(b) \leq 15 \max_{\tau\leq n_b^t} w_\tau(b)\left[\sqrt{An_b^t\iota} + \iota\right] \quad \text{for all } (t, b) \in [T] \times [A] \qquad (26)$$

with probability at least $1 - p/4$.

On this event, we have

$$\max_{F:[A]\to[A]} \sum_{i=1}^{t} \alpha_t^i[\ell_i(a^i) - \ell_i(F(b^i))] = \max_{F:[A]\to[A]} \sum_{b\in\mathcal{A}} \sum_{\tau=1}^{n_b^t} \alpha_t^1 w_\tau(b)[\ell_{k_\tau^b}(a^{k_\tau^b}) - \ell_{k_\tau^b}(F(b))]$$

$$\leq \sum_{b \in \mathcal{A}} \max_{F:[A] \to [A]} \sum_{\tau=1}^{n_b^t} \alpha_t^1 w_\tau(b) [\ell_{k_\tau^b}(a_\tau^{k_\tau^b}) - \ell_{k_\tau^b}(F(b))]$$

$$\leq \sum_{b \in \mathcal{A}} \alpha_t^1 R_t(b)$$

$$\overset{(i)}{\leq} \sum_{b \in \mathcal{A}} \alpha_t^1 \max_{\tau \leq n_b^t} w_\tau(a)(15\sqrt{An_b^t \iota} + 15\iota)$$

$$\overset{(ii)}{=} \sum_{b \in \mathcal{A}} \alpha_t^1 \frac{\alpha_t^{k_{n_b^t}^b}}{\alpha_t^1}(15\sqrt{An_b^t \iota} + 15\iota)$$

$$\overset{(iii)}{\leq} \sum_{a \in \mathcal{A}} \frac{2H}{t}(15\sqrt{An_b^t \iota} + 15\iota).$$

Here, (i) uses (26), (ii) uses the $\alpha_t^{t'}$ is increasing w.r.t. $t'$ and (iii) uses $\max_{0 \leq t' \leq t} \alpha_t^{t'} \leq 2H/t$. Finally, because $\sum_{b \in \mathcal{A}} n_b^t = t$, by the concavity of $x \mapsto \sqrt{x}$, we have with probability at least $1 - p/4$,

$$\begin{aligned} \mathrm{I} = \max_{F:[A] \to [A]} \sum_{i=1}^{t} \alpha_t^i [\ell_i(a^i) - \ell_i(F(b^i))] &\leq \frac{30H}{t} \cdot \left( \sum_{b \in \mathcal{A}} (\sqrt{An_b^t \iota} + \iota) \right) \\ &\leq \frac{30H}{t} \cdot (A\sqrt{t\iota} + A\iota) \\ &= 30HA\sqrt{\iota/t} + 30HA\iota/t. \end{aligned}$$

Combining this with (24) and (25), we finish the proof. $\qquad \square$

### F.4 AUXILIARY LEMMAS FOR WEIGHTED ADVERSARIAL BANDIT WITH PREDICTABLE WEIGHTS

In this subsection, we consider the Follow the Regularized Leader (FTRL) algorithm (Lattimore & Szepesvári, 2020) with

1. changing step size,
2. weighted regret with $\mathcal{F}_{t-1}$-measurable weights and loss distributions,
3. high probability regret bound.

We present these results because the predictable weights we would use are potentially unbounded from above; if weights are predictable and also bounded, then there may be an easier analysis.

**Interaction protocol** We first describe the interaction protocol between the environment and the player for this problem. At each episode $t$, the environment adversarially choose a weight $w_t$ which takes values in $\mathbb{R}_{>0}$, and distributions $(\mathcal{L}_{at})_{a \in [A]}$, where each $\mathcal{L}_{at}$ is a distribution that takes values in $\mathcal{P}([0,1])$ (the space of distributions supported on $[0,1]$). Then the player receives the weight $w_t$, chooses an action $a_t$, and observes a loss $\tilde{\ell}_t(a_t) \sim \mathcal{L}_{a_t,t}$. The action chosen $a_t$ can be based on all the information in $\mathcal{D}_t \equiv \{(a_i, w_i, \tilde{\ell}_i(a_i))_{i \leq t-1}, w_t\}$. Then, the environment can observe the player's action $a_t$ and the incurred loss realization $\tilde{\ell}_t(a_t)$, and use these information and some external randomness $z_t$ to choose the weight and distributions $(w_{t+1}, (\mathcal{L}_{a,t+1})_{a \in [A]})$ of the next episode. We will consider the following variant of FTRL algorithm for the player.

We denote $\ell_t \equiv (\mathbb{E}_{\tilde{\ell} \sim \mathcal{L}_{at}}[\tilde{\ell}])_{a \in [A]} \in [0,1]^A$ which is a random vector but is $\sigma((\mathcal{L}_{at})_{a \in [A]})$-measurable. For some $\theta^* \in \mathbb{R}^A$, the regret of the player up to episode $t$ is defined as

$$R_t \equiv \sum_{i=1}^{t} w_i(\ell_i(a_i) - \langle \theta^*, \ell_i \rangle).$$

---

**Algorithm 9** FTRL for Weighted Regret with Changing Step Size and Predictable Weights

1: **for** episode $t = 1, \ldots, T$ **do**
2:     Observe $w_t > 0$.
3:     $\theta_t(a) \propto_a \exp(-(\eta_t/w_t) \sum_{i=1}^{t-1} w_i \widehat{\ell}_i(a))$.
4:     Take action $a_t \sim \theta_t(\cdot)$, and observe loss $\tilde{\ell}_t(a_t)$.
5:     $\widehat{\ell}_t(a) \leftarrow \tilde{\ell}_t(a) \mathbf{1}\left\{a_t = a\right\} / (\theta_t(a) + \gamma_t)$ for all $a$.

---

Let $(z_t)_{t \geq 1}$ be a sequence of external random variables which are identically and independently distributed as $\text{Unif}([0,1])$ and are independent of all other random variables. We define $\mathcal{F}_t = \sigma(\{a_i, \tilde{\ell}_i, z_i\}_{i \leq t})$ to be the sigma algebra generated by the random chosen action $a_i$, the random loss $\tilde{\ell}_i(a_i)$, and the external random variables $z_i$ by episode $t$. We assume that the random weights and the random loss distributions $(w_t, (\mathcal{L}_{at})_{a \in [A]})_{t \geq 1}$ are a $(\mathcal{F}_t)_{t \geq 1}$-*predictable sequence*, in the sense that $(w_t, (\mathcal{L}_{at})_{a \in [A]})$ is $\mathcal{F}_{t-1}$-measurable. Then $\mathcal{F}_t$ contains all information (random chosen action, random observed loss, random weight, and random loss distributions) before action $a_{t+1}$ is taken.

We assume the predictable sequence $(w_t)_{1 \leq t \leq T}$ have a global (non-random) upper bound $W$. Then we define the log term $\iota = \log(AT\lceil \log_2 W \rceil / p)$. We set

$$\eta_t = \gamma_t = \sqrt{\frac{\iota}{At}}.$$

### F.4.1    REGRET BOUND

In the following, we consider to give a high probability weighted regret bound for Algorithm 9.

**Lemma F.2.** *Let* $(w_t, (\mathcal{L}_{at})_{a \in [A]})_{t \geq 1}$ *be any* $\mathcal{F}_t$-*predictable sequence satisfying* $1 \leq \min_{i \leq t} w_i \leq \max_{i \leq t} w_i \leq W$ *for some constant (non-random)* $W > 0$ *almost surely. Moreover, suppose* $w_i$ *is non-decreasing. Then, following Algorithm 9, with probability at least* $1 - 4p$, *for any* $\theta^* \in \Delta^A$ *and* $t \leq T$ *we have*

$$\sum_{i=1}^{t} w_i(\ell_i(a^i) - \langle \theta^*, \ell_i \rangle) \leq 15 \max_{i \leq t} w_i \cdot \left[ \sqrt{At\iota} + \iota \right],$$

*where* $\iota = \log(AT\lceil \log_2 W \rceil / p)$.

*Proof.* This lemma follows from the bound in Lemma F.3 and a concentration step that we establish below.

Define $M = \lceil \log_2 W \rceil$, and $w_i(k) = w_i \mathbf{1}\left\{w_i \leq 2^k\right\}$. Then $w_i(k)$ is also $\mathcal{F}_{i-1}$ measurable. We Consider sequence $\left\{w_i(k)(\ell_i(a^i) - \langle \theta_i, \ell_i \rangle)\right\}_{i \geq 1}$. Since $w_i$ is $\mathcal{F}_{i-1}$-measurable, $\mathbb{E}[w_i(\ell_i(a^i) - \langle \theta_i, \ell_i \rangle)|\mathcal{F}_{i-1}] = 0$, which means $\left\{w_i(k)(\ell_i(a^i) - \langle \theta_i, \ell_i \rangle)\right\}_{i \geq 1}$ is a martingale difference sequence w.r.t. filtration $(\mathcal{F}_i)$. So by Azuma-Hoeffding inequality,

$$\mathbb{P}\left( \sum_{i=1}^{t} w_i(k)(\ell_i(a^i) - \langle \theta_i, \ell_i \rangle) \leq \sqrt{2 \cdot \iota t (2^k)^2} \right) \geq 1 - \frac{p}{TM}.$$

Taking a union bound, we have

$$\mathbb{P}\left( \forall k \in [M], \sum_{i=1}^{t} w_i(k)(\ell_i(a^i) - \langle \theta_i, \ell_i \rangle) \leq 2^k \sqrt{2t\iota} \right) \geq 1 - p/T. \tag{27}$$

Denote $k' = \lceil \log_2 \max_{i \leq t} w_i \rceil$, then we have

$$\left\{ \forall k \in [M], \sum_{i=1}^{t} w_i(k)(\ell_i(a^i) - \langle \theta_i, \ell_i \rangle) \leq 2^k \sqrt{2t\iota} \right\}$$

$$\subseteq \left\{ \sum_{i=1}^{t} w_i(k')(\ell_i(a^i) - \langle \theta_i, \ell_i \rangle) \leq 2^k \sqrt{2t\iota} \right\} \subseteq \left\{ \sum_{i=1}^{t} w_i(\ell_i(a^i) - \langle \theta_i, \ell_i \rangle) \leq 2 \max_{i \leq t} w_i \sqrt{2t\iota} \right\}.$$

Then probability bound gives

$$\mathbb{P}\left(\sum_{i=1}^{t} w_i(\ell_i(a^i) - \langle \theta_i, \ell_i \rangle) \le 2 \max_{i \le t} w_i \sqrt{2t\iota}\right) \ge 1 - p/T.$$

Taking a union bound, we get

$$\sum_{i=1}^{t} w_i(\ell_i(a^i) - \langle \theta_i, \ell_i \rangle) \le 2 \max_{i \le t} w_i \sqrt{2t\iota} \quad \text{for all } t \in [T]$$

with probability at least $1 - p$. Summing the above and the regret bound shown in Lemma F.3, we finish the proof. $\qquad\square$

**Lemma F.3.** *Let $(w_t, (\mathcal{L}_{at})_{a \in [A]})_{t \ge 1}$ be any $\mathcal{F}_t$-predictable sequence satisfying $1 \le \min_{i \le t} w_i \le \max_{i \le t} w_i \le W$ for some constant (non-random) $W > 0$ almost surely. Moreover, suppose $w_i$ is non-decreasing. Then, following Algorithm 9, with probability at least $1 - 3p$, for any $\theta^* \in \Delta^A$ and $t \le T$ we have*

$$\sum_{i=1}^{t} w_i \langle \theta_i - \theta^*, \ell_i \rangle \le 10 \max_{i \le t} w_i \cdot \left[\sqrt{At\iota} + \iota\right],$$

*where $\iota = \log(AT\lceil \log_2 W \rceil / p)$.*

*Proof.* The regret $R_t(\theta^*)$ can be decomposed into three terms

$$R_t(\theta^*) = \sum_{i=1}^{t} w_i \langle \theta_i - \theta^*, \ell_i \rangle$$
$$= \underbrace{\sum_{i=1}^{t} w_i \left\langle \theta_i - \theta^*, \widehat{\ell}_i \right\rangle}_{(A)} + \underbrace{\sum_{i=1}^{t} w_i \left\langle \theta_i, \ell_i - \widehat{\ell}_i \right\rangle}_{(B)} + \underbrace{\sum_{i=1}^{t} w_i \left\langle \theta^*, \widehat{\ell}_i - \ell_i \right\rangle}_{(C)}$$

and we bound $(A)$ in Lemma F.5, $(B)$ in Lemma F.6 and $(C)$ in Lemma F.7.

Setting $\eta_t = \gamma_t = \sqrt{\frac{\iota}{At}}$, the conditions in Lemma F.5 and Lemma F.7 are satisfied. Putting them together and take union bound, we have with probability $1 - 3p$

$$R_t(\theta^*) \le \frac{w_t \log A}{\eta_t} + \frac{A}{2} \sum_{i=1}^{t} \eta_i w_i + \max_{i \le t} w_i \iota + A \sum_{i=1}^{t} \gamma_i w_i + 2 \max_{i \le t} w_i \sqrt{2t\iota} + 2 \max_{i \le t} w_i \iota / \gamma_t$$
$$\le \max_{i \le t} w_i \left[\sqrt{A\iota t} + \frac{3\sqrt{A\iota}}{2} \sum_{i=1}^{t} \frac{1}{\sqrt{t}} + \iota + 2\sqrt{2t\iota} + 2\sqrt{At\iota}\right]$$
$$\le 10 \max_{i \le t} w_i \left[\sqrt{At\iota} + \iota\right] \quad \text{for all } t \in [T]$$

This proves the lemma. $\qquad\square$

The rest of this section is devoted to the proofs of the Lemmas used in the proofs of Lemma F.3. We begin the following useful lemma adapted from Lemma 1 in Neu (2015), which is crucial in constructing high probability guarantees.

**Lemma F.4.** *For any predictable sequence of coefficients $c_1, c_2, \ldots, c_t$ s.t. $c_i \in [0, 2\gamma_i]^A$ w.r.t. $(\mathcal{F}_i)_{i \ge 1}$ and fixing t, we have with probability at least $1 - p/(AT)$,*

$$\sum_{i=1}^{t} w_i \left\langle c_i, \widehat{\ell}_i - \ell_i \right\rangle \le 2 \max_{i \le t} w_i \iota$$

*Proof.* Define $M = \lceil \log_2 W \rceil$, and $w_i(k) = w_i \mathbf{1}\{w_i \leq 2^k\}$. By definition,

$$
w_i(k)\widehat{\ell}_i(a) = \frac{w_i(k)\tilde{\ell}_i(a)\mathbf{1}\{a_i = a\}}{\theta_i(a) + \gamma_i} \leq \frac{w_i(k)\tilde{\ell}_i(a)\mathbf{1}\{a_i = a\}}{\theta_i(a) + \frac{w_i(k)\tilde{\ell}_i(a)\mathbf{1}\{a_i=a\}}{2^k}\gamma_i}
$$

$$
= \frac{2^k}{2\gamma_i}\frac{\frac{2\gamma_i w_i(k)\tilde{\ell}_i(a)\mathbf{1}\{a_i=a\}}{2^k\theta_i(a)}}{1 + \frac{\gamma_i w_i(k)\tilde{\ell}_i(a)\mathbf{1}\{a_i=a\}}{2^k\theta_i(a)}} \overset{(i)}{\leq} \frac{2^k}{2\gamma_i}\log\left(1 + \frac{2\gamma_i w_i(k)\tilde{\ell}_i(a)\mathbf{1}\{a_i = a\}}{2^k\theta_i(a)}\right)
$$

where $(i)$ is because $\frac{z}{1+z/2} \leq \log(1+z)$ for all $z \geq 0$.

Defining the sum

$$
\widehat{S}_i = \frac{w_i(k)}{2^k}\left\langle c_i, \widehat{\ell}_i \right\rangle, \quad S_i = \frac{w_i(k)}{2^k}\left\langle c_i, \ell_i \right\rangle,
$$

Then $S_i$ is $\mathcal{F}_{i-1}$-measurable since $w_i, c_i, \ell_i$ are $\mathcal{F}_{i-1}$-measurable. Using $\mathbb{E}_i[\cdot]$ to denote the conditional expectation $\mathbb{E}[\cdot|\mathcal{F}_i]$, we have

$$
\mathbb{E}_{i-1}\left[\exp\left(\widehat{S}_i\right)\right] \leq \mathbb{E}_{i-1}\left[\exp\left(\sum_a \frac{c_i(a)}{2\gamma_i}\log\left(1 + \frac{2\gamma_i w_i(k)\tilde{\ell}_i(a)\mathbf{1}\{a_i = a\}}{2^k\theta_i(a)}\right)\right)\right]
$$

$$
\overset{(i)}{\leq} \mathbb{E}_{i-1}\left[\prod_a\left(1 + \frac{c_i(a)w_i(k)\tilde{\ell}_i(a)\mathbf{1}\{a_i = a\}}{2^k\theta_i(a)}\right)\right]
$$

$$
= \mathbb{E}_{i-1}\left[1 + \sum_a \frac{c_i(a)w_i(k)\tilde{\ell}_i(a)\mathbf{1}\{a_i = a\}}{2^k\theta_i(a)}\right]
$$

$$
= 1 + S_i \leq \exp(S_i)
$$

where $(i)$ is because $z_1\log(1 + z_2) \leq \log(1 + z_1 z_2)$ for any $0 \leq z_1 \leq 1$ and $z_2 \geq -1$. Here we are using the condition $c_i(a) \leq 2\gamma_i$ to guarantee the condition is satisfied.

Equipped with the above bound, we can now prove the concentration result.

$$
\mathbb{P}\left[\sum_{i=1}^t\left(\widehat{S}_i - S_i\right) \geq \iota\right] \leq \mathbb{P}\left[\exp\left[\sum_{i=1}^t\left(\widehat{S}_i - S_i\right)\right] \geq \frac{ATM}{p}\right]
$$

$$
\leq \frac{p}{ATM}\mathbb{E}_{t-1}\left[\exp\left[\sum_{i=1}^t\left(\widehat{S}_i - S_i\right)\right]\right]
$$

$$
\leq \frac{p}{ATM}\mathbb{E}_{t-2}\left[\exp\left[\sum_{i=1}^{t-1}\left(\widehat{S}_i - S_i\right)\right]E_{t-1}\left[\exp\left(\widehat{S}_t - S_t\right)\right]\right]
$$

$$
\leq \frac{p}{ATM}\mathbb{E}_{t-2}\left[\exp\left[\sum_{i=1}^{t-1}\left(\widehat{S}_i - S_i\right)\right]\right]
$$

$$
\leq \cdots \leq \frac{p}{ATM}.
$$

So we have

$$
\mathbb{P}\left(\sum_{t=1}^t w_i(k)\left\langle c_i, \widehat{\ell}_i - \ell_i\right\rangle \leq 2^k\iota\right) \geq 1 - p/(ATM).
$$

Taking a union bound,

$$
\mathbb{P}\left(\forall k \in [M], \sum_{t=1}^t w_i(k)\left\langle c_i, \widehat{\ell}_i - \ell_i\right\rangle \leq 2^k\iota\right) \geq 1 - p/(AT). \tag{28}
$$

Denote $k' = \lceil \log_2 \max_{i \leq t} w_i \rceil$, and note that

$$\left\{ \forall k \in [M], \sum_{t=1}^{t} w_i(k) \left\langle c_i, \widehat{\ell}_i - \ell_i \right\rangle \leq 2^k \iota \right\}$$

$$\subseteq \left\{ \sum_{t=1}^{t} w_i(k') \left\langle c_i, \widehat{\ell}_i - \ell_i \right\rangle \leq 2^{k'} \iota \right\} \subseteq \left\{ \sum_{t=1}^{t} w_i \left\langle c_i, \widehat{\ell}_i - \ell_i \right\rangle \leq 2 \max_{i \leq t} w_i \iota \right\}.$$

Therefore, we have

$$\mathbb{P}\left( \sum_{i=1}^{t} w_i \left\langle c_i, \widehat{\ell}_i - \ell_i \right\rangle \leq 2 \max_{i \leq t} w_i \iota \right) \geq 1 - p/(AT).$$

This proves the lemma. $\qquad\square$

Using Lemma F.4, we can bound the $(A)(B)(C)$ separately as below.

**Lemma F.5.** *If $\eta_i \leq 2\gamma_i$ for all $i \leq t$ and $\{\eta_i/w_i\}_{i \geq 1}$ is non-increasing, with probability $1 - p$, for any $t \in [T]$ and $\theta^* \in \Delta^A$,*

$$\sum_{i=1}^{t} w_i \left\langle \theta_i - \theta^*, \widehat{\ell}_i \right\rangle \leq \frac{w_t \log A}{\eta_t} + \frac{A}{2} \sum_{i=1}^{t} \eta_i w_i + \frac{1}{2} \max_{i \leq t} w_i \iota.$$

*Proof.* We use the standard analysis of FTRL with changing step size, see for example Exercise 28.12(b) in Lattimore & Szepesvári (2020). Notice the essential step size is $\eta_t/w_t$ which is non-increasing and the essential loss vector is $w_t\widehat{\ell}_t$, we have

$$\sum_{i=1}^{t} w_i \left\langle \theta_i - \theta^*, \widehat{\ell}_i \right\rangle \leq \frac{w_t \log A}{\eta_t} + \sum_{i=1}^{t} w_i \left[ \langle \theta_i - \theta_{i+1}, \ell_i \rangle - \frac{\mathrm{KL}(\theta_{i+1} \| \theta_i)}{\eta_i} \right].$$

We claim that

$$\left\langle \theta_i - \theta_{i+1}, \widehat{\ell}_i \right\rangle - \frac{\mathrm{KL}(\theta_{i+1} \| \theta_i)}{\eta_i} \leq \frac{\eta_i}{2} \left\langle \theta_i, \widehat{\ell}_i^2 \right\rangle. \tag{29}$$

In fact, applying Theorem 26.13 in Lattimore & Szepesvári (2020) gives that

$$\left\langle \theta_i - \theta_{i+1}, \widehat{\ell}_i \right\rangle - \frac{\mathrm{KL}(\theta_{i+1} \| \theta_i)}{\eta_i} \leq \frac{\eta_i}{2} \left\langle u_i \theta_i + (1 - u_i)\theta_{i+1}, \widehat{\ell}_i^2 \right\rangle,$$

for some $u_i \in [0, 1]$. Note that our $\widehat{\ell}_i$ only have at most one non-zero entry, i.e. $\widehat{\ell}_i(a) > 0$ if and only if $a = a_i$. If $\theta_i(a_i) \geq \theta_{i+1}(a_i)$, we have

$$\frac{\eta_i}{2} \left\langle u_i \theta_i + (1 - u_i)\theta_{i+1}, \widehat{\ell}_i^2 \right\rangle = \frac{\eta_i}{2} [u_i \theta_i(a_i) + (1 - u_i)\theta_{i+1}(a_i)]\widehat{\ell}_i(a_i)^2$$

$$\leq \frac{\eta_i}{2} \theta_i(a_i)\widehat{\ell}_i(a_i)^2 = \frac{\eta_i}{2} \left\langle \theta_i, \widehat{\ell}_i^2 \right\rangle.$$

Otherwise, $\theta_i(a_i) < \theta_{i+1}(a_i)$, so we have $\left\langle \theta_i - \theta_{i+1}, \widehat{\ell}_i \right\rangle = [\theta_i(a_i) - \theta_{i+1}(a_i)]\widehat{\ell}_i(a_i) \leq 0$ and $-\mathrm{KL}(\theta_{i+1} \| \theta_i)/\eta_i \leq 0$. In both cases, (29) is correct. As a result,

$$\sum_{i=1}^{t} w_i \left\langle \theta_i - \theta^*, \widehat{\ell}_i \right\rangle \leq \frac{w_t \log A}{\eta_t} + \frac{1}{2} \sum_{i=1}^{t} \eta_i w_i \left\langle \theta_i, \widehat{l}_i^2 \right\rangle$$

$$\leq \frac{w_t \log A}{\eta_t} + \frac{1}{2} \sum_{i=1}^{t} \sum_{a \in \mathcal{A}} \eta_i w_i \widehat{\ell}_i(a)$$

$$\overset{(i)}{\leq} \frac{w_t \log A}{\eta_t} + \frac{1}{2} \sum_{i=1}^{t} \sum_{a \in \mathcal{A}} \eta_i w_i \ell_i(a) + \max_{i \leq t} w_i \iota$$

$$\leq \frac{w_t \log A}{\eta_t} + \frac{A}{2} \sum_{i=1}^{t} \eta_i w_i + \max_{i \leq t} w_i \iota$$

where $(i)$ is by using Lemma F.4 with $c_i(a) = \eta_i$. The any-time guarantee is justified by taking union bound. $\qquad \square$

**Lemma F.6.** *With probability* $1 - p$, *for any* $t \in [T]$,

$$\sum_{i=1}^{t} w_i \left\langle \theta_i, \ell_i - \widehat{\ell}_i \right\rangle \leq A \sum_{i=1}^{t} \gamma_i w_i + 2 \max_{i \leq t} w_i \sqrt{2t\iota}.$$

*Proof.* We further decompose the left side into

$$\sum_{i=1}^{t} w_i \left\langle \theta_i, \ell_i - \widehat{\ell}_i \right\rangle = \sum_{i=1}^{t} w_i \left\langle \theta_i, \ell_i - \mathbb{E}_{i-1}[\widehat{\ell}_i] \right\rangle + \sum_{i=1}^{t} w_i \left\langle \theta_i, \mathbb{E}_{i-1}[\widehat{\ell}_i] - \widehat{\ell}_i \right\rangle.$$

The first term is bounded by

$$\begin{aligned}
\sum_{i=1}^{t} w_i \left\langle \theta_i, \ell_i - \mathbb{E}_{i-1}[\widehat{\ell}_i] \right\rangle &= \sum_{i=1}^{t} w_i \left\langle \theta_i, \ell_i - \frac{\theta_i}{\theta_i + \gamma_i} \ell_i \right\rangle \\
&= \sum_{i=1}^{t} w_i \left\langle \theta_i, \frac{\gamma_i}{\theta_i + \gamma_i} \ell_i \right\rangle \leq A \sum_{i=1}^{t} \gamma_i w_i.
\end{aligned}$$

To bound the second term, we use similar argument in the proof of Lemma F.4 , we define $w = \max_{i \leq t} w_i$, $M = \lceil \log_2 W \rceil$. and $w_i(k) = w_i \mathbf{1}\{w_i \leq 2^k\}$, notice

$$\left\langle \theta_i, \widehat{\ell}_i \right\rangle \leq \sum_{a \in \mathcal{A}} \theta_i(a) \frac{\mathbf{1}\{a_t = a\}}{\theta_i(a) + \gamma_i} \leq \sum_{a \in \mathcal{A}} \mathbf{1}\{a_i = a\} = 1,$$

thus $\{w_i(k) \left\langle \theta_i, \mathbb{E}_{i-1}[\widehat{\ell}_i] - \widehat{\ell}_i \right\rangle\}_{i=1}^{t}$ is a bounded martingale difference sequence w.r.t. the filtration $\{\mathcal{F}_i\}_{i=1}^{t}$. By Azuma-Hoeffding inequality,

$$\sum_{i=1}^{t} w_i(k) \left\langle \theta_i, \mathbb{E}_{i-1}[\widehat{\ell}_i] - \widehat{\ell}_i \right\rangle \leq \sqrt{2\iota \sum_{i=1}^{t} w_i(k)^2} \leq \sqrt{2\iota \cdot t (2^k)^2}$$

with probability at least $1 - p/TM$. Taking a union bound, we get

$$\sum_{i=1}^{t} w_i(k) \left\langle \theta_i, \mathbb{E}_{i-1}[\widehat{\ell}_i] - \widehat{\ell}_i \right\rangle \leq \sqrt{2\iota \cdot t 2^{2k}}, \quad \text{for all } (t, k) \in [T] \times [M]$$

with probability at least $1 - p$. On this event, choosing $k' = \lceil \log_2 w \rceil$, we have

$$\begin{aligned}
\sum_{i=1}^{t} w_i \left\langle \theta_i, \mathbb{E}_{i-1}[\widehat{\ell}_i] - \widehat{\ell}_i \right\rangle &\leq \sum_{i=1}^{t} w_i(k') \left\langle \theta_i, \mathbb{E}_{i-1}[\widehat{\ell}_i] - \widehat{\ell}_i \right\rangle \\
&\leq 2^{k'} \sqrt{2\iota \cdot t} \leq 2 \max_i w_i \sqrt{2\iota \cdot t}.
\end{aligned}$$

This ends the proof. $\qquad \square$

**Lemma F.7.** *With probability* $1 - p$, *for any* $t \in [T]$ *and any* $\theta^* \in \Delta^A$, *if* $\gamma_i$ *is non-increasing in* $i$,

$$\sum_{i=1}^{t} w_i \left\langle \theta^*, \widehat{\ell}_i - \ell_i \right\rangle \leq 2 \max_{i \leq t} w_i \iota / \gamma_t.$$

*Proof.* Define a basis $\{e_j\}_{j=1}^A$ of $\mathbb{R}^A$ by

$$
e_j(a) = \begin{cases} 1 \text{ if } a = j, \\ 0 \text{ otherwise.} \end{cases}
$$

Then for all the $j \in [A]$, apply Lemma F.4 with $c_i = \gamma_t e_j$. Since now $c_i(a) \leq \gamma_t \leq \gamma_i$, the condition in Lemma F.4 is satisfied. As a result, for any $t \in [T]$ and $j \in [A]$, we have with probability at least $1 - p/(TA)$ that

$$
\sum_{i=1}^t w_i \left\langle e_j, \widehat{\ell}_i - \ell_i \right\rangle \leq 2 \max_{i \leq t} w_i \iota / \gamma_t.
$$

Taking a union bound, we have with probability at least $1 - p$,

$$
\sum_{i=1}^t w_i \left\langle e_j, \widehat{\ell}_i - \ell_i \right\rangle \leq 2 \max_{i \leq t} w_i \iota / \gamma_t \quad \text{for all } (j, t) \in [m] \times [T].
$$

Since any $\theta^*$ is a convex combination of $\{e_j\}_{j=1}^A$, on this event, we also have

$$
\sum_{i=1}^t w_i \left\langle \theta^*, \widehat{\ell}_i - \ell_i \right\rangle \leq 2 \max_{i \leq t} w_i \iota / \gamma_t \quad \text{for all } t \in [T].
$$

This finishes the proof. □

