# OpenReview forum: "When Can We Learn General-Sum Markov Games with a Large Number of Players Sample-Efficiently?"
_ICLR.cc/2022/Conference — ICLR 2022 Poster_

### Official Review · Reviewer_bdjf · 2021-10-30

**Correctness:** 4
**Technical Novelty And Significance:** 2
**Empirical Novelty And Significance:** Not applicable
**Recommendation:** 8
**Confidence:** 3

**Main Review:**

Strength: The paper presents the first sample efficient algorithm for multi-player general sum Markov game and prove its convergence to CCE and CE. Though there are many *similar* types of paper in the field of theoretical reinforcement learning (i.e., a new setting with new sample efficient algorithm etc.), I found this one interesting. It solves an important question and the technique is non-trivial.



Minor issues:
(1) Proposition 1 and Theorem A.2 are results of Rubinstein et al. 2016. Though the authors make this clear enough, I suggest them make it cleaner (e.g. merge into one single proposition), since it is not the contribution of this paper, and I think it is only used to motivated this paper (hence no need to expand with one page.)

(2) There are some recent paper that accelerates the classic no regret algorithms that approach CE or CCE. Though it is not directly related, I suggest cite them as they are closely related to this paper.

[1] Syrgkanis, Vasilis, et al. "Fast Convergence of Regularized Learning in Games." Advances in Neural Information Processing Systems 28 (2015)

[2] Chen, Xi, and Binghui Peng. "Hedging in games: Faster convergence of external and swap regrets." Advances in Neural Information Processing Systems 33 (2020).

**Summary Of The Paper:**

The paper proposes algorithm for learning coarse correlated equilibrium (CCE) and correlated equilibrium (CE) of multi-player general-Sum Markov Games, the propose algorithm has polynomial dependence on the number of state and the horizons. The proposed algorithm builds upon the Nash V-learning of (Bai et al. 2020), and incorporate several new ideas.

**Summary Of The Review:**

Overall, I think it is interesting paper with solid contribution. I vote for acceptance.


------------------------------------------------------
Post rebuttal:

The author revised their paper and addressed some minor issues. My positive evaluation for the paper remains and I vote for acceptance.

---

> ### Author Response · Authors · 2021-11-18
> **Response to Reviewer bdjf**
>
> Thank you for your positive feedback on our paper! We respond to your comments as follows.
> > Proposition 1 and Theorem A.2 are results of Rubinstein et al. 2016. Though the authors make this clear enough, I suggest them make it cleaner… hence no need to expand with one page.
>
> We acknowledge that Proposition 1 and Theorem A.2 are results in Rubinstein et al. 2016 instead of our contribution. Our original submission included a full version in the appendix (for slightly less than one page), as their result requires some formal definitions about query complexity that we did not feel worth the extra length in the main text (since it’s their contribution rather than ours), so we instead presented an informal statement in Proposition 1 and deferred the full setup and results into Appendix A.
> In our revision we additionally highlighted this by changing Theorem A.2 -> Proposition A.2, and added “full statement deferred to Proposition A.2” in Proposition 1. We have also slightly shortened the appendix.
>
> > There are some recent paper that accelerate the classic no regret algorithms that approach CE or CCE. Though it is not directly related, I suggest cite them as they are closely related to this paper.
>
> Thank you for pointing out these works about no regret learning in classic games, which are indeed very relevant. We have properly cited them in our revision.

---

### Official Review · Reviewer_Qf6T · 2021-11-02

**Correctness:** 4
**Technical Novelty And Significance:** 3
**Empirical Novelty And Significance:** Not applicable
**Recommendation:** 8
**Confidence:** 4

**Main Review:**

Strengths:
1. The sample complexity of  $\epsilon$-CE for general-sum games is significant and worth publishing.
2. The proof of Theorem 6 is novel to me. In particular, the authors utilize the weighted swap regret lemma to derive the gap to the best CE agent.

Weaknesses:
1. The result of $\epsilon$-CCE (Theorem 3) is based on previous results. It is not surprising given previous analysis about Nash-V learning (Tian 2021, et. al,)
2. As pointed by the authors, the dependences on $S$ and $H$ are not tight yet.

**Summary Of The Paper:**

This paper studies the problem of finite horizon multi-agent general-sum Markov games.  The proposed algorithm CE-V-Learning achieves $\epsilon$-coarse correlated equilibirum (CCE) using $\tilde{O}(H^5S\max_{1\leq i \leq m}A_i/\epsilon^2)$  episodes, and $\epsilon$-correlated equilibrium (CE) using $\tilde{O}(H^6S\max_{1\leq i \leq m}A_i/\epsilon^2)$  episodes. The sample complexity is polynomial in $\max_{1\leq i\leq m}A_i$, while previous results had an exponential dependence on $m$ ($\Pi_{1\leq i\leq m}A_i$).

**Summary Of The Review:**

Overall, the paper is well written and the theorectical results are worth publishing.

---

> ### Author Response · Authors · 2021-11-18
> **Response to Reviewer Qf6T**
>
> Thank you for your positive feedback on our paper! We agree with the comments and we believe tightening our sample complexities (for example, the dependence on $H$) would indeed be an interesting direction for future work.

---

### Official Review · Reviewer_6yah · 2021-11-02

**Correctness:** 4
**Technical Novelty And Significance:** 4
**Empirical Novelty And Significance:** Not applicable
**Recommendation:** 8
**Confidence:** 4

**Main Review:**

This is technical paper with very impressive results. I am really surprised with the running time of the provided algorithm for convergence of their algorithm to Nash equilibria in Markov potential games ($1/\epsilon^3$ improving the $1/\epsilon^6$). Note though that the updates are not simultaneous, in the sense that one agent updates at a time. Moreover, the analysis and lower bounds look quite nice. I feel this paper should get accepted!

**Summary Of The Paper:**

The paper studies general sum episodic Markov Games. They provide an algorithm which is effectively V-learning with Follow the regularized leader subroutine (FTRL is hedge algorithm in their case). They prove convergence of their algorithm to $\epsilon$- CCE (coarse correlated equilibria) and $\epsilon$-CE equilibrium policies in $1/\epsilon^2$ with dependence on \max_{i} A_{i} (cardinality of action space of an individual rather than the whole action space). This happens due to ``independent updates" of the agents. Moreover for the case of potential markov games, they manage to show $1/\epsilon^3$ convergence to $\epsilon$-Nash pure Nash policies, improving Leonardos et al paper and solving an open question about convergence to deterministic policies.
However, the agents do not update simultaneously (i.e., the updates are not concurrent!). They also provide lower bounds.

**Summary Of The Review:**

I feel the paper has quite nice technical contributions. The results are very interesting too.

---

> ### Author Response · Authors · 2021-11-18
> **Response to Reviewer 6yah**
>
> Thank you for your positive feedback on our paper! We agree that our Nash-CA algorithm for Markov Potential Games is not concurrent (at each iteration, one specifically chosen player modifies the policy), and designing more sample-efficient algorithms with concurrent updates may be an interesting direction for future work.

---

### Official Review · Reviewer_q1AK · 2021-11-09

**Correctness:** 3
**Technical Novelty And Significance:** 3
**Empirical Novelty And Significance:** 3
**Recommendation:** 6
**Confidence:** 3

**Main Review:**

Strengths:
+ The sample complexity of learning Markov games is a timely and important topic in the areas of multi-agent reinforcement learning.
+ This paper provides new insights on what type of equilibria and/or structural assumptions on Markov games may admit sample-efficient learning polynomially in the size of the joint action spaces.

Weaknesses:
- Although the authors showed polynomial sample complexity for NE of Markov potential games, there appears to be a large gap to the lower bound.
- Some part of the paper is hard to follow and need better explanations.

**Summary Of The Paper:**

This paper studied the sample complexity for learning the coarse correlated equilibrium (CCE) and correlated equilibrium (CE) of m-player general-sum Markov games (MG), as well as learning the Nash equilibrium (NE) of Markov potential games (MPG). For MG, the authors proposed an algorithm with sample complexity that grows polynomially in $\max_{i\leq m} A_i$ to achieve $\epsilon$-approximate CCE and CE. For MPG, the authors proposed an algorithm with sample complexity that grows polynomially in $\sum_{i\leq m} A_i$ to achieve $\epsilon$-approximate NE.

**Summary Of The Review:**

1. This paper considered the sample complexity of learning multi-player Markov games with relaxed equilibrium notions (CE and CCE) and special structure (Markov potential games). The results in this paper are quite interesting. However, starting from Section 4, the presentation is a bit hard to follow. First, the algorithm for CCE, the first key equilibrium of MG, is relegated to the appendix. I went over the proof quickly and it appears a technique from [Tian et al. 2021] has been adopted to improve an H factor in [Bai et al. 2020]. However, [Bai et al. 2020] considered NE and it remains unclear what difference the properties of CCE make in this paper to allow the authors to achieve the claimed sample complexity.

2. The description of the CE algorithm is also unclear. It seems the FTRL technique is the key to the design of this algorithm. Is this technique brand new or adapted from some earlier work? Also, it might be better to define some jargons (e.g., sub-expert) for readers to fully understand the meaning of the algorithm. Also, it would be better to provide more intuition behind the algorithm design for CE. The authors mentioned that they adopted and modified the policy of [Bai et al. 2020]. It would be better if the authors could be more specific on what modifications are needed here.

3. Although the paper is mostly a theory paper, it would be nice if the authors could provide some experimental results to verify the sample complexity of the proposed algorithms.

---

> ### Author Response · Authors · 2021-11-18
> **Response to Reviewer q1AK**
>
> Thank you for your support of our paper and the valuable feedback! We respond to your comments as follows.
>
> > it appears a technique from [Tian et al. 2021] has been adopted to improve an H factor in [Bai et al. 2020]. However, [Bai et al. 2020] considered NE and it remains unclear what difference the properties of CCE make in this paper to allow the authors to achieve the claimed sample complexity.
>
> We agree that Tian et al.’s choice of hyperparameter is used to improve $H$, we have stated this in the appendix. We have revised our paper (Section 3, “overview of algorithm and techniques”) to also clarify this in the main text.
>
> Re what is the difference between NE considered in Bai et al. 2020, and the CCE we considered, and how did we achieve our CCE result given theirs: (1) The result in Bai et al. 2020 and the result in our paper are both built on the weighted regret bound for V-Learning (Lemma C.3) for every fixed state $s$, step $h$ and player $i$; (2) Bai et al. convert this to Nash by their certified policy, we convert it to CCE by our certified correlated policy. The two conversions are quite similar and modular. Therefore, whatever savings in the weighted regret bound part will directly translate to savings in the final bound for learning the equilibria; (3) Additionally, using the choice of hyperparameters in Tian et al. 2021 helps to save the $H$ factor in the weighted regret bound (Lemma C.3), and thus translates to saving an $H$ in the final guarantee for learning the CCE.
>
> > The description of the CE algorithm is also unclear. It seems the FTRL technique is the key to the design of this algorithm. Is this technique brand new or adapted from some earlier work?
>
> (1) The use of a FTRL-*like* subroutine for reinforcement learning is not new; indeed, an standard FTRL type subroutine was already used in the Nash V-Learning algorithm of Bai et al. 2020, which our algorithm builds on. (2) Our particular subroutine in our CE-V-Learning algorithm is a mixed-expert FTRL algorithm, which achieves a low swap regret guarantee. This mixed-expert FTRL algorithm along with its swap regret guarantee (for one-step games / strategic games) are known in the bandit / games literature (Blum & Mansour 2007, Ito 2020). However, to the best of our knowledge, this algorithm has not been adapted before in reinforcement learning (e.g. Markov Games) literature.
>
> > Also, it might be better to define some jargons (e.g., sub-expert) for readers to fully understand the meaning of the algorithm. Also, it would be better to provide more intuition behind the algorithm design for CE.
>
> Thank you for the suggestions! We have revised the paper to provide more intuitions about our algorithm design for CE, including a more detailed explanation on the meaning of the sub-expert.
>
> > The authors mentioned that they adopted and modified the policy of [Bai et al. 2020]... It would be better if the authors could be more specific on what modifications are needed here.
>
> Our certified correlated policy is adapted from Bai et al. 2020, but with a key difference that we output a *correlated policy* in order to learn a Correlated Equilibrium (CE/CCE), whereas Bai et al.’s original certified policy procedure outputs an individual policy for each player (and thus outputs a product policy overall) in order to learn the Nash. More specifically, our Algorithm 2 gives a correlated policy because the $k$ and $l$ (in Line 1 and line 4 of Algorithm 2) is sampled randomly and the same variable is used by all the players. In other words, our certified correlated policy is the same as running the certified (individual) policies of Bai et al. 2020 for all players, but with a common (shared) $k$ and $l$.
>
>
> We have added more explanations about the certified policy procedure on page 5.
>
> > it would be nice if the authors could provide some experimental results to verify the sample complexity of the proposed algorithms.
>
> We agree that some experimental evaluations (especially verifying the scalings within the theoretically shown sample complexities) are interesting. We believe the focus of the current paper is rather theoretical, and would like to leave such experimental evaluations as future work.

---

### Author Response · Authors · 2021-11-18
**Revision uploaded**

We thank all reviewers for their valuable feedback. We have uploaded a new revision of our submission to incorporate reviewers’ suggestion and include more explanations of our algorithms and proof techniques. For clarity, all our changes are marked in red.

---

### Decision · Program_Chairs · 2022-01-20

**Decision:**

Accept (Poster)

**Comment:**

This paper proposes algorithms for learning (coarse) correlated equilibrium in multi-agent general-sum Markov games, with improved sample complexities that are polynomial in the maximum size of the action sets of different players. This is a very solid work along the line of multi-agent reinforcement learning and there is unanimous support to accept this paper. Thus, I recommend acceptance.